# IndicTrans2: Towards High-Quality and Accessible Machine Translation Models for all 22 Scheduled Indian Languages

**Jay Gala**[*1]    **Pranjal A. Chitale**[*1,2]    **Raghavan AK**[1,2]    **Varun Gumma**[3†]    **Sumanth Doddapaneni**[1,2]
**Aswanth Kumar**[6†] **Janki Nawale**[1]    **Anupama Sujatha**[1]    **Ratish Puduppully**[7]    **Vivek Raghavan**[1,4‡]
**Pratyush Kumar**[1,2,3§]    **Mitesh M. Khapra**[1,2¶]    **Raj Dabre**[5]    **Anoop Kunchukuttan**[1,2,3]

[1]*Nilekani Centre at AI4Bharat*    [2]*Indian Institute of Technology Madras*    [3]*Microsoft*    [4]*EkStep Foundation*
[5]*National Institute of Information and Communications Technology, Kyoto, Japan*    [6]*Flipkart*
[7]*Institute for Infocomm Research ($I^2R$), A*STAR, Singapore*

Reviewed on OpenReview: https://openreview.net/forum?id=vfT4YuzAYA

## Abstract

India has a rich linguistic landscape, with languages from 4 major language families spoken by over a billion people. 22 of these languages listed in the Constitution of India (referred to as *scheduled languages*) are the focus of this work. Given the linguistic diversity, high-quality and *accessible* Machine Translation (MT) systems are essential in a country like India. Before this work, there was (i) no parallel training data spanning all 22 languages, (ii) no robust benchmarks covering all these languages and containing content relevant to India, and (iii) no existing translation models that support all 22 scheduled languages of India. In this work, we aim to address this gap by focusing on the missing pieces required for enabling wide, easy, and open access to good machine translation systems for all 22 scheduled Indian languages. We identify four key areas of improvement: curating and creating larger training datasets, creating diverse and high-quality benchmarks, training multilingual models, and releasing models with open access. Our first contribution is the release of the Bharat Parallel Corpus Collection (BPCC), the largest publicly available parallel corpora for Indic languages. BPCC contains a total of 230M bitext pairs, of which a total of 126M were newly added, including 644K manually translated sentence pairs created as part of this work. Our second contribution is the release of the first $n$-way parallel benchmark covering all 22 Indian languages, featuring diverse domains, Indian-origin content, and conversational test sets. Next, we present IndicTrans2, the first translation model to support all 22 languages, surpassing existing models in performance on multiple existing and new benchmarks created as a part of this work. Lastly, to promote accessibility and collaboration, we release our models and associated data with permissive licenses at https://github.com/AI4Bharat/IndicTrans2.

## 1 Introduction

India is a linguistically diverse region, with 1,369 distinct mother tongues identified in the census conducted in 2011. Of these, 22 languages have been listed in the 8[th] Schedule of the Constitution of India. Approximately 97% of the population of India speaks one of these 22 languages as their first language. English is widely spoken and serves as the default medium of formal communication in many areas, particularly in business, education, government, and judiciary.

---

* Equal Contribution. All author contributions listed in Section 10.
† Work done as a Master's student at Nilekani Centre at AI4Bharat, Indian Institute of Technology Madras.
‡ Work done while at Nilekani Centre at AI4Bharat and EkStep Foundation.
§ Work done while at Nilekani Centre at AI4Bharat, Indian Institute of Technology Madras and Microsoft.
¶ Corresponding Author: Mitesh Khapra (miteshk@cse.iitm.ac.in).

With such linguistic diversity, the importance in India of language translation for effective communication, social inclusion, equitable access, and national integrity cannot be over-emphasized. For example, for effective dissemination of information about government policies and welfare schemes, it is necessary to translate official documents and websites into regional languages. In the context of the judiciary, it is crucial to translate court proceedings and judgments into regional languages so that the petitioners, accused, and witnesses can understand and better participate in the judicial process. Similarly, in the context of education, translation can ensure that high-quality content becomes accessible to more learners in their regional languages. Lastly, translation also plays a vital role in national integration by ensuring that people migrating/traveling to and from different parts of the country can communicate better with people in their new locations.

The last decade has seen rapid progress in Neural Machine Translation, with the latest neural models (Johnson et al., 2017; Liu et al., 2020a; Fan et al., 2020; Kim et al., 2021; Lepikhin et al., 2021; Ramesh et al., 2022; Costa-jussà et al., 2022; Siddhant et al., 2022) supporting **hundreds** of languages and **thousands** of translation directions. However, these models either do not have a good coverage of Indian languages, or their performance on Indian languages is poor, or both. Further, none of these models are evaluated on a diverse set of domains or content of Indian origin, as there are no robust benchmarks designed explicitly for Indian languages. Another evidence of the neglect of Indian languages is that in the past 16 years since its inception, the shared tasks run under the Workshop on Machine Translation (WMT) have only covered a total of 4 Indian languages summed across all these years.[1] While the Workshop on Asian Translation (WAT) (Nakazawa et al., 2022) and the Workshop on Speech and Language Technologies for Dravidian Languages (Madasamy et al., 2022) have made significant contributions, they have not garnered the same level of popularity or academic participation as the WMT. As a result, despite the rapid progress in the broader field of Machine Translation, no single commercial or open-source translation model supports *all* the 22 languages listed in the Constitution.

In this paper, we pose the following question: *What are the missing pieces required for enabling wide and easy access to high-quality machine translation for all 22 scheduled Indian languages?* We believe there are four axes of improvement required: (a) curation and creation of significantly larger **training datasets**, (b) creation of high quality and diverse **benchmarks**, (c) training and evaluation of multilingual **models**, and (d) releasing of models with **open access**. For axis (a) training datasets, we need to create high-quality "seed data" comprising manually translated parallel sentences for all 22 languages with representation from diverse domains. It is to be noted that for several of the 22 languages, no publicly available translation data exists. This manually created data has to be supplemented with a higher volume of semi-automatically generated data by bitext mining from web-scale monolingual corpora and multilingual documents. For axis (b) benchmarks, we need expert-created highly accurate benchmarks for all 22 languages across variations such as formality of language, length of sentences, domain of text, and source originality. For axis (c) models, we need to train accurate multilingual models that exploit the similarity between Indian languages and particularly benefit low-resource languages. We also need to improve processes for the evaluation of models by choosing robust metrics that are shown to correlate with human evaluation for Indian languages. In addition, we need to evaluate models with other metrics, such as improvement in post-editing performance. Finally, for axis (d) open access, created models must have permissive licenses that can be commercially deployed. For instance, Meta's NLLB models, though released in the open, have a CC-BY-NC license precluding commercial usage. In this paper, we contribute across these four axes with many notable firsts that we highlight below.

**Training datasets.** We release the **largest publicly available parallel corpora for Indic languages, the Bharat Parallel Corpus Collection (BPCC)**. As summarized in Table 1, BPCC contains a total of ~230M bitext pairs, of which a total of ~126M were newly added as part of this work. BPCC includes the following:

- Seed training data containing human translations of English sentences to all 22 Indic languages spanning multiple domains. This has a total of 644K En-X translation pairs across all languages, including 7 languages for which no manually created parallel data existed before this work.

- Bitext pairs from existing collections such as Samanantar (Ramesh et al., 2022) and NLLB (Costa-jussà et al., 2022) which were further filtered using LaBSE (Feng et al., 2022) based cosine similarity thresholds.

---

[1]This is, of course, not a comment on the organizers of WMT but a reflection of the lack of academic interest in Indian languages due to the lack of sufficient training and evaluation data

- New bitext pairs mined from additional monolingual sources such as *archive.org* and IndicCorp v2 (Doddapaneni et al., 2023) which were not covered in the existing collections mentioned above.

- New bitext pairs mined from additional document-aligned parallel sources such as NPTEL, UGCResources, Prabhupada Vani, etc. which were not covered in the existing collections mentioned above.

- A very large set of ~800 million back-translated sentences from diverse sources such as IndicCorp v2 (Doddapaneni et al., 2023), monolingual side of NLLB data (Costa-jussà et al., 2022) and CC-Matrix (Schwenk et al., 2021b).

We visualize these types of data in BPCC in Figure 7, to highlight the language coverage and our contributions in relation to existing data. As can be seen, for many languages, BPCC makes the first available datasets, and for all languages, it makes a significant increase in the datasets available.

**Benchmarks.** We create **IN22, the first $n$-way parallel benchmark covering all 22 Indian languages** with the English side being source-original. For benchmarks to be of high quality, they must represent content from diverse domains. We visualize the diversity of our created benchmark in Figure 8. Our benchmark contains high-quality human translations for sentences taken from India-specific articles belonging to 13 different domains, *viz.*, Culture, Economy, Education, Entertainment, Geography, Government, Health, Industry, Legal, News, Religion, Sports, and Tourism (see left chart of Figure 8). We refer to this subset as **IN22-Gen**. Our benchmark has another subset **IN22-Conv**, that contains translations for sentences taken from everyday conversations in the Indian context from 16 different domains, which were manually created by in-house experts starting from carefully created conversation prompts (see right chart of Figure 8).

**Models.** We release **IndicTrans2 (IT2), the first translation model to support all the 22 scheduled Indian languages**, trained on the BPCC dataset. The progress made in the quality of translation in this work with existing open models is captured in Figure 1. The plot shows the chrF++ metric for English to different languages (which is usually the more challenging translation direction for low-resource languages). Each language is represented by circles, where the size of the circle represents the number of speakers in that language. As can be seen, with IndicTrans2, we made progress in translation quality across languages and now support moderate to high-quality translation for most speakers in India. Later in the paper, we also report COMET scores, comparisons with commercial models, and human evaluations of our translations. We find that IT2 is the first model for Indian languages, which performs at par not only with open-source models like NLLB (Costa-jussà et al., 2022) but also with commercial models from Google and Microsoft. We release IndicTrans2-M2M, the first model to support direct translations between all the 22 scheduled Indic languages, supporting 462 translation directions.

**Open Access.** We aim to promote wider access to accurate translation models for all Indian languages. Therefore, we will release IndicTrans2 and its derivatives (IndicTrans2-M2M, IndicTrans2-Dist) under an open-source license, along with all training data, source code, and tools to enable replication and further improvements by the research community. Additionally, we provide IndicTrans2-Dist, approximately 1/5 the size of IndicTrans2 (~211M) with comparable performance to reduce deployment costs. We hope our paper will serve as a starting point for future research on Indic machine translation.

Figure 2 provides a comprehensive overview of the entire workflow, which involved the development of requisite human infrastructure, building high-quality seed datasets and robust India-centric benchmarks, and culminates with the release of IndicTrans2, which is the first model to support all the 22 scheduled languages. Section 3 describes the process followed for the creation of high-quality benchmarks and seed training data, which entails the establishment of a human infrastructure, followed by a detailed account of the translation workflow and the quality control procedures implemented. Subsequently, Section 4 outlines our bitext mining pipeline, incorporating both manual and automated checks that employ toxicity and language filters. After the creation of the benchmarks and training data, the next task, as covered in Section 5 is the training of IndicTrans2 with ablation of model architecture, dataset selections, and training procedures. Furthermore, Section 6 describes the robust evaluation of IndicTrans2 across existing benchmarks such as FLORES and the benchmarks we create, across diverse metrics and against both open-source and commercial models.

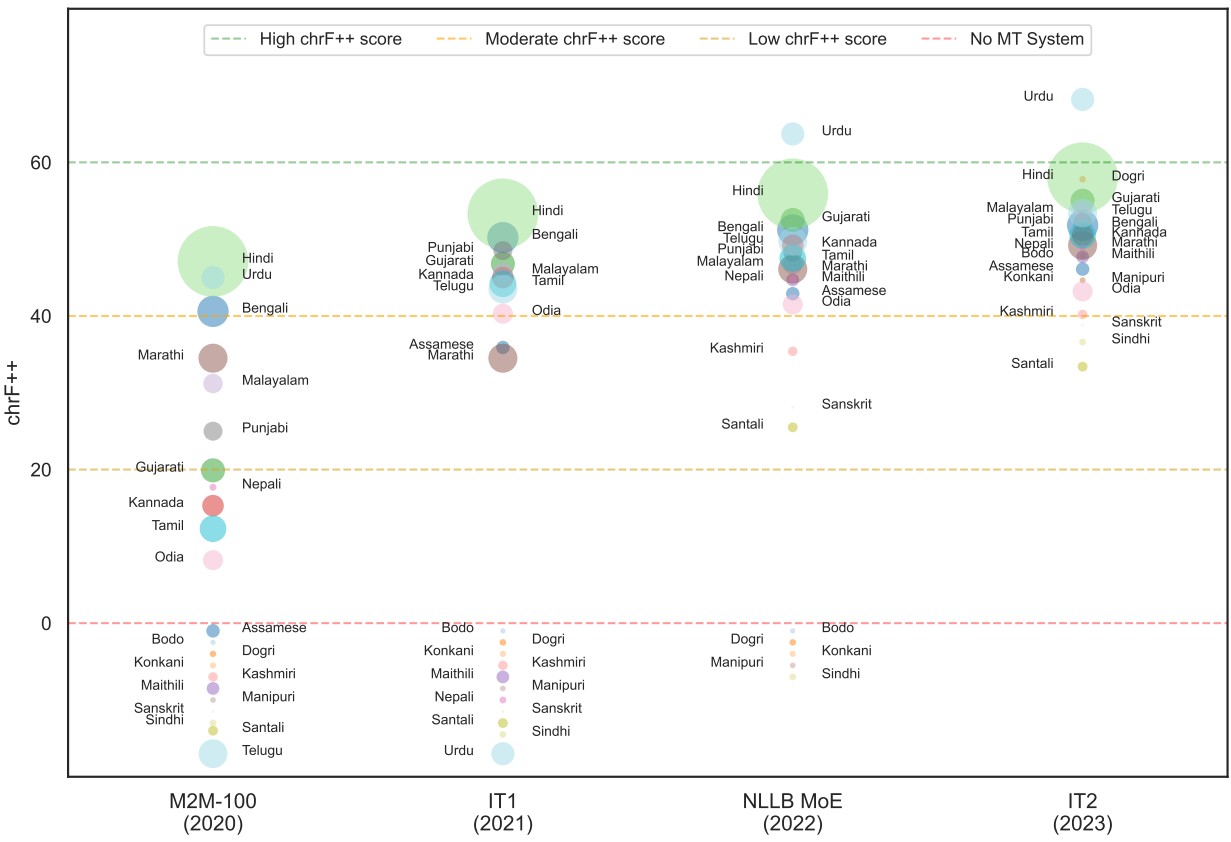

Figure 1: A visual representation of the advancements in machine translation systems for Indic languages using the IN22-Gen Evaluation set in the En-Indic direction. The depicted values have been subjected to minor adjustments to enhance readability; however, they accurately convey the overall trend. Thresholds are utilized to estimate performance boundaries for various systems across languages. The size of each language bubble is proportional to the speaker count for that language (see Table 55).

The paper concludes with a comprehensive summary and outlines potential future research directions. The Appendices provide supplementary results and additional details, including model and dataset cards.

## 2 Related Work

**Languages of India.** India, with a population of more than 1.4 billion, is a diverse country known for its rich linguistic heritage, and home to some of the world's most widely spoken languages. According to the Census of India 2011, 1369 mother tongues have been identified of which 121 languages have at least 10,000 speakers and 31 languages have at least a million speakers.[2] 22 of these languages have been listed in the $8^{th}$ Schedule of the Constitution of India[3], recognizing them as the scheduled languages of the Republic of India. According to the schedule, the Government of India is under an obligation to take measures to develop these languages such that they become an effective means of communication. **Nine of the Indic languages are amongst the most spoken languages across the globe**[4]**: Hindi ($4^{th}$), Bengali ($6^{th}$), Marathi ($13^{th}$), Telugu ($14^{th}$), Tamil ($17^{th}$), Urdu ($20^{th}$), Punjabi ($22^{nd}$), Gujarati ($24^{th}$) and Bhojpuri ($26^{th}$).** Some of these languages are also widely spoken and/or are official languages in neighboring countries *viz.*, Bangladesh, Nepal, and Pakistan. Indian languages are also fast-growing across the globe, particularly in North America, the United Kingdom, Australia, and the Middle East. Beyond the Indic languages, English is also

---

[2]https://en.wikipedia.org/wiki/Languages_of_India
[3]https://rajbhasha.gov.in/en/languages-included-eighth-schedule-indian-constitution
[4]https://en.wikipedia.org/wiki/List_of_languages_by_number_of_native_speakers

Table 1: Overall statistics for data collated from different sources (in thousands) for Indian languages and resources in this work. In this document, each language is identified with a BCP 47 tag sequence comprised of ISO 639-3 language subtag and ISO 15924 script subtag.

| | | Existing | | | | | BPCC (Newly Added) | | | |
| | | Mined | | Human | | | Mined | | Human | |
| *Name* | *Language* | Samanantar | NLLB | NLLB | ILCI | MASSIVE | Monolingual | Comparable | Wiki | Daily |
|---|---|---|---|---|---|---|---|---|---|---|
| *Assamese* | *asm_Beng* | 58.8 | 506.3 | - | 82.1 | - | 712.5 | 37.8 | 44.7 | 11.3 |
| *Bengali* | *ben_Beng* | 2,946.3 | 13,580.5 | - | 123.8 | 16.5 | 16,055.1 | 258.2 | 48.0 | 8.5 |
| *Bodo* | *brx_Deva* | - | - | - | 83.2 | - | - | <1 | 22.7 | 10.3 |
| *Dogri* | *doi_Deva* | - | - | - | - | - | - | - | 18.7 | 5.5 |
| *Konkani* | *gom_Deva* | - | - | - | 74.5 | - | - | - | 18.3 | 4.8 |
| *Gujarati* | *guj_Gujr* | 1,379.2 | 7,090.3 | - | 107.4 | - | 11,630.3 | 573.0 | 25.0 | 3.2 |
| *Hindi* | *hin_Deva* | 4,416.7 | 6,646.7 | - | 165.6 | 16.5 | 27,187.8 | 853.3 | 40.3 | 8.4 |
| *Kannada* | *kan_Knda* | 1,692.2 | 8,871.1 | - | 76.4 | 16.5 | 12,501.0 | 380.2 | 32.2 | 8.5 |
| *Kashmiri* | *kas_Arab* | - | 124.9 | 6.2 | - | - | - | - | 15.5 | 4.3 |
| | *kas_Deva* | - | 194.0 | 6.2 | - | - | - | - | - | - |
| *Maithili* | *mai_Deva* | - | 62.2 | - | - | - | - | <1 | 24.4 | 4.2 |
| *Malayalam* | *mal_Mlym* | 2,029.2 | 8,818.2 | - | 87.9 | 16.5 | 12,378.6 | 356.4 | 41.6 | 8.4 |
| *Marathi* | *mar_Deva* | 1,366.1 | 6,393.2 | - | 117.0 | - | 10,806.0 | 432.4 | 54.3 | 4.6 |
| *Manipuri* | *mni_Beng* | - | 346.9 | 6.2 | 13.1 | - | - | 20.1 | - | <1 |
| | *mni_Mtei* | - | - | - | 16.0 | - | - | - | 19.9 | 6.8 |
| *Nepali* | *npi_Deva* | - | 1,583.5 | - | 28.6 | - | 10.5 | 6.2 | 45.9 | 10.9 |
| *Odia* | *ory_Orya* | 514.9 | 2,382.6 | - | - | - | 2,863.1 | 121.5 | 33.7 | 3.2 |
| *Punjabi* | *pan_Guru* | 1,418.3 | 1,978.3 | - | 71.5 | - | 6,275.8 | 207.2 | 6.3 | 3.2 |
| *Sanskrit* | *san_Deva* | - | 244.1 | - | - | - | - | <1 | 27.7 | 5.4 |
| *Santali* | *sat_Olck* | - | - | - | - | - | - | - | 22.5 | 1.8 |
| *Sindhi* | *snd_Arab* | - | 2,128.4 | - | - | - | - | - | - | - |
| | *snd_Deva* | - | - | - | - | - | - | - | 10.5 | - |
| *Tamil* | *tam_Taml* | 1,833.2 | 8,665.2 | - | 120.7 | 16.5 | 9,690.3 | 452.8 | 21.0 | 8.6 |
| *Telugu* | *tel_Telu* | 1,780.5 | 10,062.8 | - | 73.6 | 16.5 | 11,100.0 | 437.2 | 29.7 | 8.5 |
| *Urdu* | *urd_Arab* | - | 5,321.0 | - | 101.0 | 16.5 | 484.9 | 225.3 | 41.3 | 8.4 |
| **# Total** | | 19,435.4 | 84,998.3 | 18.6 | 1,342.6 | 115.4 | 121,695.8 | 4,353.1 | 644.3 | 139.7 |

widely spoken by in India, with a speaker base of 246 million.[5] However, even with a large speaker base, many of these languages still lack an online presence and high-quality NLP technologies. Of the 22 scheduled languages, only 4 of them are so-called "Winners" according to the classification by Joshi et al. (2020). It is thus essential to support translation technologies (and NLP technologies in general) for such a large population base to bring the benefits of digital technologies to a large audience. What distinguishes the Indian subcontinent is not only the large speaker base of many languages but also the linguistic diversity of its languages. **Languages from four major language families (Indo-Aryan branch of the Indo-European family, Dravidian, Tibeto-Burman, and Austro-Asiatic) are spoken in the subcontinent. According to Wikipedia,[6] India has amongst the highest linguistic diversity at around 0.914 to 0.93, depending on the measure.** Indic languages are written in a variety of scripts, the majority of which are derived from the *Brahmi* script. **Up to 12 major scripts spanning abugida, alphabetic, and abjad script types are used** (Daniels & Bright, 1996). Underlying this diversity in languages and scripts is also a great deal of similarity at various linguistic levels, owing to language relatedness and contact over a long period (Emeneau, 1956; Subbarao, 2012; Kunchukuttan & Bhattacharyya, 2020). **The diversity of languages and their interactions provide for challenging problems and opportunities in machine translation for Indic languages**.

**Datasets.** We summarize some of the prominent parallel corpora created for Indian languages. The Indian Languages Corpora Initiative (ILCI) (Choudhary & Jha, 2011) created n-way parallel annotated corpora containing 50K

---

[5] https://en.wikipedia.org/wiki/Indian_English
[6] https://en.wikipedia.org/wiki/Linguistic_diversity_index

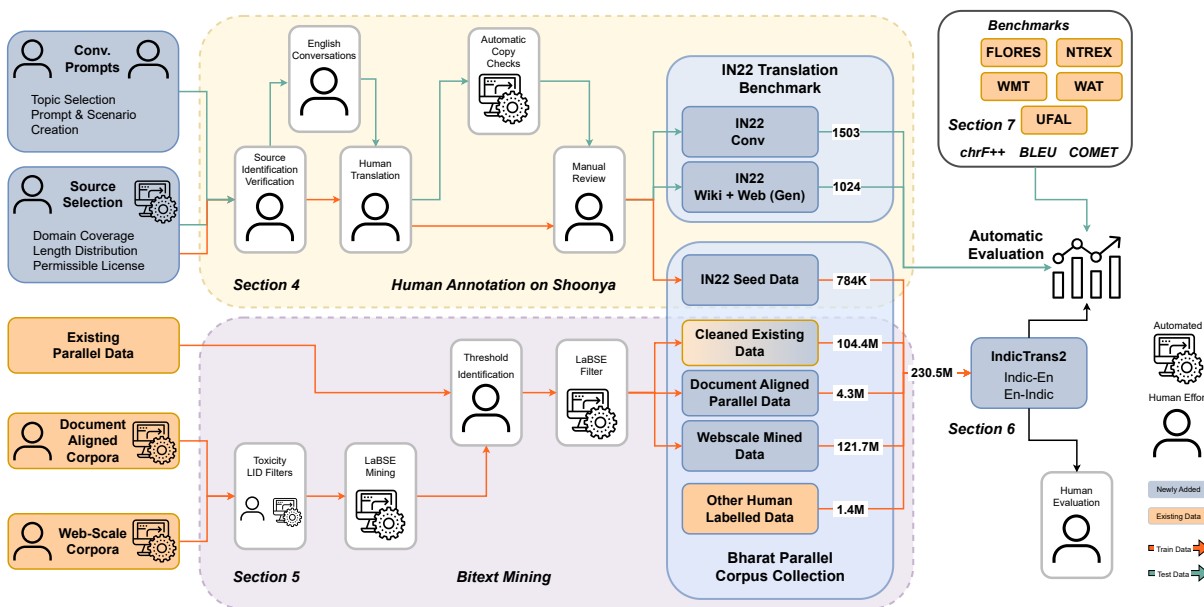

Figure 2: Overview of the workflow used for building Bharat Parallel Corpus Collection, IN22 and IndicTrans2.

sentences per language for 12 major Indian languages, covering Health and Tourism domains. However, with the advent of neural MT models, it has been established that these models need large-scale parallel corpora for superior performance (Edunov et al., 2018; Aharoni et al., 2019). Some early attempts include the IIT-Bombay English-Hindi corpus (Kunchukuttan et al., 2018) and the PMIndia corpus (Haddow & Kirefu, 2020), which aligned sentences from the Prime Minister's speeches in English and 12 Indic languages. The CVIT-PIB corpus (Philip et al., 2021) aligned parallel documents from the Press Information Bureau archives, resulting in English to 11 Indian language pairs. WAT 2021 shared task compiled existing sources to create 9 million sentence pairs between English and Indic languages. Creating parallel corpora for all Indic languages is challenging due to the lack of identifiable parallel documents and the effort required for human annotation at scale. Consequently, attention has turned towards mining parallel corpora from non-comparable sources, leveraging the multilingual nature of India's information availability, though identifying parallel pages based on URL patterns remains challenging (Resnik & Smith, 2003). Following prior works on mining data from web-scale data (Schwenk et al., 2021b), Samanantar (Ramesh et al., 2022) was mined from IndicCorp v1 (Kakwani et al., 2020) using LaBSE (Feng et al., 2022) based sentence embeddings, resulting in a 3-fold increase in data compared to existing parallel data. Combined with existing data, Samanantar contained 49.7 million sentence pairs between English and 11 Indic languages. In subsequent work, NLLB project (Costa-jussà et al., 2022) mined parallel data from CommonCrawl dumps (Wenzek et al., 2020) using LASER (Heffernan et al., 2022) based sentence embeddings. This corpus resulted in 448 million sentence English-centric pairs covering 19 Indic languages. While NLLB (Costa-jussà et al., 2022) had the largest coverage so far, all these efforts still do not cover all the 22 scheduled languages of India. This necessitates the need to create "seed" data (refer to §3) for the low-resource languages to help boost the performance of MT systems for these languages.

**Benchmarks and Shared Tasks.** Benchmarks have improved NLP systems across various tasks (Rajpurkar et al., 2016; Wang et al., 2018; 2019; Hu et al., 2020; Doddapaneni et al., 2023). Over the years, an increasing focus has been on improving MT systems for Indic languages, with sustained endeavors to develop appropriate benchmarks. The introduction of the Hindi-English MT challenge in WMT'14 marked one of the earliest attempts to establish benchmarks for Indic languages (Bojar et al., 2014). Subsequently, WMT extended its efforts by incorporating the Gujarati-English and Tamil-English language pairs in 2019 (Barrault et al., 2019) and 2020 (Barrault et al., 2020), respectively. WAT (Workshop on Asian Translation) has continuously supported IndicMT with the inclusion of the IITB Hindi-English dataset (Kunchukuttan et al., 2018) in the WAT 2016. Subsequently, WAT expanded its efforts, adding 6, 8, 10, and

15 languages in 2018, 2020, 2021, and 2022, respectively (Nakazawa et al., 2018; 2020; 2021a; 2022). Siripragada et al. (2020) introduced a benchmark consisting of roughly 2K-3K sentences from *Mann ki Baat*[7], covering 9 Indic languages translated to English. FLORES 101 (Goyal et al., 2022) was one of the first attempts to create a large-scale MT benchmark with n-way parallel *devtest* and held-out *test* sets of around 1000 sentences for 101 languages, including support for 14 Indic languages manually annotated from the Wikimedia content. This was followed up by NLLB (Costa-jussà et al., 2022), extending the total language coverage to 200, which includes 19 Indic languages listed in the Constitution (plus a few more Indic languages). NTREX (Federmann et al., 2022) expanded coverage of languages of test data from WMT 2019 (Barrault et al., 2019) to 128 languages and covers 16 Indic languages. The test set contains 1997 manually translated sentences, primarily sourced from the news domain.

**Neural MT models.** The introduction of Neural MT and the creation of large-scale parallel corpora led to significant advancements in the field of Indic MT. Broadly, they follow the *Embed - Encode - Attend - Decode* approach. Initial approaches used Recurrent Neural Networks (Bahdanau et al., 2015) and later transformer-based approaches (Vaswani et al., 2017) became more prominent. The introduction of attention and subword-based modeling addressed the issues of word ordering and data sparsity. The models were able to generate grammatically fluent and accurate outputs. Some noteworthy Neural MT models studying Indian languages include (Philip et al., 2021; Ramesh et al., 2022; Fan et al., 2020; Costa-jussà et al., 2022). These were followed up with multilingual and pre-trained MT models (Kudugunta et al., 2019; Liu et al., 2020b; Xue et al., 2021; Dabre et al., 2022). These models were able to transfer knowledge from high-resource to low-resource languages by leveraging large amounts of training data and language similarities across languages, making it possible to train a good-quality MT system for low-resource languages (Dabre et al., 2021). Over the last few years, large corpora (Ramesh et al., 2022; Costa-jussà et al., 2022) and larger models (Fan et al., 2020; Costa-jussà et al., 2022) marked significant improvements in the translation quality. Recent work has also explored translation for extremely low-resource languages with hardly any parallel corpora and limited monolingual corpora (Costa-jussà et al., 2022; Bapna et al., 2022; Maurya et al., 2023).

# 3 Creating High-quality Translation Datasets at Scale

In this section, we describe the translation process, and the Shoonya[8] infrastructure to ensure a high-quality translation workflow. We also describe in detail the translation workflow followed and quality control procedures and the salient features of the resultant datasets created: (a) BPCC-Human, the training dataset from English to 22 Indic languages, and (b) IN22, the test set for translation evaluation between English and Indian languages.

## 3.1 Translation Workflow

The overall translation workflow is described below and illustrated in Figure 3. The translation workflow comprises four stages. First, sentences for translation are chosen based on criteria such as domain coverage, length, and licensing. These sentences are sourced from diverse domains, including News, Business, and Health. Next, the selected sentences undergo a verification process where annotators ensure their quality and correctness, tagging them accordingly. The entire paragraph is rejected in case of any inaccurate sentences to prevent ambiguity. Once the verification is complete, the sentences are translated into 22 Indic languages, adhering to rigorous guidelines. Lastly, the translated content is reviewed by experienced translators who check for adherence to guidelines and overall quality, suggesting improvements or corrections as needed. If a translation is rejected, it is sent back to the original translator for revision, ensuring the highest translation standards. Specific customizations to the workflow depending on the kind of dataset being created (training/test) are discussed in subsequent sections.

All the stages in the workflow are performed on *Shoonya*,[8] an open-source[9] platform which was developed as a part of this work for supporting language annotation tasks customized for Indian languages. Additional information about the translation stages, including translation guidelines and the interface utilized for generating human-annotated translation data along with its key features, can be found in Appendix F.

---

[7] https://www.pmindia.gov.in/en/mann-ki-baat/
[8] https://ai4bharat.org/shoonya
[9] https://github.com/AI4Bharat/Shoonya

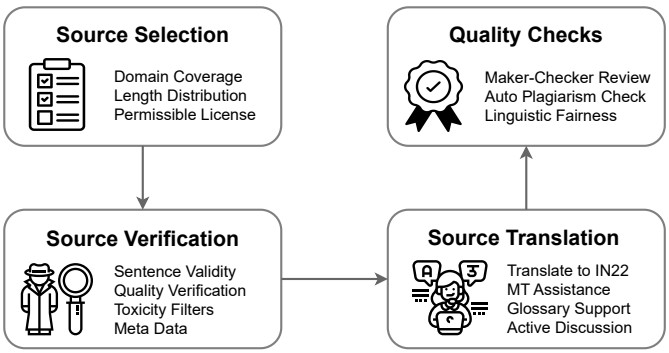

Figure 3: Translation workflow in Shoonya

## 3.2 Building the IN22 Test set

In this section, we describe the IN22 test set, which is a new manually created n-way parallel test set covering English and 22 Indic languages. We motivate the need for such a benchmark, describe its features in detail, and explain the construction of the test set.

While there are a few test sets for Indian languages, there is still a need for a comprehensive test set that satisfies the following needs of Indian language machine translation and addresses the limitations of existing test sets:

- We need a test set that covers all 22 Indic languages and enables evaluation between all possible pairs of these scheduled languages. FLORES-200 (Costa-jussà et al., 2022) has the largest coverage amongst existing test sets (n-way, 19 languages). The other test sets WAT 2020 (Nakazawa et al., 2020), WAT 2021 (Nakazawa et al., 2021a), WMT 2014 (Bojar et al., 2014), WMT 2019 (Barrault et al., 2019), WMT 2020 (Barrault et al., 2020), UFAL (Ramasamy et al., 2012) and NTREX (Federmann et al., 2022) have limited coverage, with the majority having only a few of the top-10 languages represented at the most.

- The test set should be diverse in terms of domains covered and represent a realistic distribution of sentence lengths while also encompassing topics relevant to India, which would be the primary use case for models supporting Indic languages. Existing test sets like WMT and FLORES are more general-purpose and have limited representation for Indian topics like named entities, locale, culture-specific terms, etc.

Table 2 compares existing benchmarks based on test set size, language coverage, domain coverage, and the language in which the dataset is source original.

### 3.2.1 Corpus Description

We describe the details and salient points of the IN22 test set. This test set comprises three subsets, which serve distinct evaluation scenarios:

- **Multi-Domain Wikipedia subset (512 sentences)**: This subset is designed to be multi-domain, expanding to at least five more domains than the existing benchmarks like FLORES-200 (Costa-jussà et al., 2022). Domain coverage is presented in Table 52.

- **Multi-Domain Web Sources subset (512 sentences)**: This subset was designed to represent content from sources other than Wikipedia to have more diversity in content and writing style and with more focus on India-centric content. These were mainly sourced from PDFs and from sources that are not accessible or crawlable on the web, thereby reducing the possibility of these sentences already being part of any mined data.

Table 2: Comparison of Various Benchmarks based on Test Set Size, Language Coverage, Domain Coverage, and Source Original.

| Dataset | Test Set Size | Language Coverage | Domain Coverage | Source Original |
|---|---|---|---|---|
| FLORES-200 (devtest) | 1012 | 19 | 8 | eng |
| NTREX | 1997 | 12 | news(1) | eng |
| WMT 2014 (hin) | 2507 | 1 | news(1) | both |
| WMT 2019 (guj) | ≈ 1000 | 1 | 1 | both |
| WMT 2020 (tam) | ≈ 1000 | 1 | 1 | both |
| WAT 2020 | ≈ 3500 | 7 | 1 | eng |
| WAT 2021 | ≈ 2390 | 10 | 1 | eng |
| UFAL | 2000 | 1 | 3 | eng |
| IN22-Wiki | 512 | 22 | 13 | eng |
| IN22-Web | 512 | 22 | 13 | eng |
| IN22-Conv | 1503 | 22 | 16 | eng |

- **Conversation Translation Benchmark (1503 sentences)**: This subset was designed to evaluate the performance of models in day-to-day conversations in applications like chat. The translations are drawn from a multi-turn English dialog dataset we built, enabling evaluation across all the axes, including sentence level, turn level, and document level (complete conversation).

The following are some key features of the benchmark:

- It is an $n$-way parallel test set containing 2527 original English sentences translated into 22 Indic languages with high-quality translations done by in-house translators from scratch without recourse to any existing MT system. Metadata, consisting of domains and context sentences (in raw, unedited format) for source sentences, is provided in the test set to enable a fine-grained analysis of translation quality for each example.

- IN22 enables evaluation in 500+ directions, including (i) source original translation from English to other languages. (ii) Indic to English translation evaluation and the ability to study relative language performance since the underlying sentence is the same, (iii) comparison of 462 inter-Indic translation directions.

- The test set is diverse in terms of the domains covered and the distribution of sentence lengths. The Web sources and Wikipedia subsets cover 13 domains, while the conversational subset covers 16 domains. The length distribution is chosen to reflect a realistic distribution while also having a sufficient number of long sentences, which can present a challenge to MT models. Figure 10 provide an overview of the domain v/s length distributions of our benchmarks, while Table 52 provides an overview of the domain diversity.

- Table 3 provides some statistics about the test set. Wikipedia and Web Sources have longer sentences than the conversational dataset. Conversational sentences have a higher perplexity compared to the other subsets, perhaps hinting at the lower representation of such scenarios in the GPT2 training corpus.

### 3.2.2 Source Selection

We describe the selection of the source sentences for each of the three subsets: Wikipedia, Web Sources, and Conversation. The creation of the Wikipedia subset involved selecting English source sentences from various Wikipedia categories to ensure broad coverage across different domains. Sentences were filtered based on length (less than 6 words or more than 80 words were discarded) and overlap with the FLORES-200 test set (4-gram overlap). For each sentence, a context window of 3 sentences (typically one before and one after) was constructed. The Web Sources subset focused on Indian topics and used Government of India websites and digital libraries as sources, with sentences selected using a similar procedure. The Conversation subset involved creating English conversations with predefined prompts and scenarios, which were then translated into 22 Indic languages. Overall, these subsets were created with careful consideration for domain diversity and language coverage. Appendix E.1 provides detailed information about the procedure followed for the selection of sentences for all the three subsets of IN22.

Table 3: Statistics for the three subsets in the IN22 benchmark.

|  | Subsets | | |
|---|---|---|---|
|  | Wikipedia | Web Sources | Conversational |
| Number of sentences | 512 | 512 | 1503 |
| Average sentence length (number of English characters) | 169.27 | 144.53 | 54.18 |
| Average sentence length (number of English words) | 26.30 | 23.20 | 9.88 |
| Number of context sentences available | 3 | 3 | conversation |
| Number of domains | 13 | 13 | 16 |
| Average perplexity of English (computed using GPT-2) | 63.67 | 67.22 | 72.33 |

Table 53 contains the statistics of the conversation subset of IN22 test set. The subset contains conversations sampled from 16 domains including 'arts', 'history', 'school life', etc. The domains cover a diverse set of topics such as 'Government schemes', 'Movies', 'Historical Architectures', etc. Table 54 contains an English example from the conversation subset of IN22 test set. The conversation subset of IN22 benchmark can also be repurposed as a document translation task and would be useful in the context of evaluating LLMs.

### 3.2.3 Quality Control Procedure.

In the process of test set creation, it is imperative to implement strict quality control guidelines to prevent the use of MT outputs as a starting point by translators and ensure the fairness and reliability of the resulting benchmarks. As a first step, we disable MT outputs in Shoonya for this translation task. To further ensure translators are not taking recourse to MT outputs, we follow a systematic approach that involves conducting pairwise comparisons between human translations and the outputs of widely accessible machine translation (MT) systems, such as Google, Azure, NLLB (Costa-jussà et al., 2022), and IndicTrans1 (Ramesh et al., 2022). The BLEU score (Papineni et al., 2002) serves as an effective metric for detecting exact matches between translations and MT system outputs. Initially, we generate predictions from multiple MT systems for a batch of sentences translated by an annotator. Subsequently, we compute BLEU scores, denoted as $B(S_i, T)$, with respect to the reference translations $(T)$ and each MT system output $(S_i)$. A series of conditions are assessed based on the number of MT systems supporting a particular language (denoted as $k$). For languages supported by multiple MT systems, the system with the highest BLEU score $(S_j)$ is selected, where $j = argmax_i\ B(S_i, T)$.

$$|B(S_i, T) - B(S_j, T)| \leq \delta \qquad \forall i, j \in \{1, \ldots, k\} \tag{1}$$

If the pairwise BLEU score difference between any two systems falls within an acceptable threshold (see Equation (1)), then the translations are accepted. In this work, we set the $\delta$ to be 10. Otherwise, a high difference in BLEU scores indicates that the high-scoring model might have been a source for translation. In cases of high overlap with any of the machine translation systems, a new annotator is assigned to the task, and the quality control procedure is repeated, ensuring the creation of reliable and accurate benchmarks.

### 3.3 Building the BPCC Training Set

We create BPCC-Human (BPCC-H), a manually translated, multi-domain $n$-way seed parallel corpus between English and 22 Indic languages.[10] In this section, we motivate the need for high-quality, human-translated training data, provide an overview of the dataset, and describe the process of construction of the dataset.

**Motivation for creating the seed dataset.** The primary method to create parallel corpora at scale for many languages is to mine data from publicly available sources. While this approach has shown success for languages that have good

---

[10]Currently, the seed corpora being released are not n-way parallel since different language teams are independently translating different batches of the English source sentences. This is ongoing work.

representation in monolingual corpus and multilingual models (Ramesh et al., 2022; Philip et al., 2021; Kunchukuttan et al., 2018), the same cannot extend to very-low resource languages. This makes it important to invest in building high-quality, modest-sized parallel corpora. We take inspiration from previous efforts to manually create large multilingual seed corpora explicitly for building machine translation models like ILCI (Jha, 2010), ALT (Riza et al., 2016), and NLLB-Seed (Costa-jussà et al., 2022; Maillard et al., 2023). These previous efforts have been instrumental in significantly boosting MT efforts for low-resource languages; particularly, seed data also helps in bootstrapping the development of various NLP tools such as language identifiers, topic classifiers, named entity recognition, etc., where minimal monolingual sources exist.

### 3.3.1 Corpus Description

Following are some key aspects of the BPCC-H dataset:

- BPCC-H-Wiki is the largest publicly available manually translated multi-domain parallel corpora in terms of language coverage. It contains a total of 644.3K sentence pairs, ranging from 6.3K to 54.3K pairs depending on the language, averaging around 26K sentence pairs per language pair. These translations were performed by qualified professional translators following a high-quality translation process and a systematic review of the sentence pairs, unlike crowdsourcing efforts. Per-language sentence counts can be seen in Table 1.

- BPCC-H-Wiki provides good seed parallel corpora for 4 extremely low-resource languages without public corpora, *viz.* Bodo, Dogri, Santali, and Goan Konkani. More than 10K sentence pairs are available for each of these languages. There are hardly any sources or models to mine parallel corpora for these languages.

- There are multiple scripts available for a few languages. However, for our current seed data creation efforts, we restrict ourselves to only one script per language, choosing the most widely used script for administrative purposes.

- A subset of BPCC-H, BPCC-H-Daily comprises spoken text particularly covering various types of sentences commonly used in different day-to-day scenarios, such as queries, commands, and feedback, across a range of applications including digital payment apps, grocery/food delivery apps, and government services apps. Our goal was to encompass diverse named entities in relevant domains, covering various expressions from these services. This subset, comprising 139.7K bitext pairs in 21 Indic languages except Sindhi, was developed from English sentences to expand the diversity of the parallel corpus.

### 3.3.2 Translation Details

The translation process has already been described above. Here, we discuss aspects of the translation process specific to BPCC-H.

First, we choose to translate from English source sentences to Indic languages in order to simplify the source sentence selection (easier availability of copyright-free English sentences for translation, diversity in domains, *etc.*). The Indian language side, therefore would exhibit translationese effects (Zhang & Toral, 2019). However, this is not uncommon, and many parallel corpora are English original (Costa-jussà et al., 2022; Maillard et al., 2023; FitzGerald et al., 2022).

The English source sentences were selected from Wikipedia. We identified various Wikipedia categories of interest and then identified article pages within those categories. This was done to ensure broad coverage of domains. We identified a block of three sentences following Goyal et al. (2022), of which one was to be translated, and the others would be context sentences to resolve any ambiguities during translation. The translators had the option of post-editing MT outputs from an existing model wherever feasible.

## 4 Mining Training Data at Scale

The quality of MT systems depends on access to good quality parallel data, and increasing parallel corpora improves translation quality (Khayrallah & Koehn, 2018). However, obtaining high-quality parallel corpora in large quantities is a challenging task. While human annotation is one way to source data, it is not scalable beyond a certain point to meet

the demands of data-hungry models. Thus, there is a growing need to (semi-)automatically mine large-scale training corpora to address this issue.

Over the years, various approaches have been proposed for generating parallel data for machine translation (MT) training. One set of approaches focused on mining parallel corpora from aligned documents identified from web-corpora (Resnik & Smith, 2003; Bañón et al., 2020; El-Kishky et al., 2020) or from specific document collections like EuroParl (Koehn, 2005) and the United Nations (Ziemski et al., 2016). Document alignment is a non-trivial problem for open web-corpora and relies on URL matching or translation-based matching in constrained settings. Specific document collections may be limited in domain coverage and are often scarce. Instead of limiting mining to comparable documents, recent methods have explored the mining of sentence pairs from large sentence collections using multilingual embeddings without regard for document alignment. This has allowed the mining of parallel data from arbitrary and diverse collections of data (Schwenk et al., 2021a;b; Costa-jussà et al., 2022). Similar approaches have been extended to Indic languages (Ramesh et al., 2022), establishing the utility of large-scale mining for building multilingual NMT models.

Major Indic languages have a reasonable online presence, with numerous websites publishing data in multiple Indic languages, primarily pivoting through English or Hindi. Moreover, being a multilingual nation, several government documents, books, judgments, legal proceedings, etc., are published in multiple Indic languages, which are directly comparable and are thereby aligned at a document level. Hence, we invest efforts in mining parallel corpora by leveraging large-scale monolingual data as well as document-aligned data from comparable sources.

Our mining efforts focus on 12 Indic languages: Assamese, Bengali, Gujarati, Hindi, Kannada, Malayalam, Marathi, Odia, Punjabi, Tamil, Telugu, and Urdu. These languages have a good representation in monolingual corpora, as reported in Doddapaneni et al. (2023). However, the low-resource languages have comparatively lesser monolingual data, and the quality of sentence embeddings is unknown. Therefore, we rely on high-quality human-translated data, as described in Section 3, for training low-resource languages. Nepali was also considered in an initial round of mining, and some bitext data was mined. However, it was dropped from mining subsequently since LaBSE embeddings (Feng et al., 2022) were observed to be suboptimal for Nepali. Going forward, we only focus on mining parallel corpora for the 12 languages mentioned above.

Table 1 provides statistics of the mined parallel corpora. The following is a summary of the mined corpora:

- In our mining efforts, a total of ~126 million sentence pairs were mined in addition to existing corpora, resulting in an aggregated collection of ~230.5 million sentence pairs after deduplication, which is ~5× increase in parallel corpora size as compared to Ramesh et al. (2022).

- Mining from the monolingual corpus resulted in the largest parallel corpus gains, with 121 million sentence pairs across 13 Indic languages.

- Mining from comparable corpora results in a diverse parallel corpus covering a wide range of topics like Religion, Education, Legal, etc. In total 4.35 million sentence pairs were mined across 17 Indic languages.

- Filtering existing corpora turned out to be an important exercise, as we observed around 75% of the data was discarded due to poor quality of alignment. In summary, Costa-jussà et al. (2022) was filtered and thereby reduced from 448.1 million to ~85 million sentence pairs, and Ramesh et al. (2022) reduced from 49.7 million to 19.4 million sentence pairs. We describe the filtering process below.

## 4.1 Mining from Monolingual Corpora

The primary idea behind mining parallel sentence pairs from large corpora is to represent sentences from all languages in a common embedding space using LaBSE (Feng et al., 2022), such that the distance between a pair of sentences reflects their semantic difference. To achieve this, we project all the sentences into a shared space and search for the nearest neighbors around a query sentence. Given a source sentence $S$ in language $L$, we look for the closest Approximate Nearest Neighbors (ANNs) to $S_L$ within a selected threshold. The main challenge lies in scaling this process efficiently

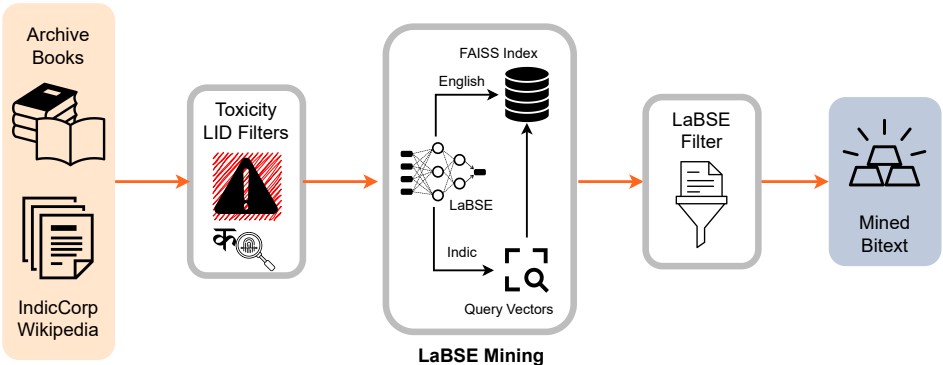

Figure 4: Mining workflow for Monolingual corpora

Table 4: The total number of monolingual sentences and extracted parallel sentences count (in millions). The size of the English monolingual corpus is 429 Million. † indicates the mining for Nepali was performed on an intermediate version of IndicCorp v2 (Doddapaneni et al., 2023).

| Language | Monolingual Corpus | Extracted Pairs |
|---|---|---|
| asm_Beng | 3.3M | 0.7M |
| ben_Beng | 269.5M | 16.0M |
| guj_Gujr | 115.5M | 11.6M |
| hin_Deva | 473.2M | 27.1M |
| kan_Knda | 101.7M | 12.5M |
| mal_Mlym | 91.8M | 12.3M |
| mar_Deva | 64.7M | 10.8M |
| npi_Deva† | - | 0.01M |
| ory_Orya | 13.4M | 2.8M |
| pan_Guru | 38.6M | 6.2M |
| tam_Taml | 64.7M | 9.6M |
| tel_Telu | 108.5M | 11.1M |
| urd_Arab | 76.2M | 0.4M |
| **# Total** | 2113M | 121M |

to project millions of sentences and compute nearest neighbors over a large search space in a scalable and efficient manner. Previous work, such as CCMatrix (Schwenk et al., 2021b), has demonstrated that ANN search can be efficiently performed at scale using quantization, efficient indexing, and retrieval. Similar approaches have been used in prior work on Indic languages, such as Samanantar (Ramesh et al., 2022). Our work follows the same approach as Samanantar for mining parallel sentences from large-scale monolingual corpora. We differ from Samanantar (Ramesh et al., 2022) primarily in the amount of monolingual data used for mining. We use a larger collection of monolingual corpora for our work, comprising IndicCorp v2 (Doddapaneni et al., 2023), Wikipedia[11] and data from Internet Archive.[12] Specifically, we have used 2.1 billion monolingual Indic sentences, significantly higher than Samanantar (Ramesh et al., 2022) (398.5 million). Moreover, the number of English sentences that we used for our bitext mining has increased from 54.3 million to 429 million. Additionally, we have also mined bitext for Urdu and Nepali.

Figure 4 shows an overview of the mining process. We provide details of the mining workflow below. The mining from monolingual sources resulted in 121 million bitext pairs. Table 4 shows the per-language statistics of the mined corpora.

---

[11] https://dumps.wikimedia.org/
[12] https://archive.org

Table 5: Pearson ($\rho$) and Kendal ($\tau$) correlation Cosine Similarity of LaBSE and LASER model with Human Ratings on the STS data released by Ramesh et al. (2022).

| | | LaBSE | | LASER | |
|---|---|---|---|---|---|
| Language | Sample Size | $\rho$ | $\tau$ | $\rho$ | $\tau$ |
| asm_Beng | 1,971 | 0.3942 | 0.2989 | 0.3797 | 0.3021 |
| ben_Beng | 3,797 | 0.5149 | 0.4392 | 0.3137 | 0.2522 |
| guj_Gujr | 2,298 | 0.5437 | 0.4475 | 0.2945 | 0.3429 |
| hin_Deva | 4,616 | 0.5575 | 0.4691 | 0.4550 | 0.4005 |
| kan_Knda | 2,838 | 0.5211 | 0.4184 | 0.2640 | 0.2634 |
| mal_Mlym | 2,760 | 0.5331 | 0.4354 | 0.4368 | 0.3339 |
| mar_Deva | 1,984 | 0.4773 | 0.3916 | 0.3540 | 0.2660 |
| ory_Orya | 1,264 | 0.1148 | 0.1152 | 0.0361 | 0.0332 |
| pan_Guru | 2,222 | 0.5952 | 0.4725 | 0.3812 | 0.3435 |
| tam_Taml | 2,882 | 0.5099 | 0.4084 | 0.2296 | 0.2367 |
| tel_Telu | 2,516 | 0.4426 | 0.3780 | 0.2164 | 0.1936 |
| Average | - | 0.4731 | 0.3886 | 0.3055 | 0.2698 |

**Data Curation.** Our data curation process commenced with the collection of documents from diverse sources, including IndicCorp v2 (Doddapaneni et al., 2023), Wikipedia[11] and Internet Archive data[12] which were aggregated at the document level. However, as our objective was to mine sentence-level parallel data, we used the Indic NLP library (Kunchukuttan, 2020) to segment these documents into individual sentences. Subsequently, we implemented a strict quality control procedure, where we perform language identification (LID) at the sentence level using LID filters from Costa-jussà et al. (2022). As previous studies have shown, web-scale data often contains offensive content (Kreutzer et al., 2022), therefore we use an "offensive word list" to filter out such content. This list is augmented with data from Toxicity-200 (Costa-jussà et al., 2022) and Doddapaneni et al. (2023). Additionally, we remove sentences that are too short ($< 4$ words) or too long ($> 40$ words) as we found that the quality and reliability of embeddings deteriorate beyond these lengths. After this quality control procedure, we apply strict deduplication to eliminate any potential duplicates on the normalized sentences in the monolingual corpora.

**Sentence Embedding Model.** Prior work such as Samanantar (Ramesh et al., 2022) and NLLB (Costa-jussà et al., 2022) have employed the LaBSE (Feng et al., 2022) and LASER3 (Heffernan et al., 2022) models for bitext mining respectively. However, to determine the optimal sentence embedding model for our mining purposes, we analyze the correlation of the Semantic Textual Similarity Rating (Agirre et al., 2016) with the cosine similarity scores obtained using both sentence embedding models. We consider the STS dataset released by Ramesh et al. (2022) with a human rating for a set of 11 languages. Our analysis suggests that the cosine similarity scores of LaBSE sentence embeddings exhibit a stronger correlation with the human ratings on a macro scale, as shown in Table 5. Therefore, we adopt the LaBSE model as the primary sentence embedding model for our bitext mining and filtering pipeline and only fall back to LASER3 for the languages not supported by LaBSE. We use LASER3 for languages such as Kashmiri (Devanagari), Kashmiri (Arabic), Maithili, Manipuri (Bengali), Nepali, Sanskrit, and Sindhi (Arabic).

**Indexing.** To ensure a common embedding space for all languages, we utilized LaBSE (Feng et al., 2022) to compute the sentence embeddings for all the sentences. Our approach for mining parallel sentences involves searching through English; thus we indexed all the English sentences and treated the Indic language sentences as queries. To accommodate the large corpus of 429 million English sentences, we partitioned them into 5 shards and indexed each shard separately. In line with previous work (Ramesh et al., 2022), we utilized a FAISS Index[13] with 100K clusters and employed Product Quantization (Jégou et al., 2011) to reduce the dimensionality of the embeddings from 768 to 64, with each dimension represented by an 8-bit integer value.

---

[13]https://github.com/facebookresearch/faiss

Table 6: URLs and domains of the sources used for comparable corpora mining.

| Source | URL | Domain |
|---|---|---|
| isha | https://isha.sadhguru.org/in/en/wisdom | Religion, Education, Culture |
| mkb | https://www.pmindia.gov.in/en/mann-ki-baat | Government, News, Education |
| nios | https://nios.ac.in/online-course-material.aspx | Education |
| nptel | https://nptel.ac.in/courses | Education |
| pib | https://pib.gov.in/AllRelease.aspx | Government, News, Legal |
| spoken tutorial | https://spoken-tutorial.org/tutorial-search | Education |
| ugc | http://ugceresources.in | Education |
| vanipedia | https://tinyurl.com/2sf547tn | Religion, Education, Culture |

**Retrieval.** To retrieve parallel sentence pairs for a given query sentence ($S_L$) in language $L$, we use LaBSE (Feng et al., 2022) to compute the embedding of the query sentence and perform a search on the FAISS Index constructed from the English sentences. First, we retrieve the top $k$ ($k = 1024$) clusters by computing the cosine similarity between the cluster centroids and the query embedding. Subsequently, we search for ANNs within these clusters to retrieve the closest match. However, as pointed out by Ramesh et al. (2022), the similarity scores can vary when using quantized vectors ($64d$) while preserving the relative ranking among the sentence pairs. To ensure high-quality matches, we recompute the cosine similarity using the original $768d$ vectors and only retain pairs with a similarity score above a threshold of 0.80, indicating a strong semantic match. The process is repeated on each of the 5 English partitions, and only the highest-scoring match is retained.

## 4.2 Mining from Comparable Corpora

For Indian languages, we explore the mining of parallel corpora from comparable sources, i.e., multilingual websites containing high-quality parallel documents. We first align potentially parallel documents using heuristics to reduce the search space, followed by the extraction of high-quality parallel sentences from aligned documents.

**Data Curation.** We first identify several websites that publish content in multiple Indic languages. The articles on these websites are aligned across different languages, indicating they are exact translations of each other. Owing to this, the search space is reduced considerably as compared to monolingual corpus mining. The selected sources are diverse in domains covering a range of topics like Education, Legal, Religion, etc., and of high quality as verified by language experts. An overview of the sources is available in Table 6. We follow the same pre-processing steps to segment the documents into sentences, followed by language identification and toxicity filters.

**Indexing.** Similar to monolingual corpora, we use the LaBSE (Feng et al., 2022) model to index both the source and target sentences. Since the search space is much smaller in comparable corpora, we perform a full search over the entire target sentences in the corresponding document.

**Retrieval.** Let $S = \{s_1, s_2, \cdots, s_m\}$ be the set of source sentences and $T = \{t_1, t_2, \cdots, t_n\}$ be the set of target sentences. Let $f(s_i, t_i)$ be the scoring function for calculating the semantic similarity. Given that $m$ and $n$ are considerably smaller than the size of the monolingual corpus, we perform a total of $m \times n$ scoring computations. Following Artetxe & Schwenk (2019), we use the margin-based scoring (Equation 2) to find the closest semantic match between a given source and target sentences. The sentences under consideration are represented by the pair $(x, y)$. We denote the $k$ unique nearest neighbors of $x$ and $y$ in the other language as $NN_{k(x)}$ and $NN_{k(y)}$, respectively. We perform margin-based mining in both forward and backward directions to eliminate the candidate pairs with inconsistent alignment and retain only those that intersect, resulting in high-quality bitext pairs. Following Costa-jussà et al. (2022) we use a margin threshold of 1.06 with 4 nearest neighbors. Additionally, we set a cosine threshold of 0.80 for the high-resource languages and perform LID filtering to remove substandard sentence pairs. Considering the high memory requirements and the high variability of margin scores based on cluster sizes when operating in shards, employing margin-based mining for monolingual corpus with the current infrastructure was not feasible.

Table 7: Statistics of the bitext mining from comparable corpora (till Oct 2022).

| Language | Source | Extracted Pairs |
|---|---|---|
| asm_Beng | mkb, nios, pib, spoken-tutorial, vanipedia | 38,656 |
| ben_Beng | isha, mkb, nios, nptel, pib spoken-tutorial, ugc, vanipedia | 263,394 |
| brx_Deva | spoken-tutorial | 700 |
| guj_Gujr | isha, mkb, nios, nptel, pib spoken-tutorial, ugc vanipedia | 594,847 |
| hin_Deva | isha, mkb, nios, nptel, pib spoken-tutorial, ugc vanipedia | 891,464 |
| kan_Knda | isha, mkb, nios, nptel, pib spoken-tutorial, ugc vanipedia | 386,408 |
| mai_Deva | spoken-tutorial | 84 |
| mal_Mlym | isha, mkb, nios, nptel, pib spoken-tutorial, ugc vanipedia | 365,893 |
| mar_Deva | isha, mkb, nios, nptel, pib spoken-tutorial, ugc vanipedia | 453,371 |
| mni_Beng | mkb, pib | 22,322 |
| npi_Deva | isha, spoken-tutorial, vanipedia | 6,247 |
| ory_Orya | mkb, nios, pib spoken-tutorial, vanipedia | 125,143 |
| pan_Guru | mkb, nios, pib spoken-tutorial, vanipedia | 216,108 |
| san_Deva | spoken-tutorial | 702 |
| tam_Taml | isha, mkb, nios, nptel, pib spoken-tutorial, ugc, vanipedia | 455,965 |
| tel_Telu | isha, mkb, nios, nptel, pib spoken-tutorial, ugc, vanipedia | 449,239 |
| urd_Arab | mkb, nios, pib, vanipedia | 232,496 |
| **# Total** | | 4,503,039 |

$$\text{margin}(x, y) = \frac{\cos(x, y)}{\displaystyle\sum_{z \in NN_k(x)} \frac{\cos(x, z)}{2k} + \sum_{z \in NN_k(y)} \frac{\cos(y, z)}{2k}} \tag{2}$$

Following mining from Comparable Corpora, we extract 4.5 million sentence pairs across 17 Indic languages. The statistics and the sources for the mined bitext are available in Table 7.

### 4.3 Filtering Existing Mined Parallel Corpora

Over the years, several parallel corpora have been released for Indic languages (Kunchukuttan et al., 2018; Nakazawa et al., 2021b; Philip et al., 2021; Tiedemann, 2012) *inter alia*. The corpora are of varying quality and created using different approaches. We filter these existing corpora using some of the well-known practices to ensure we retain a high-quality subset for model training.

Particularly, a large collection of parallel corpora was mined as part of the NLLB project (Costa-jussà et al., 2022) using LASER3 embeddings (Heffernan et al., 2022). The corpus was mined using the margin-based threshold described in Equation (2), with a threshold of 1.06. The original dataset was not released by the authors of Costa-jussà et al. (2022). However, Allen AI[14] has replicated the efforts of Costa-jussà et al. (2022) and released the dataset closely matching the numbers reported by the authors of (Costa-jussà et al., 2022). Going forward, we use this dataset for our use-case and refer to it as Allen-NLLB [15]. The corpus contains 448 million sentence pairs across 19 Indic languages, with more than 10 million sentence pairs in 12 languages. However, on performing a manual inspection of the bitext, it was observed that a large majority of the sentences had misalignment and suboptimal parallel sentence pairs. Therefore, before using this corpus for training MT models, it is important to filter the corpus to remove the noisy sentence pairs.

Following our bitext mining in Section 4.1 and Section 4.2, we use the LaBSE model (Feng et al., 2022) with a cosine similarity threshold of 0.80 to filter the Allen-NLLB corpus. We also use the LASER3 model (Heffernan et al., 2022) as a fallback model for languages that are not supported by LaBSE (*viz.* Nepali, Maithili, Sanskrit, Sindhi (Arabic),

---

[14]https://allenai.org/
[15]https://huggingface.co/datasets/allenai/nllb

Table 8: Statistics of pre-filtering and post-filtering on existing mined parallel corpora consisting of NLLB (Costa-jussà et al., 2022) and Samanantar (Ramesh et al., 2022).

| Language | Pre-Filtering | Post-Filtering | Proportion (%) |
|---|---|---|---|
| asm_Beng | 5,285,401 | 565,282 | 10.70 |
| ben_Beng | 70,400,333 | 16,514,684 | 23.46 |
| guj_Gujr | 14,458,054 | 8,442,476 | 58.39 |
| hin_Deva | 43,149,229 | 11,056,172 | 25.62 |
| kan_Knda | 38,368,723 | 10,532,571 | 27.45 |
| kas_Arab | 647,348 | 125,243 | 19.35 |
| kas_Deva | 1,042,450 | 194,528 | 18.66 |
| mai_Deva | 4,438,382 | 62,359 | 1.40 |
| mal_Mlym | 49,599,699 | 10,832,342 | 21.84 |
| mar_Deva | 35,585,104 | 7,742,065 | 21.76 |
| mni_Beng | 490,089 | 347,108 | 70.83 |
| npi_Deva | 19,624,054 | 1,583,922 | 8.07 |
| ory_Orya | 14,700,484 | 2,887,960 | 19.65 |
| pan_Guru | 14,057,042 | 3,391,710 | 24.13 |
| san_Deva | 3,095,396 | 244,367 | 7.89 |
| snd_Arab | 8,924,699 | 2,129,054 | 23.86 |
| tam_Taml | 47,777,362 | 10,489,852 | 21.96 |
| tel_Telu | 51,248,532 | 11,826,104 | 23.08 |
| urd_Arab | 25,303,579 | 5,322,290 | 21.03 |
| **# Total** | 448,195,960 | 104,290,089 | 23.27 |

Kashmiri (Devanagari), Kashmiri (Arabic), Santali). Table 8 shows that upon filtering, the dataset is reduced from 448.1 million sentence pairs to 104.2 million sentence pairs, *i.e.* close to 76% of data has been dropped with quality filtering. For Santali, post LASER3 filtering, it was observed that the majority of the sentence pairs were dropped during the filtering process. Post-hoc human evaluation confirmed that most of the parallel data for Santali-English in the Allen-NLLB are noisy. We see the highest drops in Maithili, Sanskrit, and Nepali, which are considered to be low-resource languages. Surprisingly, even in high-resource languages like Hindi and Bengali, we see that close to 75% of the data has been dropped during filtering. Similarly, we also apply the same filtering criteria to Samanantar Corpus (Ramesh et al., 2022), as it was noted that Samanantar was mined with an older version of LaBSE model (Feng et al., 2022). Section 7.2 describes our analysis of the data quality v/s scale trade-off.

# 5 Modeling

## 5.1 Training Data

To train our translation models, we utilize a range of data sources, including data mined from text corpora (monolingual corpora & comparable sources), human-annotated collections (BPCC-H-Wiki and BPCC-H-Daily), and filtered versions of existing corpora (Ramesh et al., 2022; Costa-jussà et al., 2022). We describe our filtering techniques in Section 4.3. While these sources constitute the majority of our training corpus, we also incorporate additional human-labeled seed data from NLLB-seed (Costa-jussà et al., 2022; Maillard et al., 2023), ILCI (Jha, 2010; Choudhary & Jha, 2011), and MASSIVE (FitzGerald et al., 2022), totaling approximately 1.47 million sentence pairs. The ILCI (Jha, 2010; Choudhary & Jha, 2011) data is primarily distributed across domains such as health, tourism, agriculture, and entertainment, and contributes around 1.34 million parallel sentences across 16 languages. Furthermore, we augment our data with the Indic portions of MASSIVE (FitzGerald et al., 2022), which was released as Spoken Language Understanding data and closely resembles the data in BPCC-H-Daily. Professional annotators manually translate the sentences in this dataset and contribute 139,000 sentence pairs across seven languages. In total, we have approximately 230.5 million sentence pairs, out of which 2.2 million are gold sentence pairs that are manually annotated by professional translators. The distribution of the data sources across all languages is presented in Table 1.

Table 9: Statistics of the bi-text training data after deduplication with benchmarks.

| Language | Dataset Size | Language | Dataset Size |
|---|---|---|---|
| asm_Beng | 1,443,125 | mni_Beng | 386,916 |
| ben_Beng | 32,725,076 | mni_Mtei | 42,753 |
| brx_Deva | 1,13,839 | npi_Deva | 1,687,436 |
| doi_Deva | 24,160 | ory_Orya | 5,834,074 |
| gom_Deva | 97,660 | pan_Guru | 9,816,009 |
| guj_Gujr | 20,491,094 | san_Deva | 278,374 |
| hin_Deva | 39,144,013 | sat_Olck | 25,128 |
| kan_Knda | 23,285,105 | snd_Arab | 2,128,391 |
| kas_Arab | 135,843 | snd_Deva | 10,503 |
| kas_Deva | 200,094 | tam_Taml | 20,740,179 |
| mai_Deva | 87,888 | tel_Telu | 23,250,217 |
| mal_Mlym | 23,521,937 | urd_Arab | 6,176,951 |
| mar_Deva | 18,932,834 | | |
| | | **# Total** | 230,579,599 |

## 5.2 Preprocessing

We follow the following steps in sequential order for our data preprocessing pipeline.

**Standard Preprocessing.** We apply standard preprocessing, which includes removing redundant spaces, removing special characters, and normalizing the punctuations. Additionally, we convert the Indic numerals to English numerals using a dictionary-based mapping. This facilitates the use of English numerals both at the input and output stages of our model. However, a post-processing stage can be used to map English numerals back to their Indic equivalents, if required.

**Data Deduplication.** To prevent any potential data leakages, we apply strict deduplication with all the available benchmarks mentioned in Table 2. Our deduplication process involves standard preprocessing steps as mentioned above, followed by text lowercasing, removal of all punctuations, removal of spaces, and identification of potential matches on the monolingual side of both source and target sentences with the benchmarks. Correspondingly, any bi-text pairs associated with these monolingual matches are discarded, and only the remaining data is considered for training our models. As a result of this deduplication, our processed dataset contains a total of ~230.5M bi-text pairs. The per-language distribution is presented in Table 9

**Additional Preprocessing.** Based on human evaluation of the IndicTrans1 model (Ramesh et al., 2022), it was observed that the model exhibits poor performance in dealing with special cases: emails, URLs, dates, numbers, and special characters like percentages. These special cases share a common characteristic indicating that they should ideally not be translated by the model but should be reproduced as it is in the translation. To address this issue, we employ regular expression patterns to identify text spans corresponding to these special cases. Subsequently, we wrap these spans of text with special tags (<dnt> text span </dnt>) on the input side of the model, thereby providing implicit supervision to the model to retain these special cases in their original form in the translation. Note that, during training, we wrap the text spans within special tags only if they appear in both the source and target sentences.

**Script Unification.** Many Indic languages use scripts from the Brahmi family. To facilitate better transfer learning, wherever feasible, we apply rule-based script conversion using IndicNLP library (Kunchukuttan, 2020) to represent most of these languages in a single script (Devanagari). Thus, effectively our models are trained with five scripts, namely Perso-Arabic (Sindhi, Urdu, Kashmiri), Ol Chiki (Santali), Meitei (Manipuri), Latin (English), and Devanagari (all the rest of the languages).

### 5.3  Tokenization

Subword-level tokenization (Sennrich et al., 2016b; Kudo & Richardson, 2018) is an effective approach for segmenting text into smaller sub-word units to build neural machine translation (NMT) systems that are robust against out-of-vocabulary (OOV) issues. In this work, we train two separate tokenizers with the byte-pair-encoding (BPE) algorithm (Sennrich et al., 2016b) using SentencePiece[16] library (Kudo & Richardson, 2018) for English and Indic languages using a sampled corpus comprising monolingual sentences from IndicCorp v2 (Doddapaneni et al., 2023) and NLLB data (Costa-jussà et al., 2022). We chose SentencePiece library because of its in-built support for normalization. To ensure fair representation for each language, we upsample the low-resource languages and limit the high-resource languages to 3M sentences each. We use a vocab size of $32K$ and $128K$ for our English and Indic SPM models, respectively. We prepare the monolingual data for training our English and Indic SPM models using the preprocessing pipeline described in section 5.2 except for the additional preprocessing. We also add special tags (`<dnt>` and `</dnt>`) to the trained SPM models.

After tokenization, we prepend special indicator tags following prior multilingual NMT models (Johnson et al., 2017; Tan et al., 2019; Tang et al., 2021). In our case, we add both the source and target language tags to indicate the translation direction. Specifically, when translating text from English to Hindi, we format the sample as `eng_Latn hin_Deva {processed text}`.

### 5.4  Architecture

We train our English-centric neural models based on the transformer encoder-decoder architecture (Vaswani et al., 2017) using the fairseq library[17] (Ott et al., 2019). Our architecture comprises 18 encoder layers and 18 decoder layers, an input dimension of 1024, pre-normalization (Xiong et al., 2020) for all modules, a feedforward dimension of 8192, and 16 attention heads. The total parameter count is 1.1B. Additionally, we use the GELU activation (Hendrycks & Gimpel, 2016) instead of ReLU (Nair & Hinton, 2010).

### 5.5  Training

To perform well across a wide range of domains, we adopt FLORES-200 (Costa-jussà et al., 2022) multi-domain development set as our validation set rather than combining development sets from different benchmarks. However, this development set does not cover all the languages supported by our models. As a result, we extend the FLORES-200 development (Costa-jussà et al., 2022) set to additionally incorporate five more languages (*viz.* Bodo, Dogri, Konkani, Sindhi (Devanagari), Manipuri (Meitei)) to have a complete validation set to jointly optimize and achieve superior performance on all the 22 scheduled Indic languages (including 25 language script combinations). We also make the expanded version of the FLORES-200 development set (Costa-jussà et al., 2022) publicly available, and this has also been integrated into the official FLORES repository [18].

We employ the BLEU metric specifically for checkpointing purposes, using validation BLEU scores to indicate the model's performance on the aforementioned validation set. This choice is motivated by BLEU providing valuable insights into the model's macro-level performance, making it a useful diagnostic tool for tracking the model's progress during training. However, it may not be the most suitable choice for fine-grained evaluations. This differs from IndicTrans1 (Ramesh et al., 2022), which utilizes validation loss for checkpointing. By incorporating the checkpointing based on validation BLEU scores, we can ensure that the training of our models progresses based on their performance on the validation set, leading to an overall improved model.

Our model training paradigm comprises two distinct phases: auxiliary training and downstream training, which are described below.

**Auxiliary Training.**   The first phase of our model training paradigm, termed auxiliary training, involves training intermediate models to augment large amounts of monolingual corpora through back translation. Back-translation

---

[16] https://github.com/google/sentencepiece
[17] https://github.com/facebookresearch/fairseq
[18] https://github.com/openlanguagedata/flores

Table 10: Details of the hyperparameters used for stage 1 training and stage 2 fine-tuning. Please note that we reset the learning scheduler, dataloaders, and optimizer for stage 2 fine-tuning.

| Hyperparameters | Stage 1 training | Stage 2 fine-tuning |
|---|---|---|
| Optimizer | Adam (Kingma & Ba, 2014) | Adam (Kingma & Ba, 2014) |
| Beta values $(\beta_1, \beta_2)$ | $(0.9, 0.98)$ | $(0.9, 0.98)$ |
| Learning rate | $5e$-4 | $3e$-5 |
| Scheduler | Inverse sqrt | Inverse sqrt |
| Criterion | Cross-entropy | Cross-entropy |
| Label smoothing (Szegedy et al., 2016) | 0.1 | 0.1 |
| Warmup learning rate | $1e$-7 | $1e$-7 |
| Warmup steps | $4,000$ | $2,000$ |
| Gradient clipping | 1.0 | 1.0 |
| Dropout (Srivastava et al., 2014) | 0.2 | 0.2 |
| Patience | 10 | 10 |
| Effective batch size | 262K | 32K |
| Mixed precision training | FP16 | FP16 |
| Maximum update steps | 1M | 1M |
| Validation interval | $2,500$ | $1,000$ |
| Maximum sequence length | 256 | 256 |
| Checkpoint metric | BLEU @ beam = 1 | BLEU @ beam = 1 |

(Sennrich et al., 2016a; Edunov et al., 2018) is a technique that is effective in improving the performance of machine translation models. We adopt a deterministic curriculum strategy as proposed by Mohiuddin et al. (2022), wherein we first train the models from scratch on the entire parallel corpora listed in Table 1, followed by stage 2 fine-tuning on high-quality seed data including BPCC-H-Wiki and the NLLB seed (Costa-jussà et al., 2022; Maillard et al., 2023), to improve the models further. Our approach differs from theirs in that we exclusively consider high-quality human-generated data for stage 2 model fine-tuning rather than selecting the top $p\%$ of bitext pairs from the original data based on a quality measure. Another prominent advantage of using our human-generated data is that it provides multi-domain coverage, thereby allowing us to optimize across multiple domains, which may not be feasible when selecting a subset of bitext pairs based on quality. We list all the hyperparameters used in both stage 1 and stage 2 training in Table 10.

**Downstream Training.** In the second phase, we train our models on the augmented parallel corpora that combine original data with back-translated data. Mainly, we follow tagged back translation (Caswell et al., 2019) to provide additional supervision to the model to distinguish between the different data sources during training. We prepend the special symbol to the synthetically augmented data while keeping the original data intact. We follow the same training hyperparameters and two-stage training strategy as the auxiliary training. Table 10 shows all the hyperparameters used in both stage 1 and stage 2 training.

## 5.6 Data Augmentation

Using existing parallel corpora as training data may eventually lead to saturation in model performance. To address this, researchers have proposed data augmentation techniques to enhance data diversity and improve model performance. One such approach involves augmenting pseudo-parallel corpora by leveraging diverse monolingual corpora. Back translation (Sennrich et al., 2016a; Edunov et al., 2018) is a widely used technique to synthetically augment training data for improving translation models. Given the large scale of our models, we adopt this approach and generate back-translated data, which is approximately 1.75 times the size of the original training data. To generate back translation data, we first identify potential sources of monolingual data for English and Indic languages, intending to maximize both domain coverage and distributional diversity to improve the models. We use the intermediate checkpoints of IndicTrans2 to generate the backtranslated data and combine the augmented data along with the training data to further improve our models.

Table 11: Statistics of the monolingual data used for backtranslation.

| Language | English BT Data | Indic BT Data | Language | English BT Data | Indic BT Data |
|---|---|---|---|---|---|
| asm_Beng | 14,569,760 | 5,433,796 | mni_Beng | 17,437,961 | 60,224 |
| ben_Beng | 17,928,856 | 34,987,743 | mni_Mtei | 17,709,470 | 33,233 |
| brx_Deva | 17,597,825 | 144,246 | npi_Deva | 20,567,992 | 29,997,511 |
| doi_Deva | 18,157,864 | 44,291 | ory_Orya | 19,528,727 | 15,341,924 |
| gom_Deva | 13,478,802 | 2,937,179 | pan_Guru | 17,476,704 | 29,968,101 |
| guj_Gujr | 21,447,703 | 29,994,809 | san_Deva | 11,198,794 | 9,744,059 |
| hin_Deva | 20,648,256 | 37,472,261 | sat_Olck | 9,799,342 | 32,346 |
| kan_Knda | 10,970,576 | 32,496,971 | snd_Arab | 8,918,509 | 4,298,898 |
| kas_Arab | 12,717,571 | 44,276 | snd_Deva | 6,479,694 | 25,264 |
| kas_Deva | 11,599,085 | 154,465 | tam_Taml | 22,647,544 | 32,488,783 |
| mai_Deva | 15,598,363 | 1,813,669 | tel_Telu | 21,767,767 | 32,494,937 |
| mal_Mlym | 17,888,824 | 32,495,047 | urd_Arab | 20,006,656 | 33,471,969 |
| mar_Deva | 15,849,536 | 34,994,281 | | | |
| | | | **# Total** | 401,992,181 | 400,970,283 |

**English Data for Back Translation.** For back translation, we source English data from several sources, including the English side of IndicCorp v2 (Doddapaneni et al., 2023), the English side of the Indic subset of the NLLB data (Costa-jussà et al., 2022), and English data from a few high-resource pairs (eng_Latn – {fra_Latn, por_Latn, spa_Latn, ces_Latn}) of NLLB data (Costa-jussà et al., 2022), along with additional miscellaneous sources like Simple Wikipedia[19] and DD News.[20] We subjected this set of English sentences to standard preprocessing, as outlined in Section 5.2, and then filtered the set to retain only sentences with a minimum of five and a maximum of 100 words. As described in Section 5.2, we deduplicate this set of sentences with all the benchmarks available. Additionally, we deduplicate this set with the training data to ensure more diversity in English data and sample candidate sentences from a non-overlapping set. From this reduced candidate set, we randomly sampled approximately 400 million sentences for back translation, following an approximate distribution of 55% IndicCorp, 20% NLLB Indic, 20% NLLB HighRes, and 5% Miscellaneous sources. To ensure language-script diversity, we randomly subdivide the 400 million set into 25 parts, corresponding to the supported language-script combinations. We utilize the En-Indic model with a beam value of 5 to generate back-translated data. We proportionally distribute the English data across different language-script combinations based on the normalized chrF++ (Popović, 2017) scores across all language-script combinations described below in Equation (3) on the expanded version of FLORES-200 validation set (Goyal et al., 2022; Costa-jussà et al., 2022) described in section 5.5. Table 11 describes the distribution of the English data we consider for back-translation for each language-script combination.

$$\text{Count}(\text{lang}_i) = \frac{\text{chrF++}(\text{lang}_i)}{\sum_j \text{chrF++}(\text{lang}_j)} \times N \tag{3}$$

Here, chrF++($\text{lang}_i$) represents the normalized chrF++ score for language-script combination $\text{lang}_i$, and N is the total number of English monolingual sentences to be used for back translation.

**Indic Data for Back Translation.** We source the Indic monolingual data from IndicCorp v2 (Doddapaneni et al., 2023) and the Indic side of the NLLB data (Costa-jussà et al., 2022) to generate back-translated data to improve our En-Indic model. However, it is essential to note that our sources for Indic monolingual data are limited, which limits the amount of data we can sample from each language-script combination. As a result, we do not adopt any proportional sampling based on the model's performance on the FLORES-200 validation set, as we do when generating back-translated data from monolingual English data. Therefore, we follow a simple strategy to include all the available monolingual data from languages, where the availability of diverse monolingual data is scarce (less than 20 million

---

[19] https://simple.wikipedia.org/wiki/Main_Page
[20] https://ddnews.gov.in/

sentences) and uniformly sample from the high-resource languages. We apply the same preprocessing and data deduplication steps as described above for back-translation from English. We use the Indic-En model with a beam value of 5 for generating back-translation data. We provide the details of the Indic monolingual data distribution used for back translation in Table 11.

### 5.7 Postprocessing

Since our En-Indic model is trained on script-unified data, the output it generates must be mapped back to the native script of the target language. Therefore, we perform rule-based script conversion using the IndicNLP library (Kunchukuttan, 2020) and map the script-unified output to the corresponding native Indic script. Importantly, this postprocessing is only necessary for the En-Indic model, as the outputs of the Indic-En model are already in the desired format.

## 6 Evaluation

### 6.1 Models Compared

We compare our trained models with publicly and commercially available existing models and systems:

- **IndicTrans1.** Ramesh et al. (2022) curated large parallel corpora by large-scale mining and trained multilingual transformer models (474M parameters) on this mined Samanantar dataset. These models support only 11 major Indian languages.

- **NLLB.** Costa-jussà et al. (2022) trained a multi-way many-to-many 54.5B Mixture of Experts (MoE) model supporting 200 languages. This model supports 20 language-script combinations from the set of scheduled Indic languages, providing coverage in at least one script for 19 of the 22 scheduled Indic languages.

- **M2M-100.** Fan et al. (2020) released many-to-many models supporting translation between 100 languages with language-family-specific decoders trained using English-centric data and non-English-centric data. We use their best model (12B parameters) supporting 12 of the 22 scheduled Indic languages for our comparison.

- **Microsoft Azure Translate.**[21] Microsoft Azure Translate is a commercial translation engine supporting translation between 16 out of the 22 scheduled Indic languages at the time of writing.

- **Google Translate.**[22] Google Translate is a commercial translation engine supporting translation between 19 out of the 22 scheduled Indic languages at the time of writing.

- **GPT-3.5.** GPT-3.5 is a commercially available, large language model developed by OpenAI,[23] based on the GPT-3 architecture (Brown et al., 2020), but with additional improvements and optimizations like instruction fine-tuning, reinforcement learning with human feedback (Ouyang et al., 2022), and enhanced conversational support. It is a decoder-only model trained using the causal language modeling objective and is currently available as a propriety system accessible via a paid API. We evaluate the `gpt-3.5-turbo` model, which accepts chat format messages, on our IN22 benchmark in a zero-shot setting.

For proprietary models, it is difficult to do fair comparisons since little information is available about models and training. Thus, the reported results should be seen as a reasonable approximation. In this work, we will henceforth adopt the specific shorthand notations: the IndicTrans1 model will be referred to as IT1, the M2M-100 model as M100, the NLLB 1.2B distilled model as N1.2, the NLLB 54.5B MoE model as N54, Google Translate as Goog, Microsoft Azure Translate as Az, and our IndicTrans2 model as IT2. The predictions of Microsoft Azure, Google Translate, and GPT3.5 were generated using the respective APIs, with data retrieved on 10th May 2023.

---

[21] https://azure.microsoft.com/en-us/products/cognitive-services/translator
[22] https://cloud.google.com/translate
[23] https://platform.openai.com/docs/models/gpt-3-5

## 6.2 Benchmarks

We evaluate our trained models (auxiliary and downstream) on our IN22 benchmark and all the publicly available benchmarks: FLORES-200 (Goyal et al., 2022; Costa-jussà et al., 2022), WAT 2020 (Nakazawa et al., 2020), WAT 2021 (Nakazawa et al., 2021a), WMT 2014 (Bojar et al., 2014), WMT 2019 (Barrault et al., 2019), WMT 2020 (Barrault et al., 2020), UFAL (Ramasamy et al., 2012) and NTREX (Federmann et al., 2022).

We list the details of the existing benchmarks below.

- **IN22** is a comprehensive benchmark for evaluating machine translation performance in multi-domain, $n$-way parallel contexts across 22 Indic languages. It comprises three distinct subsets, namely IN22-Wiki, IN22-Web, and IN22-Conv. The Wikipedia and Web sources subsets offer diverse content spanning news, entertainment, culture, legal, and India-centric topics. Meanwhile, the conversation domain subset is designed to assess translation quality in typical day-to-day conversational-style applications.

  From now on, we merge Wikipedia and Web Sources subsets, to create a consolidated set referred to as IN22-Gen for translation evaluation. Our motivation for this is that these two subsets share a common language style, albeit with varying topics, whereas the Conversation subset is different in both language style and usage context.

- **FLORES-101/200** (Goyal et al., 2022; Costa-jussà et al., 2022) is a multi-domain general-purpose benchmark designed for evaluating translations across 200 languages, including 19 Indic languages. The English sentences are source-original and have been translated into other languages. It comprises sentences sourced from Wikimedia entities with equal portions of news, travel, and non-fiction content from children's books. Tables 2 and 52 provide further details on the statistics and fine-grained domain coverage.

- **NTREX** (Federmann et al., 2022) is a news-domain benchmark that expands coverage of languages of test data from WMT 2019 (Barrault et al., 2019) to 128 languages. Out of these, 13 are scheduled Indic languages.

- **WMT** has created benchmarks for selected Indic languages as part of shared tasks in 2014 (Hindi) (Bojar et al., 2014), 2019 (Gujarati) (Barrault et al., 2019) and 2020 (Tamil) (Barrault et al., 2020).

- **WAT 2020/2021** (Nakazawa et al., 2020; 2021a) included support for translations for 8 Indic languages in the news domain. In addition, they released data for Hindi-English in Information Technology and WikiNews domains. WAT 2021 (Nakazawa et al., 2021a) created a benchmark for translation between 10 Indic languages and English.

- **UFAL** (Ramasamy et al., 2012) is an English-Tamil bilingual benchmark created from publicly available websites. The benchmark consists of English sentences from domains such as cinema, news, and some biblical sources.

Moving forward, we consider IN22 and FLORES-200 (Costa-jussà et al., 2022) as the primary benchmarks to evaluate all the translation models. The results obtained from these benchmarks are reported and discussed in Section 7. Additionally, the performance of the models on other benchmarks is presented in Appendix B. Note that almost all the test sets are English-original, but have been used for Indic-to-English evaluation as well as Indic-Indic evaluation.

## 6.3 Metrics

Several metrics have been developed over the years for automatically assessing translation quality, including string-based metrics such as BLEU (Papineni et al., 2002), chrF (Popović, 2015), and chrF++ (Popović, 2017), and model-based metrics such as BLEURT (Sellam et al., 2020), COMET (Rei et al., 2020; 2022) and PRISM (Thompson & Post, 2020). Recent research (Kocmi et al., 2021; Freitag et al., 2021; 2022) has shown that model-based metrics tend to exhibit a stronger correlation with human judgment. However, these model-based metrics are limited to languages represented in the underlying pre-trained model. They are trained on human judgment data from a few languages, and their performance on many low-resource languages has not been evaluated. We briefly describe all the metrics used in our work below.

**BLEU.** BLEU (Papineni et al., 2002) has been a standard and widely used metric for evaluating machine translation quality. However, a significant limitation of the standard BLEU metric is its tokenization dependency. To overcome this, sacreBLEU[24] (Post, 2018) provides standardization in terms of tokenization to ensure a fair comparison. We use sacreBLEU for evaluating our En-Indic and Indic-En trained models. We use the in-built default `mteval-v13a` tokenizer[25] for Indic-En[26] and Indic tokenizer from IndicNLP (Kunchukuttan, 2020) for En-Indic[27] evaluations. Therefore, we first tokenize the machine translations and reference translations using Indic tokenizers from IndicNLP[28] (version 0.92) and Urduhack[29] (ALi, 2019) libraries before running sacreBLEU.

**chrF++.** chrF++ (Popović, 2017), an extension of the chrF metric (Popović, 2015) that additionally considers word unigrams and bigrams, and is better correlated with human judgments and uses sacreBLEU to compute chrF++ scores. Similar to the tokenizers used for BLEU, for Indic-En[30] evaluation, we use the in-built default `mteval-v13a` tokenizer, while for En-Indic[31] evaluation, we use Indic tokenizers from IndicNLP and Urduhack libraries to tokenize the machine translations and reference translations before running sacreBLEU.

**COMET.** COMET is a model-based machine translation evaluation metric introduced by Rei et al. (2020) to address some of the limitations of existing metrics such as BLEU. However, one of the prominent concerns about COMET is its extensibility to low-resource languages. Therefore, in this study, we report COMET-DA scores for the top 13 Indian languages: Assamese, Bengali, Gujarati, Hindi, Kannada, Malayalam, Marathi, Nepali, Odia, Punjabi, Tamil, Telugu, and Urdu that are supported by the XLM-RoBERTa (Conneau et al., 2020) model. Specifically, we conduct a reference-based evaluation using the COMET-22 DA model[32] (Rei et al., 2022).

**Choosing the Primary Metric.** COMET, the most recommended model-based metric (Kocmi et al., 2021), does not support all the 22 Indic languages since they are not represented in XLM-R (Conneau et al., 2020) which is the underlying model on which COMET is based. Conversely, BLEU has several significant limitations, including its tokenization dependency and preferential bias towards translations that are closer to the reference translations in terms of lexical and word order (Ananthakrishnan et al., 2006). Particularly in the context of morphologically rich Indian languages, BLEU is limited in addressing morphological variants since it relies on exact word matches. Furthermore, chrF++ is more suitable for evaluating translation quality in languages with complex morphology and inflections, such as Indian languages. In this work, we, therefore, primarily rely on chrF++ as our primary metric for evaluating translation quality. We also report additional metrics such as BLEU (Papineni et al., 2002) and COMET (Rei et al., 2022). In addition, we also perform paired bootstrap resampling-based statistical significance tests (Koehn, 2004) for all the metrics following the default configurations.

## 6.4 Generation

To generate predictions using IndicTrans2, initially, we preprocess and tokenize the source sentences from the benchmark test set, following the steps described in Section 5.2 and Section 5.3, respectively. Subsequently, we feed the tokenized sentences into the trained models as input to generate candidate translations. We utilize beam search with a beam value of 5 for our trained models. Finally, we employ post-processing techniques, as detailed in Section 5.7, to map the script unified output to the corresponding native script. For other baseline systems, we follow their documented inference procedure. For all the open-source baseline models, we use the same beam size of 5.

---

[24] https://github.com/mjpost/sacrebleu

[25] https://github.com/moses-smt/mosesdecoder/blob/master/scripts/generic/mteval-v13a.pl

[26] Indic-En sacreBLEU BLEU signature:
`nrefs:1|case:mixed|eff:no|tok:13a|smooth:exp|version:2.3.1`

[27] En-Indic sacreBLEU BLEU signature:
`nrefs:1|case:mixed|eff:no|tok:none|smooth:exp|version:2.3.1`

[28] https://github.com/anoopkunchukuttan/indic_nlp_library

[29] https://github.com/urduhack/urduhack

[30] Indic-En sacreBLEU chrF++ signature:
`nrefs:1|case:mixed|eff:yes|nc:6|nw:2|space:no|version:2.3.1`

[31] En-Indic sacreBLEU chrF++ signature:
`nrefs:1|case:mixed|eff:yes|nc:6|nw:2|space:no|version:2.3.1`

[32] https://huggingface.co/Unbabel/wmt22-comet-da

Table 12: chrF++ scores of all the systems on the IN22-Gen Evaluation set in the En-Indic and Indic-En directions. The best-performing system is bolded, while underlined results indicate significant performance difference where IT2 outperforms the system. The row *Avg.* means the average score of all the languages that system X supports. $\Delta$ represents the difference between the average scores of IT2 and the average scores of system X for the subset of languages that both X and IT2 support. A positive value for $\Delta$ indicates IT2 is better than X and vice-versa. † indicates completely off-target translations.

| | En-Indic | | | | | | | Indic-En | | | | | | |
| language | IT1 | M100 | N1.2 | N54 | IT2 | Goog | Az | IT1 | M100 | N1.2 | N54 | IT2 | Goog | Az |
|---|---|---|---|---|---|---|---|---|---|---|---|---|---|---|
| *asm_Beng* | 35.9 | - | 41.7 | 42.9 | **47.1** | 45.5 | 45.0 | 56.1 | - | 63.1 | **66.5** | 65.8 | 65.1 | 60.8 |
| *ben_Beng* | 48.6 | 40.6 | 47.8 | 49.2 | **51.8** | 49.9 | 49.8 | 58.4 | 52.8 | 60.8 | 63.5 | 63.2 | **64.1** | 60.2 |
| *brx_Deva* | - | - | - | - | **47.8** | - | - | - | - | - | - | **62.1** | - | - |
| *doi_Deva* | - | - | - | - | **57.8** | 47.8 | - | - | - | - | - | **72.6** | 67.3 | - |
| *gom_Deva* | - | - | - | - | **45.2** | 41.4 | 41.1 | - | - | - | - | **59.2** | 57.8 | 51.1 |
| *guj_Gujr* | 47.2 | 19.9 | 48.3 | 49.5 | **53.5** | 52.2 | 50.8 | 60.3 | 11.8 | 63.9 | 66.3 | **66.5** | **66.5** | 62.4 |
| *hin_Deva* | 53.3 | 47.1 | 52.8 | 53.9 | **56.7** | 54.6 | 54.1 | 60.7 | 54.9 | 62.2 | 64.8 | **65.4** | 64.8 | 62.0 |
| *kan_Knda* | 46.7 | 15.3 | 47.3 | 48.6 | **51.0** | 48.1 | 49.4 | 58.8 | 12.6 | 62.4 | 65.1 | 64.2 | 64.5 | 61.7 |
| *kas_Arab* | - | - | 34.6 | 35.4 | **40.2** | - | - | - | - | 54.9 | 58.2 | **60.4** | - | - |
| *mai_Deva* | - | - | 44.9 | 44.7 | **48.7** | 38.3 | 45.2 | - | - | 62.1 | 65.1 | 64.8 | 64.0 | 61.0 |
| *mal_Mlym* | 45.7 | 31.2 | 45.4 | 46.7 | **50.9** | 49.0 | 48.6 | 56.9 | 44.8 | 59.8 | 62.8 | **64.5** | 62.7 | 60.4 |
| *mar_Deva* | 44.3 | 34.5 | 44.7 | 46.1 | **51.0** | 47.1 | 48.2 | 57.7 | 46.9 | 60.9 | 63.6 | 63.7 | **64.4** | 60.3 |
| *mni_Mtei* | - | - | - | - | **44.6** | 35.0 | - | - | - | - | - | **57.9** | 50.7 | - |
| *npi_Deva* | - | 17.7 | 44.8 | 44.8 | **49.0** | 45.5 | 46.3 | - | 40.1 | 65.0 | 68.0 | 67.7 | **69.0** | 63.8 |
| *ory_Orya* | 40.3 | 8.2 | 42.4 | 41.5 | 43.9 | 40.5 | **45.4** | 60.0 | 14.4 | 63.7 | **66.7** | 66.2 | 64.6 | 61.1 |
| *pan_Guru* | 48.0 | 25.0 | 48.5 | 49.5 | 50.6 | **52.7** | 50.4 | 57.2 | 38.2 | 60.4 | 63.1 | **63.4** | 62.7 | 58.5 |
| *san_Deva* | - | - | 25.5 | 28.1 | **38.8** | 32.0 | - | - | - | 48.2 | 51.3 | **54.8** | 53.8 | - |
| *sat_Olck* | - | - | 1.0 † | 25.5 | **33.4** | - | - | - | - | 36.3 | 41.4 | **45.3** | - | - |
| *snd_Deva* | - | - | - | - | **36.6** | - | - | - | - | - | - | **57.3** | - | - |
| *tam_Taml* | 45.5 | 12.3 | 47.0 | 47.5 | **49.5** | 48.5 | 49.4 | 53.9 | 26.3 | 56.9 | 59.1 | **59.8** | 59.6 | 56.8 |
| *tel_Telu* | 46.5 | - | 48.1 | 49.5 | **52.4** | 50.8 | 50.6 | 57.7 | - | 61.3 | 64.4 | **64.8** | 64.6 | 61.2 |
| *urd_Arab* | - | 45.0 | 62.1 | 63.7 | 68.2 | 63.9 | **69.0** | - | 52.6 | 68.3 | 71.2 | **73.0** | 71.8 | 68.2 |
| *Avg.* | 45.6 | 27.0 | 42.8 | 45.1 | 48.6 | 46.8 | 49.6 | 58.0 | 35.9 | 59.4 | 62.4 | 63.1 | 63.2 | 60.6 |
| $\Delta$ | 5.2 | 25.4 | 6.4 | 4.1 | - | 4.2 | 1.7 | 6.3 | 29.3 | 3.7 | 0.7 | - | 1.1 | 4.2 |

## 6.5 Evaluation

Following the generation of candidate translations, we evaluate their quality using the automatic metrics mentioned in Section 6.3. We apply standard processing techniques to compute the evaluation metrics, followed by running sacreBLEU. We use the standard Moses tokenizer for English, while for Indic languages, we perform tokenization using IndicNLP and Urduhack libraries. We release our evaluation procedure and scripts to ensure reproducibility. We follow the same evaluation procedure for all systems listed in Section 6.1.

# 7 Results and Discussion

## 7.1 Comparison with Existing Systems

**Evaluation on IN22-Gen Set.** We evaluate the translation quality of multiple En-Indic and Indic-En MT models on the IN22-Gen set. The results are presented in Table 12. We observe that IndicTrans2 significantly improves translation quality over IndicTrans1 (Ramesh et al., 2022) with an average improvement of 5.2 points in the En-Indic direction and 6.3 points improvement in the Indic-En direction. The proposed model outperforms the best commercial and open-source models for En-Indic translation by 1.7 and 4.1 points, respectively. For Indic-En translation, the IndicTrans2 is comparable to existing models, with a delta of +0.7 and +1.1 for best open-source and commercial models, respectively. The results further highlight the substantial improvements made on low-resource languages such

Table 13: chrF++ scores of all the systems on the FLORES-200 devtest set in the En-Indic and Indic-En direction. The best-performing system is bolded, while underlined results indicate significant performance difference where IT2 outperforms the system. *Avg.* means the average score of all the languages that system X supports. $\Delta$ represents the difference between the average scores of IT2 and the average scores of system X for the subset of languages that both X and IT2 support. A positive value for $\Delta$ indicates IT2 is better than X and vice-versa. † indicates completely off-target translations.

| | En-Indic | | | | | | | Indic-En | | | | | | |
| language | IT1 | M100 | N1.2 | N54 | IT2 | Goog | Az | IT1 | M100 | N1.2 | N54 | IT2 | Goog | Az |
|---|---|---|---|---|---|---|---|---|---|---|---|---|---|---|
| asm_Beng | 33.5 | - | 38.6 | 39.0 | **43.3** | 40.9 | 42.8 | 48.1 | - | 55.3 | **57.8** | 56.9 | **57.7** | 53.4 |
| ben_Beng | 49.5 | 44.3 | 50.1 | 52.2 | **54.3** | 53.8 | 53.4 | 56.9 | 54.7 | 60.3 | 62.2 | 62.4 | **63.2** | 59.9 |
| guj_Gujr | 50.4 | 21.9 | 52.0 | 53.6 | **56.0** | 55.5 | 55.6 | 58.7 | 12.1 | 65.2 | 66.6 | 67.0 | **68.0** | 62.9 |
| hin_Deva | 56.6 | 53.2 | 56.5 | 58.2 | 59.6 | **60.2** | 59.6 | 61.3 | 60.0 | 65.0 | 66.5 | 67.5 | **68.0** | 65.3 |
| kan_Knda | 50.9 | 16.5 | 53.0 | 54.3 | 56.1 | **56.2** | 56.1 | 54.6 | 12.0 | 59.5 | 61.0 | 61.5 | **62.1** | 58.6 |
| kas_Arab | - | - | 37.2 | 38.0 | **39.7** | - | - | - | - | 57.8 | **60.2** | 59.7 | - | - |
| kas_Deva | - | - | 18.7 | 18.8 | **19.2** | - | - | - | - | 47.7 | **50.6** | 48.3 | - | - |
| mai_Deva | - | - | 46.1 | 47.5 | 50.5 | 41.4 | **51.0** | - | - | 66.6 | 68.3 | **69.5** | 68.8 | 65.2 |
| mal_Mlym | 49.8 | 37.8 | 49.2 | 52.6 | **57.3** | **57.3** | 56.8 | 57.2 | 51.7 | 61.8 | 62.9 | 64.3 | **64.5** | 61.3 |
| mar_Deva | 45.9 | 38.6 | 46.5 | 48.3 | 51.3 | **51.4** | 49.4 | 56.4 | 50.4 | 61.6 | 63.8 | 64.3 | **65.3** | 61.5 |
| mni_Beng | - | - | 37.1 | **42.1** | 38.2 | - | - | - | - | 50.5 | 50.7 | **52.9** | - | - |
| npi_Deva | - | 15.5 | 49.2 | 46.4 | **57.2** | 55.7 | 53.4 | - | 41.1 | 65.2 | 66.9 | 68.1 | **68.7** | 63.9 |
| ory_Orya | 44.2 | 8.5 | 47.6 | 47.0 | 49.2 | **53.9** | 50.2 | 55.5 | 14.3 | 61.8 | 64.4 | **64.9** | 64.3 | 60.5 |
| pan_Guru | 50.6 | 26.8 | 50.9 | 51.3 | 53.5 | **54.3** | 54.2 | 60.0 | 44.5 | 64.5 | 66.3 | 66.4 | **67.1** | 62.7 |
| san_Deva | - | - | 25.8 | 27.1 | **31.6** | 31.3 | - | - | - | 47.8 | 50.7 | **51.6** | 51.2 | - |
| sat_Olck | - | - | 0.9 † | 27.0 | **28.4** | - | - | - | - | 38.7 | **44.3** | 39.3 | - | - |
| snd_Arab | - | 28.6 | 48.9 | 49.6 | 44.9 | 50.4 | **51.1** | - | 19.6 | 64.0 | **66.3** | 65.1 | 66.6 | 59.8 |
| tam_Taml | 49.5 | 13.2 | 53.3 | 54.0 | **57.2** | 56.0 | 56.1 | 54.1 | 33.0 | 58.9 | 60.8 | **61.3** | 61.5 | 57.9 |
| tel_Telu | 52.6 | - | 55.0 | 56.5 | **59.4** | 59.0 | 57.5 | 58.2 | - | 63.4 | 65.5 | 66.1 | **66.7** | 63.4 |
| urd_Arab | - | 39.9 | 49.4 | 50.3 | **52.2** | 51.3 | 51.6 | - | 48.8 | 60.9 | 62.9 | 62.0 | **63.7** | 59.3 |
| *Avg.* | 48.5 | 28.7 | 43.3 | 45.7 | 48.0 | 51.8 | 53.3 | 56.5 | 36.9 | 58.8 | 60.9 | 61.0 | 64.2 | 61.0 |
| $\Delta$ | 5.8 | 25.4 | 4.7 | 2.3 | - | 0.3 | 0.2 | 7.4 | 27.7 | 2.2 | 0.1 | - | -0.5 | 3.5 |

as Dogri (+10), Konkani (+3.8), Kashmiri (+4.8), Maithili (+3.8), Manipuri (+9.6) for En-Indic and Dogri (+5.3), Manipuri (+7.2), Santali (+3.9) for Indic-En translations when compared to the next best model. The observed gains can be attributed to using high-quality human-annotated BPCC-H Wiki data for training MT models. These findings suggest that the proposed model is well-suited for adoption in the Indian subcontinent, aligning with the objective of building models suitable for Indian languages. Additionally, we also report the COMET (Rei et al., 2022) and BLEU (Papineni et al., 2002) scores for our models in Table 39 and Table 42 (in Appendix B) where we observe similar trends, indicating that the observations are robust across different metrics.

**Evaluation on FLORES-200.** We also evaluate the MT models on the FLORES-200 benchmark (Costa-jussà et al., 2022). Through this evaluation, we aim to assess the model's translation quality on more general content, complementing the evaluation on our IN22 test set which is India-centric. Therefore, by evaluating our models on both IN22 and FLORES-200, we can effectively gauge the model's translation quality in different settings. The results in Table 13 obtained from the FLORES-200 test set show a similar trend as IN22, with IndicTrans2 being the best open-source model performing competitively with commercial models. The results also show a significant improvement from IndicTrans1 to IndicTrans2, with +5.8 and +7.4 points improvement in En-Indic and Indic-En translations, respectively. We also report the COMET and BLEU scores for the FLORES-200 benchmark in Table 41 and Table 44 (in Appendix B).

**Evaluation on IN22-Conv Set.** While both the IN22-Gen Set and FLORES-200 (Costa-jussà et al., 2022) focus on written sentences, the real-world usage of MT is often task-oriented and involves conversational language. To address this, all the models are further evaluated on the IN22-Conv Set, which is designed to test the translation quality of MT models on conversational language and daily use scenarios. The results of all the models on the IN22-Conv Set are

Table 14: chrF++ scores of all the systems on the IN22-Conv Evaluation set in the En-Indic and Indic-En directions. The best performing system is bolded, while underlined results indicate significant performance difference where IT2 outperforms the system. *Avg.* means the average score of all the languages that system X supports. Δ represents the difference between the average scores of IT2 and the average scores of system X for the subset of languages that both X and IT2 support. A positive value for Δ indicates IT2 is better than X and vice-versa. † indicates completely off-target translations.

| | En-Indic | | | | | | | Indic-En | | | | | | |
|---|---|---|---|---|---|---|---|---|---|---|---|---|---|---|
| *language* | IT1 | M100 | N1.2 | N54 | IT2 | Goog | Az | IT1 | M100 | N1.2 | N54 | IT2 | Goog | Az |
| *asm_Beng* | 36.4 | - | 42.6 | 43.4 | **46.8** | 43.6 | 46.6 | 52.5 | - | 58.7 | 59.8 | 62.9 | **64.0** | 62.1 |
| *ben_Beng* | 47.5 | 39.7 | 47.1 | 48.5 | **49.7** | 48.9 | 48.8 | 55.2 | 48.1 | 55.4 | 57.0 | 58.4 | **59.6** | 58.3 |
| *brx_Deva* | - | - | - | - | **45.3** | - | - | - | - | - | - | **56.3** | - | - |
| *doi_Deva* | - | - | - | - | **53.9** | 40.1 | - | - | - | - | - | **65.0** | 62.9 | - |
| *gom_Deva* | - | - | - | - | **42.5** | 40.3 | 38.7 | - | - | - | - | **51.7** | 51.6 | 46.1 |
| *guj_Gujr* | 49.1 | 21.0 | 48.7 | 49.8 | **53.1** | 51.9 | 51.8 | 56.9 | 6.5 | 60.8 | 61.4 | **62.0** | 62.2 | 61.1 |
| *hin_Deva* | 48.6 | 42.7 | 47.6 | 48.3 | 49.6 | **50.6** | 48.7 | 57.4 | 50.6 | 58.7 | 59.7 | **60.1** | 60.0 | 59.3 |
| *kan_Knda* | 32.6 | 13.7 | 32.2 | 33.3 | **33.8** | 33.1 | 33.5 | 44.0 | 7.2 | 45.3 | 46.2 | 47.5 | 48.0 | **48.1** |
| *kas_Arab* | - | - | 25.7 | 27.1 | **35.6** | - | - | - | - | 44.6 | 45.2 | **52.6** | - | - |
| *mai_Deva* | - | - | 41.6 | 41.0 | 44.3 | 35.6 | 38.2 | - | - | 55.2 | 56.7 | 57.8 | **59.1** | 55.8 |
| *mal_Mlym* | 43.8 | 32.0 | 40.9 | 40.8 | **45.7** | 45.2 | 44.9 | 50.6 | 38.8 | 51.0 | 52.6 | 54.3 | **54.6** | 54.4 |
| *mar_Deva* | 43.7 | 33.9 | 44.8 | 47.3 | **48.6** | 46.6 | 46.3 | 54.2 | 40.4 | 56.2 | 57.5 | 58.5 | **59.4** | 58.3 |
| *mni_Mtei* | - | - | - | - | **40.2** | 31.2 | - | - | - | - | - | **52.5** | 46.3 | - |
| *npi_Deva* | - | 15.3 | 44.9 | 44.3 | **51.5** | 46.1 | 46.4 | - | 21.0 | 59.9 | 60.6 | 63.0 | **63.9** | 62.0 |
| *ory_Orya* | 38.9 | 7.6 | 41.3 | 40.9 | 40.2 | 37.7 | **42.1** | 55.6 | 11.5 | 59.3 | 59.8 | **60.3** | 59.0 | 58.7 |
| *pan_Guru* | 54.0 | 25.4 | 54.3 | 55.5 | 57.8 | **61.1** | 56.8 | 58.1 | 32.4 | 60.1 | 61.4 | **62.7** | 61.1 | 61.1 |
| *san_Deva* | - | - | 26.4 | 30.3 | 35.5 | 32.8 | - | - | - | 38.9 | 40.2 | 48.3 | **49.2** | - |
| *sat_Olck* | - | - | 0.8 | 18.0 | **34.6** | - | - | - | - | 33.6 | 37.4 | **43.5** | - | - |
| *snd_Deva* | - | - | - | - | **30.3** | - | - | - | - | - | - | **49.6** | - | - |
| *tam_Taml* | 37.7 | 19.2 | 37.2 | 37.1 | **39.1** | 38.7 | **39.1** | 44.1 | 22.5 | 45.7 | 46.8 | 45.8 | **46.8** | 46.4 |
| *tel_Telu* | 42.5 | - | 39.9 | 40.5 | **45.5** | 44.6 | 44.9 | 48.5 | - | 51.3 | 53.3 | 52.9 | **53.9** | 53.6 |
| *urd_Arab* | - | 42.5 | 55.9 | 55.5 | **61.6** | 60.6 | 59.6 | - | 47.9 | 61.5 | 62.3 | **65.5** | 65.3 | 64.9 |
| *Avg.* | 43.2 | 26.6 | 39.5 | 41.3 | 44.8 | 43.8 | 45.8 | 52.5 | 29.7 | 52.7 | 54.0 | 56.0 | 57.1 | 56.7 |
| Δ | 3.2 | 21.6 | 5.7 | 3.9 | - | 2.8 | 1.5 | 4.4 | 28.3 | 3.3 | 2.0 | - | 0.1 | 0.9 |

presented in Table 14. Across the board, the results show moderately strong translation quality by all the models. Overall, a similar trend is observed for En-Indic translations, with IndicTrans2 outperforming the best open-source models and commercial models. Similarly, in the case of Indic-En translations, IndicTrans2 outperforms the best open-source models and performs competitively with commercial models. The results further highlight significant improvements in the quality of translations for low-resource languages such as Dogri (+13.8), Kashmiri (+8.5), Manipuri Meitei (+9), Sanskrit (+2.7), and Santali (+16.6) in the En-Indic direction and Kashmiri (+7.4), and Santali (+6.1) in the Indic-En direction respectively, compared to the best available existing systems. Given that IndicTrans2 supports all 22 scheduled languages and performs well across all of them, the model is expected to have good usability in both informational and conversational settings. Additionally, we also report the COMET (Rei et al., 2022) and BLEU (Papineni et al., 2002) scores for our models in the Table 40 and Table 43 (in Appendix B).

**Evaluation on Other Benchmarks.** We perform evaluations on other publicly available benchmarks and the detailed results are presented in Appendix B, while a summary of the observations is presented in this section. Specifically, we evaluate the models on WAT 2020 (Nakazawa et al., 2020) and WAT2021 (Nakazawa et al., 2021a), which were created from the PMIndia corpus containing data from speeches and news from the Prime Minister of India. Across the board, the results presented in Table 30 and Table 31 show that IndicTrans2 outperforms all open-source and commercial models in both Indic-En and En-Indic translation directions, with the exception of IndicTrans1. However, it is important to note that performance improvement for IndicTrans1 stems from the fact that their validation set consisted of the development sets of various shared task benchmarks like WAT, WMT, and FLORES-200. On the contrary, our

work used the FLORES-200 development set as the validation set with the aim of attaining strong performance across multiple domains. Along the same lines, we evaluate our models on the NTREX (Federmann et al., 2022) Evaluation set, which is derived from the news domain. The results presented in Table 27 and Table 28 show similar findings with IndicTrans2 performing the best among all the compared models with +3 and +2.6 points improvement over the best open-source model in En-Indic and Indic-En directions respectively. However, on the UFAL test set involving Tamil language, among open-source models, we observe that our model lags behind the IndicTrans1 and NLLB 1.2B model in the En-Indic direction (Table 36).

**Best Open-Source Model.** Our study evaluated the translation quality of IndicTrans2 and other open-source models on various benchmarks. While IN22 and FLORES-200 (Costa-jussà et al., 2022) evaluated the models on diverse domain content such as sports, news, and conversational texts, we further tested the models on WAT2020 (Nakazawa et al., 2020), WAT2021 (Nakazawa et al., 2021a), and NTREX (Federmann et al., 2022). **Across all multi-domain benchmarks, we observed that IndicTrans2 consistently outperformed other open-source models, demonstrating its better translation capabilities.** However, it is important to note that performance improvement for IndicTrans1 on WAT2020 (Nakazawa et al., 2020) and WAT2021 (Nakazawa et al., 2021a) can be attributed due to explicit optimization across different benchmarks by incorporating development sets of various shared tasks, in addition to FLORES-200. In contrast, our development set only comprises FLORES-200. Detailed results for all the benchmarks and models are presented in Appendix B (refer Tables 27, 30 and 31). Additionally, IndicTrans2 has the highest coverage of languages and written scripts, with support for 22 Indic languages and 25 language-script combinations. Further, while the current SOTA open-source model, the NLLB 54B MoE model (Costa-jussà et al., 2022), is impressive in its capabilities, it is impractical for deployment due to its high latency and resource requirements. Our study addresses this challenge by **developing comparatively compact models that can compete with large-scale models even when trained on smaller datasets, emphasizing quality and cost-effectiveness.** Results on different benchmarks confirm the robust performance of our model across various domains and distributions. Therefore, we can conclude that our model has fair generalization capabilities, performing well across most of the benchmarks.

**Supporting New Languages and Scripts.** Our work bridges the gap left by existing open-source and commercial systems by extending IndicTrans1 (Ramesh et al., 2022) to support all 22 scheduled Indic languages, including low-resource languages and multiple scripts. We train the first open-source model with reasonable performance for the following languages: Bodo, Dogri, and Konkani. For some languages, we support translation in scripts that were hitherto unsupported like Sindhi (Devanagari script) or are only supported by commercial systems like Manipuri (Meitei). In addition, we also improve translation quality significantly for low-resource languages such as Dogri, Maithili, Manipuri (Meitei), and Nepali. The human-annotated seed parallel data (refer Table 1) for these languages help us outperform other models which rely on unsupervised methods and/or mined data for these low-resource languages. This suggests that investments in creating small parallel corpora for low-resource languages can substantially improve translation quality, corroborating findings from Costa-jussà et al. (2022).

**Comparison across language families.** Our analysis reveals that on low-resource languages from the Sino-Tibetan and Austroasiatic language families models tend to consistently underperform compared to mid and high-resource languages in the Indo-Aryan and Dravidian families. Conversely, on mid and high-resource languages, all models seem to exhibit comparable performance. These observations suggest that the major differences in performance are coming from the low-resource language families. Notably, no other open-source or commercial model covers all four language families. The results for all the models on our primary benchmarks are presented in Figure 5.

Additionally, we conduct a small-scale human evaluation exercise to verify if the quality of our model outputs correlates with the improvements observed using automatic metrics. This preliminary human evaluation exercise focused on the En-Indic direction and included 50 examples each from the Wikipedia and Web sources subset to yield a total of 100 sentence pairs from IN22-Gen and is described in Appendix C. However, future efforts should focus on large-scale human evaluation to understand the potential biases and shortcomings of our IndicTrans2 models and assess their feasibility in practical use-case scenarios.

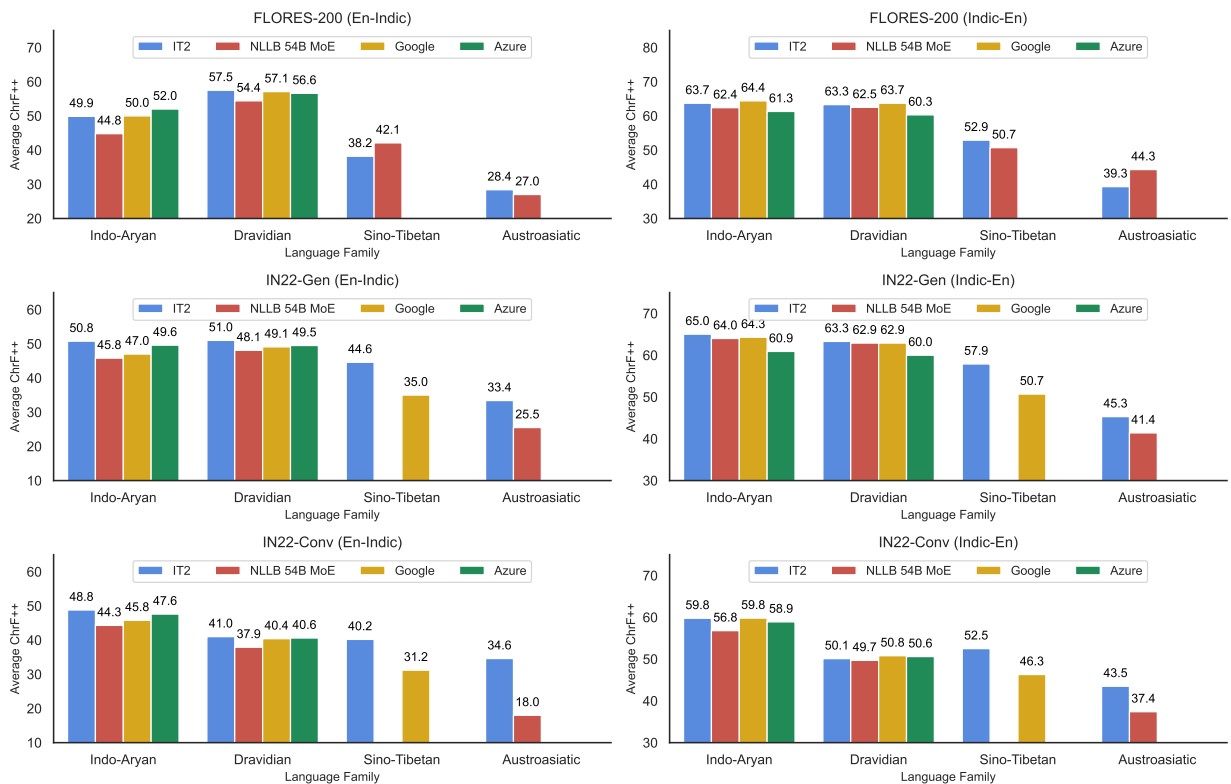

Figure 5: Average performance improvements in terms of chrF++ across language families on IN22 and FLORES-200 (Costa-jussà et al., 2022) benchmarks.

## 7.2 Understanding Data Scale vs Quality tradeoff

Prior works such as NLLB (Costa-jussà et al., 2022) have focused on scaling the data to improve the model performance. They use a margin-based mining approach with a threshold of 1.06. However, from an in-house manual inspection, it was observed that the data was noisy. As a result, we conducted an ablation study to understand the trade-off between data scale and quality for effectively training multilingual MT models. In this ablation, we consider existing mined parallel corpora such as Samanantar (Ramesh et al., 2022) and NLLB (Costa-jussà et al., 2022) and specifically focus on the subset of 11 languages that are common to both. We apply an additional quality filter, where we eliminate the bitext pairs that fall below the LABSE (Feng et al., 2022) cosine similarity threshold of 0.80. This resulted in a reduction from 384M (Unfiltered data) to 94M (filtered data) in total. Subsequently, we train two separate models with the same architecture (refer to Section 5.4) and stage 1 hyperparameters (refer to Table 10) as our final IndicTrans2 models on filtered and unfiltered versions of the data. The results shown in Table 15 demonstrate that the models trained on the high-quality filtered subset perform on par or even superior to the model trained on the unfiltered data. This suggests that **eliminating the noisy and suboptimal bitext pairs through this additional filter improves the model performance and accelerates model convergence.** We, therefore, adopt this filtering threshold for our final training, ensuring that our model benefits from the improved data quality.

## 7.3 Impact of Sequential Training with Human Annotated Data

We train our models sequentially, where stage 1 involves training on a combination of all the existing data, mined data, and high-quality seed data, while stage 2 involves fine-tuning with high-quality seed data (as described in Section 5.5). Our seed data involves a combination of NLLB Seed (Costa-jussà et al., 2022; Maillard et al., 2023) and our human-annotated data BPCC-H-Wiki (refer Table 1). As seed data for Sindhi (Arabic) is not present in both the sources, we

Table 15: chrF++ scores of the models trained on unfiltered (pre-filtering) and filtered data (post-filtering), on the FLORES-200 Evaluation set in the En-Indic and Indic-En directions. The best-performing system is bolded. $\Delta$ represents the difference between the scores of the model trained on filtered data and unfiltered data. A positive value for $\Delta$ indicates that the model trained on filtered data (post-filtering) is better than unfiltered (pre-filtering) and vice-versa.

| | Dataset Size | | En-Indic | | | Indic-En | | |
| language | Pre-Filter | Post-Filter | Pre-Filter | Post-Filter | $\Delta$ | Pre-Filter | Post-Filter | $\Delta$ |
|---|---|---|---|---|---|---|---|---|
| asm_Beng | 5.3M | 0.5M | 34.6 | **39.0** | 4.4 | 49.2 | **51.9** | 2.7 |
| ben_Beng | 70.4M | 16.5M | 52.2 | **53.1** | 0.9 | 60.0 | **60.2** | 0.2 |
| guj_Gujr | 14.4M | 8.4M | 51.4 | **52.4** | 1.0 | **64.0** | 63.9 | -0.1 |
| hin_Deva | 43.1M | 11M | 58.1 | **58.7** | 0.6 | 64.4 | **64.6** | 0.2 |
| kan_Knda | 38.3M | 10.5M | 52.7 | **53.3** | 0.6 | 58.6 | **58.7** | 0.1 |
| mal_Mlym | 49.6M | 10.8M | 52.8 | **55.1** | 2.3 | 60.2 | **61.1** | 0.9 |
| mar_Deva | 35.6M | 7.74M | 46.9 | **48.5** | 1.6 | 60.6 | **60.7** | 0.1 |
| ory_Orya | 14.7M | 2.9M | 42.6 | **46.1** | 3.5 | 58.8 | **60.0** | 1.2 |
| pan_Guru | 14M | 3.3M | 49.1 | **50.6** | 1.5 | 62.7 | **63.1** | 0.4 |
| tam_Taml | 47.7M | 10.4M | 53.3 | **55.3** | 2.0 | 58.0 | **58.2** | 0.2 |
| tel_Telu | 51.2M | 11.8M | 56.0 | **56.8** | 0.8 | 63.0 | **63.2** | 0.2 |
| Avg. | - | - | 50.0 | **51.7** | 1.7 | 60.0 | **60.5** | 0.6 |

Table 16: Performance improvements of En-Indic and Indic-En models on chrF++ metric on our primary evaluation benchmarks w.r.t. sequential training.

| Benchmark | En-Indic | Indic-En |
|---|---|---|
| FLORES-200 | +1.5 | +0.6 |
| IN22-Gen | +2.2 | +0.5 |
| IN22-Conv | +2.7 | +1.9 |
| Average | +2.1 | +1.0 |

use the Sangam transliteration API[33] (Lehal & Saini, 2014) to transliterate the Sindhi BPCC-H-Wiki data (~10.5K) from Devanagari script to Perso-Arabic script. We observe that **fine-tuning our models with high-quality seed data is beneficial** and leads to an average improvement of 2.1 points and 1 point in En-Indic and Indic-En directions, respectively, on our primary evaluation benchmarks in terms of chrF++ metric (see Table 16). These findings align with previous works (Mohiuddin et al., 2022), which show that deterministic data selection curriculum involves pretraining on general domain corpora followed by fine-tuning with high-quality data subset of general domain corpora results in solid performance improvements over the preliminary models. A critical distinction from the above approach is that we only use the human-annotated seed data for fine-tuning, rather than retrieval of top $p\%$ samples from training data based on lexical similarity. Our observations indicate that although sequential training yields gains on an aggregate level, it is important to note that for specific languages such as Sindhi (Arabic) (where we use transliterated data), our En-Indic model tends to degrade (~3 points in chrF++) in terms of performance, highlighting that it is crucial to use high-quality human annotated data for fine-tuning.

Furthermore, Table 17 reports the performance of IndicTrans2 models for various training stages on IN22-Gen Set. Notably, the highest improvement was observed in Santali for the En-Indic direction in both $\Delta_1$ and $\Delta_2$. It is also worth highlighting that the human-annotated seed data from previous work and our current work serves as the primary and most influential source for mid-resource and low-resource languages, including Dogri, Konkani, Sindhi (Devanagari), Santali, and Manipuri (Meitei) as shown in Table 1. Despite the smaller size of seed data compared to mined corpora, finetuning on this leads to superior performance across different benchmarks (refer Tables 12 to 14). Although $\Delta_1$ and $\Delta_2$ may be smaller for a few languages due to the saturation of the data diversity during multi-stage training, the seed data proves to be beneficial on an aggregate level, further reinforcing its positive impact.

---

[33]https://sangam.learnpunjabi.org/

Table 17: chrF++ score on IN22-Gen Evaluation Set for various training stages. OG refers to the model trained on the original training corpora, while OG-Seed refers to the seed data fine-tuned version of the OG model. $\Delta_1$ represents the gains obtained by fine-tuning the original model with seed data. DA refers to the model trained on the combination of original training data with augmented data, while DA-Seed refers to the seed data fine-tuned version of the DA model. $\Delta_2$ represents the gains obtained by fine-tuning on seed data after data augmentation.

| | En-Indic | | | | | | Indic-En | | | | | |
| language | OG | OG-Seed | $\Delta_1$ | DA | DA+Seed | $\Delta_2$ | OG | OG-Seed | $\Delta_1$ | DA | DA+Seed | $\Delta_2$ |
|---|---|---|---|---|---|---|---|---|---|---|---|---|
| asm_Beng | 43.4 | 45.6 | 2.2 | 44.8 | **47.1** | 2.3 | 61.9 | 62.1 | 0.2 | 64.9 | **65.8** | 0.9 |
| ben_Beng | 48.2 | 50.3 | 2.1 | 48.8 | **51.8** | 3.0 | 60.6 | 60.8 | 0.2 | 62.4 | **63.2** | 0.8 |
| brx_Deva | 44.5 | 47.1 | 2.6 | 46.3 | **47.8** | 1.5 | 58.1 | 58.4 | 0.3 | 61.9 | **62.1** | 0.2 |
| doi_Deva | 55.4 | 55.7 | 0.3 | 56.2 | **57.8** | 1.6 | 68.6 | 68.5 | -0.1 | 72.7 | **72.6** | -0.1 |
| gom_Deva | 42.2 | 43.8 | 1.6 | 43.2 | **45.2** | 2.0 | 55.9 | 56.5 | 0.6 | 58.7 | **59.2** | 0.5 |
| guj_Gujr | 49.4 | 51.6 | 2.2 | 50.0 | **53.5** | 3.5 | 64.0 | 63.9 | -0.1 | 65.7 | **66.5** | 0.8 |
| hin_Deva | 53.5 | 54.6 | 1.1 | 53.6 | **56.7** | 3.1 | 62.8 | 63.4 | 0.6 | 64.7 | **65.4** | 0.7 |
| kan_Knda | 47.3 | 49.7 | 2.4 | 47.7 | **51.0** | 3.3 | 61.7 | 62.0 | 0.3 | 63.2 | **64.2** | 1.0 |
| kas_Arab | 37.7 | 38.8 | 1.1 | 38.3 | **40.2** | 1.9 | 55.6 | 56.1 | 0.5 | 60.0 | **60.4** | 0.4 |
| mai_Deva | 45.9 | 47.3 | 1.4 | 46.2 | **48.7** | 2.5 | 62.1 | 61.9 | -0.2 | 64.6 | **64.8** | 0.2 |
| mal_Mlym | 47.9 | 49.7 | 1.8 | 48.4 | **50.9** | 2.5 | 60.7 | 61.5 | 0.8 | 63.1 | **64.5** | 1.4 |
| mar_Deva | 45.7 | 48.6 | 2.9 | 46.6 | **51.0** | 4.4 | 60.7 | 61.1 | 0.4 | 62.3 | **63.7** | 1.4 |
| mni_Mtei | 39.6 | 41.3 | 1.7 | 41.8 | **44.6** | 2.8 | 53.2 | 53.3 | 0.1 | 57.6 | **57.9** | 0.3 |
| npi_Deva | 44.5 | 47.5 | 3.0 | 45.4 | **49.0** | 3.6 | 64.4 | 64.4 | 0.0 | 67.1 | **67.7** | 0.6 |
| ory_Orya | 40.1 | 41.9 | 1.8 | 41.0 | **43.9** | 2.9 | 63.1 | 63.4 | 0.3 | 65.3 | **66.2** | 0.9 |
| pan_Guru | 49.5 | **50.6** | 1.1 | 50.2 | **50.6** | 0.4 | 61.0 | 61.4 | 0.4 | 62.9 | **63.4** | 0.5 |
| san_Deva | 35.9 | 37.7 | 1.8 | 36.9 | **38.8** | 1.9 | 50.9 | 51.1 | 0.2 | 54.4 | **54.8** | 0.4 |
| sat_Olck | 24.2 | 27.3 | 3.1 | 26.5 | **33.4** | 6.9 | 43.6 | 43.8 | 0.2 | 44.5 | **45.3** | 0.8 |
| snd_Deva | 34.8 | 36.2 | 1.4 | 35.3 | **36.6** | 1.3 | 53.6 | 53.7 | 0.1 | 56.5 | **57.3** | 0.8 |
| tam_Taml | 47.3 | 48.7 | 1.4 | 47.9 | **49.5** | 1.6 | 57.2 | 57.5 | 0.3 | 59.1 | **59.8** | 0.7 |
| tel_Telu | 49.6 | 51.3 | 1.7 | 50.0 | **52.4** | 2.4 | 62.3 | 62.6 | 0.3 | 64.0 | **64.8** | 0.8 |
| urd_Arab | 63.8 | 67.1 | 3.3 | 65.4 | **68.2** | 2.8 | 69.5 | 69.9 | 0.4 | 72.5 | **73.0** | 0.5 |

Table 18: Comparison of average chrF++ scores between our stage 2 auxiliary model and the best open-source baseline on FLORES-200 (Costa-jussà et al., 2022) Evaluation set at the end of stage 2 auxiliary training. OG-seed denotes the model trained on the original data followed by fine-tuning with seed data. $\Delta$ denotes the difference between the scores of our stage 2 auxiliary model and the best open-source baseline.

| | N54 | OG-Seed | $\Delta$ |
|---|---|---|---|
| xx-eng_Latn | 60.9 | 58.1 | -2.8 |
| eng_Latn-xx | 45.7 | 47.8 | 2.1 |

## 7.4 Impact of Data Augmentation

Section 5.6 describes the procedure and heuristics for synthetic data generation to further improve our auxiliary models. Initially, we adopted the back-translation approach for generating the augmented data. We primarily base our decision to start with an auxiliary En-Indic model for generating back-translation data for Indic-En translation due to its competitive or better performance compared to the best open-source baseline (see Table 18). We combine the original data and the English back-translated data, obtained using our auxiliary En-Indic model, to train our new Indic-En model from scratch, followed by high-quality seed data fine-tuning. In this case, following prior study (Caswell et al., 2019), we use "__bt__" indicator tags to provide some supervision to the model to distinguish original data from the back-translated data. We observe a considerable performance improvement across all our primary evaluation benchmarks on our Indic-En model, as shown in Figure 6 when we perform training on the combination of original and back-translated data (refer Table 17).

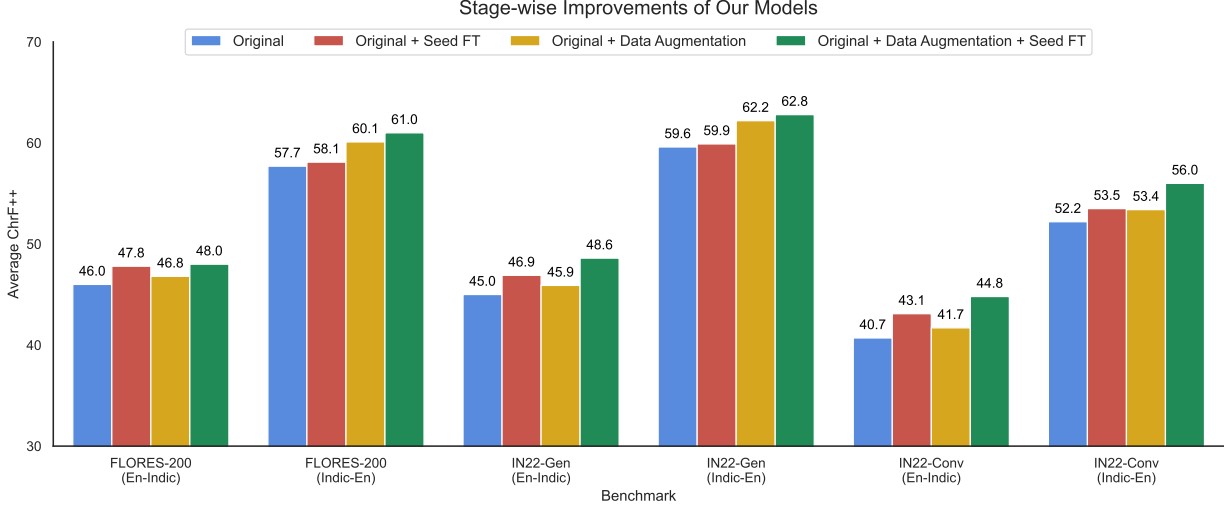

Figure 6: Average Performance of our En-Indic and Indic-En models across different stages in terms of chrF++ metric on our primary evaluation sets.

Following iterative back translation (Hoang et al., 2018), we use the stage 2 fine-tuned downstream Indic-En model to generate the back-translation data due to its superior performance compared to the auxiliary Indic-En model. Similarly, we combine the Indic back-translated data along with the original data using indicator tags and train our new En-Indic model from scratch, followed by fine-tuning with seed data. However, we do not observe any gains for the new En-Indic model compared to the stage 2 auxiliary fine-tuned En-Indic model. Further investigation is needed to determine the exact reasons for the performance limitations of our newly trained En-Indic model, but we suspect that unlike for Indic-En translation, the increase in the Indic target side data is insufficient, both in terms of domain coverage and amount. This conjecture is based on the fact that a significant portion of both the original training corpus and the back-translated data is sourced from the news domain, resulting in considerable overlap in their distributional coverage. The lack of diversity in domains may potentially hinder the model from reaching its optimal capabilities. Furthermore, for Indic-En translation, the amount of target side English data almost triples in amount when back-translated data is added to the original parallel corpus. However, in the case of English-Indic translation, where multiple target languages are involved, the relative augmentation per language is comparatively lower, which might potentially explain the marginal enhancement observed in the English-Indic direction. Increased availability of Indic language monolingual corpora, ideally from various domains, should help remedy this issue.

Since backtranslation did not help in the En-Indic direction, we looked at the findings from distillation works like Kim & Rush (2016); Gumma et al. (2023), and trained an En-Indic model on the combination of original data and forward translated data/distillation data (flipping the English BT data). In this case, we use "`__ft__`" indicator tags instead of "`__bt__`" indicator tags. Here, we observe marginal performance improvements for our newly trained En-Indic model on combining original data and forward translated data, as shown in Figure 6 (refer Table 17). Although this model is not particularly better than the one obtained using back-translation, it does exhibit better performance, and thus we consider this as our final En-Indic model. Overall, our En-Indic model is competitive or better when compared to the baselines, but further research is necessary to explore effective methods to improve the En-Indic model.

## 7.5 Indic-Indic Evaluation

Our IndicTrans2 models have exhibited strong performance across various benchmarks, as detailed in Section 7.1. Building upon these findings, we aim to conduct a comprehensive evaluation of the Indic-Indic translation capabilities of our IndicTrans2 models in both pivot-based and direct setups.

Table 19: chrF++ scores of Indic-Indic evaluation on FLORES-200 (Costa-jussà et al., 2022) of our IndicTrans2-Pivot (IT2-Pivot) model, IndicTrans2-M2M (IT2-M2M) model, compressed IndicTrans2-M2M (IndicTrans2-Dist-M2M) model and NLLB 54B MoE model. "xx-{lang}" and "{lang}-xx" denote the average chrF++ scores to that language and from that language, respectively.

| | xx-{lang} | | | | {lang}-xx | | | |
|---|---|---|---|---|---|---|---|---|
| *language* | N54 | IT2-Pivot | IT2-M2M | IT2-Dist-M2M | N54 | IT2-Pivot | IT2-M2M | IT2-Dist-M2M |
| *asm_Beng* | 36.7 | **38.0** | 37.9 | 37.4 | 39.5 | **41.0** | 39.7 | 39.3 |
| *ben_Beng* | 44.5 | **45.7** | 44.7 | 43.7 | 41.4 | **43.0** | 42.1 | 41.6 |
| *guj_Gujr* | 44.8 | **45.9** | 44.8 | 44.2 | 43.4 | **44.9** | 43.8 | 43.3 |
| *hin_Deva* | 48.4 | **48.6** | 47.7 | 46.8 | 42.9 | **44.6** | 43.8 | 43.6 |
| *kan_Knda* | 46.6 | **47.3** | 45.9 | 45.1 | 40.6 | **42.3** | 41.2 | 40.8 |
| *kas_Arab* | 32.6 | **33.8** | 33.1 | 32.8 | 40.7 | **41.7** | 39.9 | 39.2 |
| *mai_Deva* | 37.9 | **41.5** | 40.5 | 40.4 | 45.0 | **45.9** | 44.9 | 44.7 |
| *mal_Mlym* | 45.7 | **47.8** | 46.2 | 45.1 | 41.2 | **43.3** | 42.0 | 41.5 |
| *mar_Deva* | 41.9 | **43.6** | 42.5 | 41.7 | 42.4 | **44.1** | 43.0 | 42.5 |
| *npi_Deva* | 43.6 | **46.9** | 45.8 | 45.4 | 43.1 | **45.0** | 44.0 | 43.5 |
| *ory_Orya* | 41.1 | **41.6** | 40.8 | 40.2 | 42.7 | **44.3** | 43.3 | 42.8 |
| *pan_Guru* | 44.4 | **44.6** | 43.8 | 43.1 | 43.4 | **44.6** | 43.5 | 43.2 |
| *san_Deva* | 25.6 | **28.9** | 28.7 | 28.6 | 35.7 | **38.1** | 36.5 | 35.9 |
| *sat_Olck* | 25.7 | **26.6** | 26.3 | 26.1 | 32.4 | 31.4 | **32.5** | 31.5 |
| *tam_Taml* | 47.3 | **48.7** | 47.3 | 46.1 | 40.1 | **41.7** | 40.1 | 39.7 |
| *tel_Telu* | 47.0 | **48.5** | 47 | 46 | 41.9 | **43.7** | 42.6 | 41.8 |
| *urd_Arab* | 43.7 | **44.4** | 43.9 | 43.1 | 41.1 | **42.7** | 41.6 | 41 |

### 7.5.1 Pivoting

Pivoting (Gispert & Mariño, 2006; Utiyama & Isahara, 2007; Bertoldi et al., 2008) is a widely used approach in non-English centric translation scenarios, where direct parallel corpora are limited or unavailable. It involves utilizing a high-resource language as an intermediary, translating from the source to the pivot language and then to the target language. The pivot method is a strong baseline for non-English centric translation compared to many other methods proposed to address this task (Freitag & Firat, 2020; Chen et al., 2017; Firat et al., 2016; Arivazhagan et al., 2019; Al-Shedivat & Parikh, 2019). In our study, we leverage our Indic-En model followed by the En-Indic model to facilitate Indic-Indic translation, as our IndicTrans2 models are trained using English-centric parallel corpora and use English as the pivot language. To assess the Indic-Indic translation performance, we evaluate our IndicTrans2 models on $n$-way parallel test sets such as FLORES-200 (Costa-jussà et al., 2022) and IN22 benchmarks. The generation and evaluation procedure for Indic-Indic translations is the same as described in Section 6.4 and Section 6.5.

The performance in Indic-Indic translation for our pivot-based IndicTrans2 and NLLB (Costa-jussà et al., 2022) is shown in Table 19 for FLORES-200, Table 20 for IN22-Gen and Table 21 for IN22-Conv, using average chrF++ scores over common languages across NLLB, our pivot as well as direct systems described in Section 7.5.2. For each language (lang), "xx-{lang}" denotes the average scores from all the common languages in that language, whereas "{lang}-xx" denotes the average scores from that language into all the common languages. Table 19 shows that our pivot-based IndicTrans2 outperforms or is on par with the multi-way trained NLLB 54B MoE model across all Indic-Indic directions on FLORES-200 (Costa-jussà et al., 2022). It is important to note that we directly evaluate the NLLB 54B model by using the translation outputs[34] released by Costa-jussà et al. (2022). However, for the evaluation on the IN22 benchmark, we use the NLLB 1.2B distilled model instead of the NLLB 54B MoE model due to resource constraints due to the sheer number of translation directions. **Our pivot-based IndicTrans2 significantly outperforms the NLLB 1.2B distilled model**, as shown in Tables 20 and 21. NLLB 1.2B distilled model provides a lower-bound estimate of the performance. However, we anticipate a smaller difference between our pivot-based IndicTrans2 and the best NLLB 54B MoE model. Based on our previous results, we expect IndicTrans2 scores to be comparable if not

---

[34]https://tinyurl.com/nllbflorestranslations

Table 20: chrF++ scores of Indic-Indic evaluation on IN22-Gen test set of our IndicTrans2-Pivot (IT2-Pivot) model, IndicTrans2-M2M (IT2-M2M) model, compressed IndicTrans2-M2M (IndicTrans2-Dist-M2M) model and NLLB 1.2B distilled model. "xx-{lang}" and "{lang}-xx" denote the average chrF++ scores to that language and from that language, respectively. † indicates completely off-target translations.

| | xx-{lang} | | | | {lang}-xx | | | |
|---|---|---|---|---|---|---|---|---|
| language | N1.2 | IT2-Pivot | IT2-M2M | IT2-Dist-M2M | N1.2 | IT2-Pivot | IT2-M2M | IT2-Dist-M2M |
| asm_Beng | 35.5 | **40.7** | 40.5 | 39.4 | 38.8 | **44.0** | 42.7 | 40.9 |
| ben_Beng | 39.9 | **45.1** | 44.8 | 43.2 | 37.4 | **43.2** | 42.3 | 41.2 |
| guj_Gujr | 39.2 | **45.4** | 44.3 | 42.9 | 39.0 | **43.8** | 43.2 | 39.9 |
| hin_Deva | 43.7 | **49.2** | 48.8 | 47.1 | 39.1 | **43.4** | 43.0 | 42.3 |
| kan_Knda | 39.4 | **44.6** | 44.5 | 43 | 38.4 | **43.9** | 43.1 | 39.8 |
| kas_Arab | 28.5 | **35.4** | 34.8 | 33.7 | 35.6 | **41.8** | 41.3 | 39.8 |
| mai_Deva | 36.6 | **42.0** | 41.9 | 40.3 | 39.1 | **44.2** | 43.7 | 42.8 |
| mal_Mlym | 38.5 | **44.9** | 43.5 | 42 | 36.4 | **42.9** | 42.2 | 40.6 |
| mar_Deva | 37.6 | **44.4** | 43.6 | 41.5 | 38.2 | **43.8** | 43.0 | 42.4 |
| npi_Deva | 37.3 | **41.4** | 41.1 | 39.6 | 39.0 | **44.8** | 44.0 | 43.1 |
| ory_Orya | 36.1 | **38.2** | 38.0 | 36.8 | 39.4 | **44.9** | 44.3 | 41.3 |
| pan_Guru | 39.0 | **43.2** | 42.2 | 40.9 | 36.8 | **41.7** | 40.5 | 39.1 |
| san_Deva | 23.3 | **35.8** | **35.8** | 34.6 | 32.8 | **39.8** | 39.0 | 37.6 |
| sat_Olck | 0.0† | **31.2** | **31.2** | 30 | 0.0† | 35.0 | **37.2** | 35.8 |
| tam_Taml | 40.1 | **45.0** | 44.1 | 42.6 | 35.4 | **41.3** | 40.2 | 39.3 |
| tel_Telu | 40.0 | **45.7** | 44.5 | 42.9 | 37.5 | **43.2** | 42.5 | 41.9 |
| urd_Arab | 47.7 | **54.6** | 53.2 | 50.8 | 39.4 | **45.2** | 44.4 | 43.6 |

Table 21: chrF++ scores of Indic-Indic evaluation on IN22-Conv test set of our IndicTrans2-Pivot (IT2-Pivot) model, IndicTrans2-M2M (IT2-M2M) model, compressed IndicTrans2-M2M (IndicTrans2-Dist-M2M) model and NLLB 1.2B distilled model. "xx-{lang}" and "{lang}-xx" denote the average chrF++ scores to that language and from that language, respectively. † indicates completely off-target translations.

| | xx-{lang} | | | | {lang}-xx | | | |
|---|---|---|---|---|---|---|---|---|
| language | N1.2 | IT2-Pivot | IT2-M2M | IT2-Dist-M2M | N1.2 | IT2-Pivot | IT2-M2M | IT2-Dist-M2M |
| asm_Beng | 33.7 | **38.6** | 38.6 | 37.7 | 35.8 | **41.1** | 40.6 | 39.9 |
| ben_Beng | 37.6 | **41.6** | 41.5 | 40.5 | 34.8 | **39.8** | 39.8 | 39.1 |
| guj_Gujr | 38.1 | **43.5** | 43.1 | 42.1 | 36.5 | **40.9** | 40.5 | 39.7 |
| hin_Deva | 39.9 | **42.5** | 42.4 | 41.7 | 36.3 | **40.5** | 40.4 | 39.9 |
| kan_Knda | 28.2 | **30.8** | 30.7 | 30.1 | 30.8 | **35.5** | 34.6 | 33.7 |
| kas_Arab | 18.6 | 30.7 | **31.1** | 30.7 | 30.5 | **37.4** | 37.4 | 35.7 |
| mai_Deva | 32.2 | 37.9 | **38.4** | 37.8 | 34.8 | **40.0** | 39.6 | 38.9 |
| mal_Mlym | 34.9 | **39.7** | 39.0 | 38.0 | 32.9 | **37.7** | 37.3 | 36.4 |
| mar_Deva | 35.6 | **41.0** | 40.4 | 39.2 | 35.7 | **40.0** | 39.9 | 39.4 |
| npi_Deva | 35.6 | **42.2** | 42.0 | 41.2 | 36.2 | **41.1** | 40.9 | 40.1 |
| ory_Orya | 33.7 | 34.4 | **34.5** | 33.9 | 36.2 | **41.2** | 40.7 | 39.9 |
| pan_Guru | 40.9 | **45.5** | 45 | 44.0 | 35.6 | **40.3** | 39.8 | 39.2 |
| san_Deva | 22.3 | 31.8 | **32** | 31.5 | 26.8 | **34.8** | 34.4 | 33.1 |
| sat_Olck | 0.0† | 30.7 | **31.2** | 30.4 | 0.0† | 32.1 | **34.7** | 33.8 |
| tam_Taml | 33.2 | **36.2** | 35.6 | 34.9 | 30.7 | **34.3** | 34.0 | 33.3 |
| tel_Telu | 35.0 | **39.6** | 39.1 | 37.9 | 33.2 | **37.5** | 37.3 | 36.6 |
| urd_Arab | 43.7 | **49.2** | 48.8 | 47.9 | 36.5 | **41.7** | 41.4 | 40.7 |

better than the best NLLB 54B MoE model. This highlights the effectiveness of our robust English-centric models and their potential in Indic-Indic translation scenarios.

### 7.5.2 Direct Models

While the pivot-based solution demonstrates strong Indic-Indic performance, its inherent sequential dual model pipeline results in increasing the inference time by a factor of 2 compared to the English-centric model. To address this limitation, it is essential to build direct Indic-Indic (IndicTrans2-M2M) models that facilitate Indic-Indic translation with nearly the same inference cost as English-centric model. However, the scarcity of Indic-Indic data makes training such models from scratch challenging. As a result, inspired by prior works (Kim et al., 2019; Ma et al., 2020), we leverage pre-trained components from our English-centric models to initialize the IndicTrans2-M2M model. Specifically, we initialize the IndicTrans2-M2M model using the Encoder from the Indic-En model and the Decoder from the En-Indic model. It is important to note that these two pre-trained components undergo independent training and lack synchronization, resulting in a lack of zero-shot performance post-initialization. Nevertheless, these pre-trained components serve as strong initializations to start with and can be further adapted with limited data.

The BPCC-Wiki subset contains 9.2M bitext pairs spanning 462 Indic-Indic directions. This seed corpus is not completely n-way in the current form (see Section 3.3), and the data scales might be extremely low for some language pairs. As a result, we leverage data augmentation to synthetically generate $n$-way parallel corpora just by performing $n$ inferences instead of $^nC_2$. Specifically, we use our IndicTrans2 En-Indic model to generate 100K synthetic bitext pairs for each translation direction by selecting 100K English monolingual sentences from IndicCorpv2 (Doddapaneni et al., 2023). This amounts to a total of 46.2M pairs across 462 Indic-Indic language pairs. Our fine-tuning dataset for adapting the IndicTrans2-M2M model consists of seed corpus and synthetic corpus, resulting in a total of 55.4M bitext pairs across 462 directions. It is important to note that our IndicTrans2-M2M model covers all 22 scheduled languages but lacks direct support for script variants like Kashmiri (Devanagari), Manipuri (Bengali), and Sindhi (Arabic) due to the unavailability of seed data for these scripts. Tables 19 to 21 shows that our IndicTrans2-M2M achieves competitive performance with a 1-point decrease in the chrF++ metric compared to the pivot-based approach at half of the inference cost. Furthermore, we also apply the same recipe to IndicTrans2-Dist (described in Section 7.6) to improve the inference latency and compress it to about 350M parameters while achieving competitive performance with the IndicTrans2-M2M 1.2B parameter model (see Tables 19 to 21).

### 7.6 Distilled Models

We distill our IndicTrans2 (1.1B parameters, 12Gb size) models into smaller, efficient counterparts called IndicTrans2-Dist (211M parameters, 2Gb size) to enhance deployment feasibility in low-infrastructure settings. Following the *deep and thin* architecture approach (Gumma et al., 2023), we retain the encoder-decoder layer count but reduce other fully-connected dimensions. Acknowledging the robustness of our teacher model, we leverage a smaller, representative dataset subset of ~110 million pairs across all 22 languages for a more data-efficient distillation process. We adopt Word-Level distillation (Hinton et al., 2015; Kim & Rush, 2016), facilitating direct student model training without a separate distilled dataset. The student model is initially distilled from IndicTrans2 and subsequently fine-tuned using the BPCC seed data. Tables 47 to 49 in Appendix D list the hyperparameters and architecture of IndicTrans2-Dist models.

In adherence to metrics used before, we report chrF++ scores of the distilled models on IN22-Gen in Table 22. The chrF++ scores on FLORES-200 and IN22-Conv are presented in Tables 50 and 51 in Appendix D respectively. In contrast to our earlier findings, we find that fine-tuning with seed data was not so beneficial for the distilled models. **Our distilled models trained with Word-Level distillation perform competitively with our best IT2 models and show an average drop of 0.87 on Indic-En and 0.17 on En-Indic across all three benchmarks.** It is important to note that we do not use any backtranslation data for distillation. Notably, we observe higher gains due to distillation on the IN22-Conv than on the IN22-Gen and FLORES-200 in the Indic-En direction. Low-resource languages like Dogri, Bodo and Arabic script languages like Kashmiri and Urdu face a drop of more than 2.5 chrF++ points in the Indic-En direction, whereas Santali has a gain of 2.7 points in IN22-Gen and 2.8 points in IN22-Conv as compared to the Indic-En teacher model. Almost all high-resource languages like Hindi and Bengali observe a negligible reduction in performance with

Table 22: chrF++ scores of Indic-En and En-Indic distilled models on IN22-Gen. Distilled (*Dist*) is the model trained with Word-level KD. $\Delta$ is the difference between the distilled Model fine-tuned on seed data (*Dist-Seed*) & IT2. Higher values of $\Delta$ are preferable.

| | Indic-En | | | | En-Indic | | | |
| *language* | IT2 | Dist | Dist-Seed | $\Delta$ | IT2 | Dist | Dist-Seed | $\Delta$ |
|---|---|---|---|---|---|---|---|---|
| *asm_Beng* | 65.8 | 65.6 | 65.6 | -0.2 | 47.1 | 46.4 | 47.1 | 0.0 |
| *ben_Beng* | 63.2 | 63.1 | 63.3 | 0.1 | 51.8 | 51.5 | 51.6 | -0.2 |
| *brx_Deva* | 62.1 | 59.3 | 59.3 | -2.8 | 47.8 | 47.6 | 47.7 | -0.1 |
| *doi_Deva* | 72.6 | 70.2 | 70.2 | -2.4 | 57.8 | 56.3 | 56.8 | -1.0 |
| *gom_Deva* | 59.2 | 57.3 | 57.2 | -2.0 | 45.2 | 44.5 | 44.8 | -0.4 |
| *guj_Gujr* | 66.5 | 65.5 | 65.5 | -1.0 | 53.5 | 52.9 | 53.2 | -0.3 |
| *hin_Deva* | 65.4 | 63.7 | 63.8 | -1.6 | 56.7 | 56.4 | 56.7 | 0.0 |
| *kan_Knda* | 64.2 | 64.3 | 64.3 | 0.1 | 51.0 | 50.4 | 50.9 | -0.1 |
| *kas_Arab* | 60.4 | 57.6 | 57.8 | -2.6 | 40.2 | 39.0 | 39.5 | -0.7 |
| *mai_Deva* | 64.8 | 64.4 | 64.4 | -0.4 | 48.7 | 48.5 | 48.7 | 0.0 |
| *mal_Mlym* | 64.5 | 63.2 | 63.3 | -1.2 | 50.9 | 50.4 | 50.8 | -0.1 |
| *mar_Deva* | 63.7 | 63.2 | 63.3 | -0.4 | 51.0 | 50.4 | 50.6 | -0.4 |
| *mni_Mtei* | 57.9 | 58.0 | 58.0 | 0.1 | 44.6 | 43.2 | 43.6 | -1.0 |
| *npi_Deva* | 67.7 | 67.6 | 67.5 | -0.2 | 49.0 | 48.7 | 49.0 | 0.0 |
| *ory_Orya* | 66.2 | 65.8 | 65.9 | -0.3 | 43.9 | 43.5 | 43.9 | 0.0 |
| *pan_Guru* | 63.4 | 62.0 | 61.9 | -1.5 | 50.6 | 50.6 | 50.4 | -0.2 |
| *san_Deva* | 54.8 | 53.8 | 53.9 | -0.9 | 38.8 | 37.9 | 38.2 | -0.6 |
| *sat_Olck* | 45.3 | 47.5 | 48.0 | 2.7 | 33.4 | 33.0 | 33.8 | 0.4 |
| *snd_Deva* | 57.3 | 56.0 | 56.6 | -0.7 | 36.6 | 36.6 | 36.6 | 0.0 |
| *tam_Taml* | 59.8 | 58.4 | 58.4 | -1.4 | 49.5 | 49.3 | 49.3 | -0.2 |
| *tel_Telu* | 64.8 | 63.0 | 63.0 | -1.8 | 52.4 | 52.4 | 52.4 | 0.0 |
| *urd_Arab* | 73.0 | 70.8 | 70.9 | -2.1 | 68.2 | 67.8 | 67.8 | -0.4 |
| *Average* | 62.8 | 61.8 | 61.9 | -0.9 | 48.6 | 48.1 | 48.3 | -0.3 |

distillation. In contrast to the findings of Gumma et al. (2023), we observe that the most significant factor is a robust teacher model coupled with high-quality, diverse data to develop compact student models that are comparable to the teacher. However, extensive experiments are needed to further validate and strengthen these observations in the future.

## 8 Conclusion

In this paper, we presented our efforts on building machine translation systems supporting all 22 languages in the $8^{th}$ schedule of the Constitution of India. We created the multi-domain IN22 benchmark and the BPCC parallel corpus, both of which are first-of-their-kind evaluation and training corpora, the latter consisting of ~230M bitext pairs, covering 22 Indic languages. We trained and evaluated robust English-centric models containing 1.1B parameters as well as their compact versions with 211M parameters, which can be used in compute-heavy as well as compute-scarce settings. Additionally, we repurpose pre-trained components from our English-centric models for efficient training of a direct Indic-Indic model containing 1.2B parameters as well as its compact version with 350M parameters. Our evaluations focus on multiple automatic metrics such as BLEU, chrF++ (primary), and COMET which show that our models are comparable, if not better, than publicly available open and commercial systems.

To summarize, our contributions comprehensively cover all three axes for translation systems, namely models, data, and benchmarks. We will open-source the data, benchmarks, and model artifacts publicly and hope that our work will serve as a foundation as well as a guide for further advancements in translation systems for Indic as well as low-resource languages.

# 9 Limitations and Future Work

Our work has several significant positive outcomes, including the release of the first open-source model that is competitive with commercial models and supports all 22 scheduled Indian languages. However, some limitations open up avenues for future research across each of the following axes: Data, Models, Benchmark, Evaluation, and Deployment.

**Data.** One of the foremost challenges is the scarcity of high-quality human-annotated data for mid-resource or low-resource languages, making it difficult to develop robust models on these languages. Furthermore, the limited availability of content in these languages on the web prevents the use of mining-based approaches to overcome data scarcity effectively. As a result, our IndicTrans2 models demonstrate limited generalization capabilities for languages such as Manipuri (Meitei), Santali, and Sindhi (Devnagari). Another important concern is the limited effectiveness of existing sentence embedding models when applied to Indic languages, which can lead to noisy and suboptimal pairs. To address these challenges, it is crucial to calibrate sentence embedding models using human-annotated data to improve their correlation with human annotations. Moreover, expanding the language coverage of these sentence embedding models to encompass all 22 scheduled languages will be pivotal in facilitating mining efforts for mid-resource or low-resource languages.

**Modeling.** Our current work serves as an initial effort to develop IndicTrans2 models supporting 22 scheduled Indic languages, including low-resource ones. Although consistently outperforming baseline systems, a performance gap exists between low-resource and high-resource languages (as shown in Section 7.1). To bridge this gap, we need to explore effective methods to leverage language relatedness for cross-lingual transfer and improve generalization in low-resource settings. Furthermore, while our IndicTrans2 models released with this work prioritize general-purpose use cases, it is equally important to investigate sparse parameter-efficient approaches for effective domain adaptation while also preserving the model's general-purpose utility. Furthermore, our current IndicTrans2 supports translations across 22 scheduled Indic languages, encompassing multiple scripts that cater to a vast majority of Indian speakers. However, numerous Indic languages remain unincorporated, and exploring techniques to extend the current models without catastrophic forgetting is an important research direction.

**Benchmark.** Accurate evaluation of translation models requires original test sets that encompass a wide range of linguistic phenomena and translation challenges. The current test sets that are released are $n$-way constructed with English as the original language, which is a common approach for including numerous languages. This implies that when we evaluate Indic to English translation on benchmarks like FLORES-200 or IN22, our source is translationese instead of original. Prior research has emphasized the importance of utilizing source-original test sets to get a fair evaluation of translation performance (Zhang & Toral, 2019; Federmann et al., 2022). Moreover, the development of an Indic original benchmark would provide an additional aspect for assessing whether the subtleties of Indic language original sentences are accurately captured in English translations. Therefore, we are currently working towards creating Indic-original benchmarks to facilitate the fair evaluation of Indic-En translations. Soon, we intend to release Indic-original to English translation benchmarks for all 22 scheduled Indic languages.

**Evaluation.** Evaluation of translation models is critical for understanding their strengths and weaknesses and guiding further improvements. This evaluation typically involves two main approaches: human evaluation and automatic evaluation. Our current work includes a preliminary human evaluation study on a sample of 100 sentences from our IN22-Gen benchmark for En-Indic translations. However, future efforts should focus on conducting a broad and large-scale human evaluation study that focuses on the free-form evaluation and task-oriented contexts to understand the potential biases and shortcomings of our IndicTrans2 models and assess their feasibility in practical use-case scenarios, thereby identifying areas for improvement. Additionally, developing better automatic evaluation metrics, particularly suited for Indic languages, is vital for achieving a more comprehensive and quantitative assessment of translation quality and facilitating model improvements. Current model-based metrics may not fully support certain languages, emphasizing the need to explore effective ways to calibrate them for Indic languages and improve the correlation with human judgments.

**Fairness.** Our IndicTrans2 models are trained on extensive data collected from the web, which may introduce social biases. To ensure broader and safer accessibility, it is crucial to thoroughly identify and address these biases. Prior works demonstrate that distilled models can further propagate or amplify biases from the teacher model (Ahn et al., 2022; Gupta et al., 2022; Dhar et al., 2021), underscoring the importance of conducting a comprehensive study and developing alignment methods to mitigate such biases.

## 10  Author Contributions

This project is a large team effort, with immense contributions from all the people involved. To list down the contributions of the authors, we document the areas and list the authors contributing significantly to each of these areas. In each area, the contributors are listed sorted by last name. The lead authors, Jay Gala, and Pranjal A. Chitale, have contributed across multiple areas and co-ordinated many activities.

**Parallel Corpus Collection and Mining:** Raghavan AK, Jay Gala, and Aswanth Kumar.

**Human Translation:** Pranjal A. Chitale, Jay Gala, Mitesh M. Khapra, Pratyush Kumar, Anoop Kunchukuttan, Janki Nawale, and Anupama Sujatha.

**Model Training:** Pranjal A. Chitale, Raj Dabre, Jay Gala, and Varun Gumma.

**Distillation:** Pranjal A. Chitale, Raj Dabre, Jay Gala, and Varun Gumma.

**Model Evaluation:** Pranjal A. Chitale, Raj Dabre, Sumanth Doddapaneni, Jay Gala, Varun Gumma, Anoop Kunchukuttan, and Ratish Puduppully.

**Research Leads:** Raj Dabre, Mitesh M. Khapra, Pratyush Kumar, and Anoop Kunchukuttan.

**Project Conceptualization and Direction:** Mitesh M. Khapra, Pratyush Kumar, Anoop Kunchukuttan, and Vivek Raghavan.

## Acknowledgements

Embarking on this mission was only possible due to the support of numerous organizations, individuals and members of the Indian language technology ecosystem. We would like to take a few sentences to thank all of them.

**Sponsors/Donors**: First and foremost, we thank the Ministry of Electronics and Information Technology (MeitY), Government of India, for setting up the ambitious Digital India Bhashini Mission with the goal of advancing Indian language technology. The human infrastructure comprising of a large team of translators, reviewers and language experts who worked on this project were supported by the generous grant given by Digital India Bhashini Mission to IIT Madras to serve as the Data Management Unit for the mission.

We are indebted to Shri Nandan Nilekani and Shrimati Rohini Nilekani for believing in us and supporting our work through generous grants from EkStep Foundation and Nilekani Philanthropies. These grants were used for (i) supporting many of the students, research associates, and developers who worked on this project, (ii) fulfilling many of our compute needs, and (iii) recruiting project managers to oversee the massive pan-India data collection activity undertaken as a part of this work.

We thank Microsoft for their grant to support the creation of benchmarks for Indian languages.

We thank the Centre for Development and Advancement of Computing, Pune (CDAC Pune) for access to its Param Siddhi super-computer which was used for mining bitext pairs at scale.

**IIT Madras**: We thank Prof. V Kamakoti (Director, IIT Madras), Prof. Mahesh V Panchagnula (Dean, IIT Madras), Prof. Ravindra Gettu (Dean, IIT Madras) and Prof. Manu Santhanam (Dean, IIT Madras) for their constant encour-

agement and administrative support. In particular, we are thankful for the office space provided to AI4Bharat which houses some of our students, researchers, language experts and administrative team.

**Indian language technology community**: We extend our heartfelt gratitude to the expansive Indian language technology community, comprising academia, startups, and the deep tech industry, both within India and across the globe. It is with immense gratitude that we acknowledge the incredible foundation laid by the giants of this community, whose pioneering work has paved the way for our endeavors. We are truly grateful for the knowledge, insights, and advancements that we have built upon, as we stand on the shoulders of these remarkable contributors. In particular, we thank Prof. Rajeev Sangal (Professor Emeritus, IIIT Hyderabad), Prof. Pushpak Bhattacharyya (IIT Bombay), Prof. Dipti Mishra (IIIT Hyderabad), Prof. Hema Murthy (IIT Madras), Prof. Umesh S (IIT Madras), Prof. Rajat Moona (IIT Gandhinagar), Prof. Ganesh Ramakrishnan (IIT Bombay), Partha Talukdar (Google Research India), Dr. Swaran Lata (MeitY), Dr. Sobha L (AU-KBC) and Dr. Ritesh Kumar (Dr. B.R. Ambedkar University) for their critical insights and constructive feedback in improving the translation guidelines used for creating the datasets released as a part of this work (we apologize if we have missed anyone).

**Research organisations**: We thank Google for open-sourcing the LaBSE embeddings which we used extensively for mining and filtering bitext pairs. We thank Meta for open-sourcing their semantic search infrastructure, FAISS, which we use for indexing and mining bitext pairs. We thank Allen-AI for reproducing the work of NLLB and releasing a large mined parallel corpus for Indian languages.

**Language Experts**: We express our deepest gratitude to our exceptional and highly dedicated team of language experts, including translators and reviewers, whose invaluable contributions have been instrumental in the creation of the seed data and benchmark data. Their unwavering commitment to adhering to guidelines and their remarkable ability to work seamlessly as a cohesive unit, despite being geographically dispersed, is truly commendable. The quality and accuracy of the manual datasets developed as part of this endeavor owes much to their unwavering efforts. We extend our heartfelt thanks to every member of our remarkable language team for their outstanding dedication and invaluable contributions.

**Administration Team**: We are profoundly thankful to the remarkable individuals, Krishnan Karunganni S and Ravishankar Venkateswaran, for their exceptional dedication, patience, and extraordinary leadership in managing such an expansive team of talented translators. Their unwavering commitment to orchestrating and guiding this diverse group of language experts is truly commendable. Through their exceptional organizational skills and expertise, they ensured seamless coordination and maintained the highest standards of quality throughout the translation process. We also thank our support staff Shanthi S, Bhanumathy M, Bhavana R, Suganya Kumaresan, and Kalaivanan A, who helped with recruitment and procurement.

**Development Team**: We also thank our development team comprising of our in-house engineers, as well as, engineers from Tarento for building Shoonya which enabled all the manual translation work. In the absence of Shoonya, it would have been impossible to manage such a diverse team spread across the country working towards a common goal. We thank members of our development team for their patience in working with the language experts and building features that helped improve both the speed and quality of translation.

**Partners**: We would also like to thank our start-up partners, *viz.*, Desicrew, Devanagari, Language Services Bureau and Keypoint Technologies, who helped in meeting some of our manual translation goals.

**Reviewers**: We would like to thank Dr. Benjamin Marie (4i) for reviewing the modeling and evaluation sections of our paper and helping us gain confidence in the credibility of our evaluation process.

**NICT**: Raj Dabre would like to thank Dr. Eiichiro Sumita and Dr. Masao Utiyama of ASTREC at NICT, for the freedom and encouragement to collaborate with AI4Bharat.

Last, but not the least, we thank the Almighty for giving us the courage to embark on this mission!

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

# A    Data Contribution and Coverage

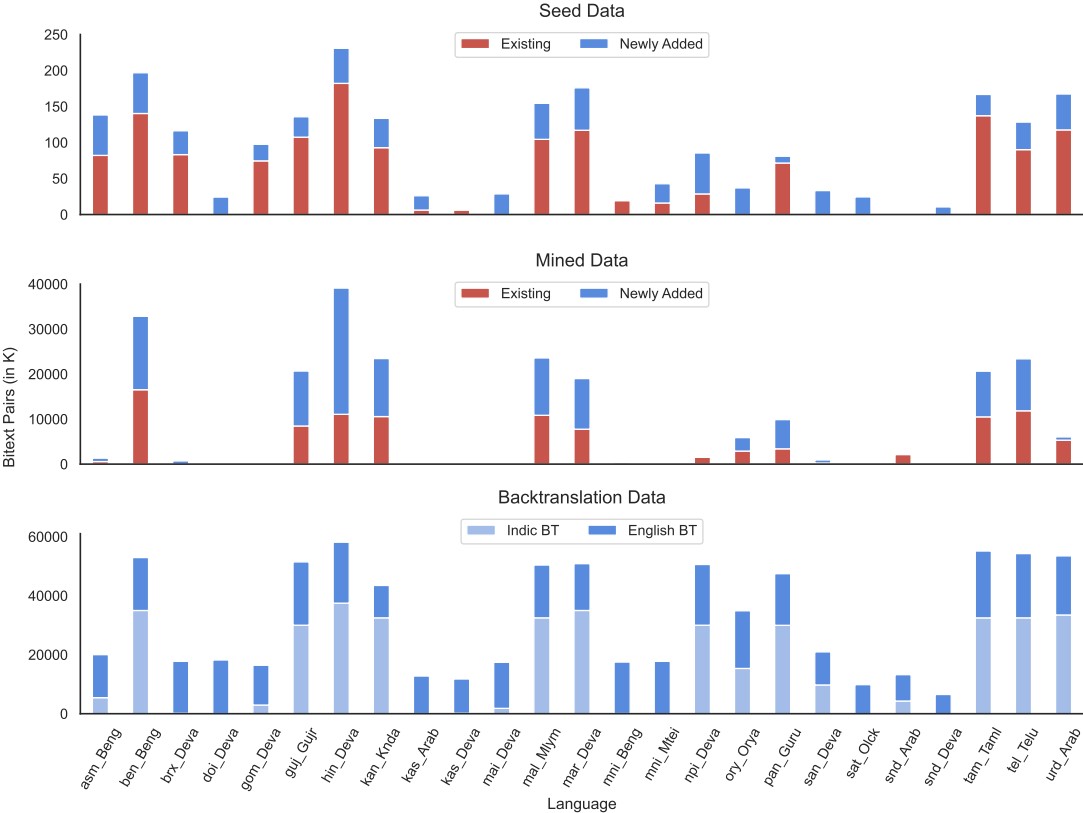

Figure 7: Overview of our training data contributions across different axes: Seed, Mined, and Backtranslation. Indic BT indicates bitext pairs with the English side as synthetic and Indic side as original, whereas English BT indicates vice versa.

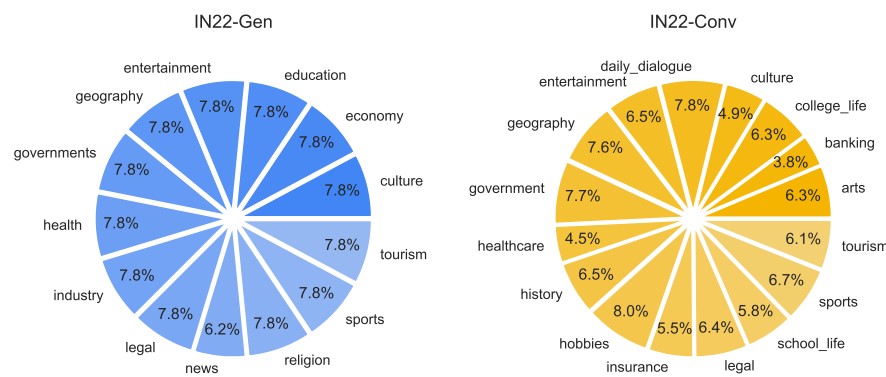

Figure 8: Overview of the domain coverage of our newly created IN22-Gen (left) and IN22-Conv (right) benchmarks.

# B  Additional Results

## B.1  Zero-Shot Translation Capabilities of IndicTrans2 Through Cross-Lingual Transfer

Zero-shot translation (Johnson et al., 2017) is a challenging task, but it is becoming increasingly feasible with the development of more powerful MT models. Zero-shot translation refers to the ability of an MT model to translate from a source language to a target language, even if it has never seen any training data for the language pair before. This is primarily attributed to cross-lingual transfer learning that involves knowledge transfer from one language to another. There are several benefits to good zero-shot performance. First, it indicates that the MT model has good generalization capabilities, which means that the model is able to learn the underlying structure of languages rather than simply memorizing specific translation pairs. Second, it suggests that the MT model can learn language representations shared across different languages. In addition, this makes it easier to extend the model to new languages, even with limited data.

Table 23: chrF++ scores of our IT2 in the zero-shot setting in the Indic-En direction on Indic languages on the FLORES-200 Evaluation set. The best-performing system is bolded, and $\Delta$ represents the difference between the zero-shot score of IT2 and the score of the SOTA model. IT2 results are presented based on the decoding for the Maithili language tag.

| language | N1.2 | N54 | IT2 | $\Delta$ |
|---|---|---|---|---|
| awa_Deva | 63.2 | **65.4** | 62.4 | -3.0 |
| bho_Deva | 57.3 | **58.5** | 53.6 | -4.9 |
| hne_Deva | 70.6 | **72.2** | 62.1 | -10.1 |
| mag_Deva | 70.3 | **72.0** | 67.4 | -4.6 |

In this study, we investigate the cross-lingual transfer and generalizability of our IndicTrans2 models. Our focus lies on performing zero-shot evaluation on a set of additional low-resource Indic languages, which are supported by the NLLB (Costa-jussà et al., 2022) models (1.2B distilled and 54B MoE) and are included as part of the FLORES-200 (Costa-jussà et al., 2022) evaluation set. Specifically, we restrict our evaluations to the Indic-En model, as the structure and syntax of these low-resource languages as target translation is unseen by the model and therefore, result in off-target translation. However, in the case of the Indic-En direction, such an analysis is feasible since the target language, English, is supported by the model. We consider languages like Awadhi, Bhojpuri, Chhattisgarhi, and Magahi that are written in the Devanagari script, which is the prominent script supported by our models. We also have test sets available in FLORES-200 for evaluation. We employ a top-down approach based on language similarity to facilitate zero-shot decoding. Specifically, we select the top-3 related languages that are closest to the aforementioned languages under consideration. Using this approach, we identify Hindi, Maithili, and Nepali as the three closest languages and leverage their language codes for zero-shot decoding of the new Indic languages. We follow the same generation and evaluation procedure mentioned in Section 6.4 and Section 6.5. We observe that decoding with the language tag of Maithili yields the best performance on the test set across all four languages, followed by Hindi and Nepali. This finding highlights that Maithili is closer to these languages in the embedding space than Hindi or Nepali. Table 23 demonstrates that our IndicTrans2 model differs by around 4 points on average except for Chhattisgarhi when compared to the NLLB 54B MoE model that is explicitly using the sentence pairs of the aforementioned languages in training. **Overall, our IndicTrans2 model shows promising results in zero-shot performance on low-resource languages, highlighting the potential for extending to new languages with limited data in the future**.

## B.2  Translation Capabilities of Zero-Shot Prompted LLMs

Large language models (LLMs) such as GPT (Brown et al., 2020; OpenAI, 2023) have recently shown impressive zero-shot performance on various tasks. In this work, we compare the zero-shot translation capabilities of GPT3.5 (as described in Section 6.1) with our best IndicTrans2 model. The prompt template "Translate the following sentence into {{lang}}\n {{text}}" was used for evaluation. Table 24 demonstrates that **our IndicTrans2 models outperform GPT3.5 by a significant margin on both the IN22-Gen and IN22-Conv sets in both En-Indic and Indic-En directions. However, it is important to note that this gap is comparatively lower on the IN22-Conv**

Table 24: chrF++ scores of GPT3.5 (`gpt-3.5-turbo`) on the IN22-Gen (left) and IN22-Conv (right) Evaluation sets in the En-Indic and Indic-En directions. *Avg.* means the average score of all the top-13 languages. Δ represents the difference between the scores of IT2 and GPT3.5. Positive Δ indicates IT2 is better than X and vice-versa.

| | IN22-Gen | | | | | | IN22-Conv | | | | | |
| | En-Indic | | | Indic-En | | | En-Indic | | | Indic-En | | |
| *language* | GPT3.5 | IT2 | Δ | GPT3.5 | IT2 | Δ | GPT3.5 | IT2 | Δ | GPT3.5 | IT2 | Δ |
|---|---|---|---|---|---|---|---|---|---|---|---|---|
| *asm_Beng* | 25.9 | **47.1** | 21.2 | 46.9 | **65.8** | 18.9 | 27.2 | **46.8** | 19.6 | 43.6 | **62.9** | 19.3 |
| *ben_Beng* | 39.9 | **51.8** | 11.9 | 52.1 | **63.2** | 11.1 | 39.9 | **49.7** | 9.8 | 52.9 | **58.4** | 5.5 |
| *guj_Gujr* | 35.6 | **53.5** | 17.9 | 51.7 | **66.5** | 14.8 | 36.0 | **53.1** | 17.1 | 50.9 | **62.0** | 11.1 |
| *hin_Deva* | 47.1 | **56.7** | 9.6 | 57.7 | **65.4** | 7.7 | 46.0 | **49.6** | 3.6 | 57.0 | **60.1** | 3.1 |
| *kan_Knda* | 34.5 | **51.0** | 16.5 | 51.7 | **64.2** | 12.5 | 27.9 | **33.8** | 5.9 | 42.1 | **47.5** | 5.4 |
| *mal_Mlym* | 31.6 | **50.9** | 19.3 | 47.8 | **64.5** | 16.7 | 30.4 | **45.7** | 15.3 | 44.0 | **54.3** | 10.3 |
| *mar_Deva* | 33.9 | **51.0** | 17.1 | 50.3 | **63.7** | 13.4 | 34.0 | **48.6** | 14.6 | 47.6 | **58.5** | 10.9 |
| *npi_Deva* | 37.2 | **49.0** | 11.8 | 54.2 | **67.7** | 13.5 | 38.3 | **51.5** | 13.2 | 52.0 | **63.0** | 11.0 |
| *ory_Orya* | 27.8 | **43.9** | 16.1 | 48.0 | **66.2** | 18.2 | 25.6 | **40.2** | 14.6 | 45.2 | **60.3** | 15.1 |
| *pan_Guru* | 36.2 | **50.6** | 14.4 | 51.7 | **63.4** | 11.7 | 40.6 | **57.8** | 17.2 | 53.3 | **62.7** | 9.4 |
| *tam_Taml* | 34.0 | **49.5** | 15.5 | 41.3 | **59.8** | 18.5 | 29.7 | **39.1** | 9.4 | 38.0 | **45.8** | 7.8 |
| *tel_Telu* | 34.3 | **52.4** | 18.1 | 46.5 | **64.8** | 18.3 | 32.1 | **45.5** | 13.4 | 42.4 | **52.9** | 10.5 |
| *urd_Arab* | 47.6 | **68.2** | 20.6 | 58.8 | **73.0** | 14.2 | 49.0 | **61.6** | 12.6 | 57.1 | **65.5** | 8.4 |
| *Avg.* | 35.8 | **52.0** | 16.2 | 50.7 | **65.2** | 14.6 | 35.1 | **47.9** | 12.8 | 48.2 | **58.0** | 9.8 |

**set, likely because GPT3.5 was fine-tuned towards fluency in conversational and interactive contexts.** In addition, the average Δ across both the IN22-Gen and IN22-Conv sets is lower for high-resource languages such as Hindi (+6.6 for Indic-En and +5.4 for En-Indic) than low-resource languages such as Assamese (+19.1 for Indic-En and +20.4 for En-Indic). Overall, our IndicTrans2 models outperform GPT3.5 by an average of 12.2 points and 14.5 points in Indic-En and En-Indic directions, respectively, on our IN22 benchmark. **Even though LLMs show promising zero-shot capabilities in multilingual settings, we observe that these still lag behind the task-specific models, particularly for low-resource languages.** Exploring how richer translations can be extracted from LLMs is an open problem and can be a worthy future study.

### B.3 Comparison with SeamlessM4T Multimodal Translation Model

SeamlessM4T (Communication et al., 2023) is a recently released multimodal translation model supporting 16 Indic languages. In the interest of the community, we report preliminary results of this model on our primary benchmarks, such as FLORES-200, IN22-Gen and IN22-Conv in Tables 25 and 26. We use the SeamlessM4T-Large and SeamlessM4T-Large v2 variants, which are both 2.3B parameter models and the best model released as a part of the work.

### B.4 Results on NTREX

NTREX (Federmann et al., 2022) is a news-domain benchmark that expands coverage of languages of test data from WMT 2019 (Barrault et al., 2019) to 128 languages. Out of these, 13 are scheduled Indic languages. The detailed results are reported in Tables 27 to 29.

### B.5 Results on WAT2020 & WAT2021

WAT (Nakazawa et al., 2020; 2021a) included support for translations for 8 Indic languages in the news domain. In addition, they released data for Hindi-English data in IT and WikiNews domains. WAT 2021 (Nakazawa et al., 2021a) created a benchmark for translation between 10 Indic languages and English. The detailed results are reported in Tables 30 to 35.

Table 25: chrF++ scores of SM4T (SeamlessM4T-Large), SM4Tv2 (SeamlessM4T-Large v2) and IT2 on the FLORES-200 Evaluation sets in the En-Indic and Indic-En directions. *Avg.* means the average score of all the supported languages.

| | En-Indic | | | Indic-En | | |
|---|---|---|---|---|---|---|
| *language* | SM4T | SM4Tv2 | IT2 | SM4T | SM4Tv2 | IT2 |
| *asm_Beng* | 41.4 | 38.7 | **43.3** | 56.4 | 55.9 | **56.9** |
| *ben_Beng* | 52.0 | 50.3 | **54.3** | 61.9 | 60.3 | **62.4** |
| *guj_Gujr* | 53.3 | 51.9 | **56.0** | 66.2 | 65.0 | **67.0** |
| *hin_Deva* | 58.3 | 57.5 | **59.6** | 66.0 | 62.7 | **67.5** |
| *kan_Knda* | 54.2 | 52.4 | **56.1** | 60.4 | 59.1 | **61.5** |
| *mai_Deva* | 46.8 | 43.1 | **50.5** | 66.7 | 66.1 | **69.5** |
| *mal_Mlym* | 53.0 | 50.3 | **57.3** | 63.2 | 60.9 | **64.3** |
| *mar_Deva* | 48.8 | 46.3 | **51.3** | 63.0 | 62.1 | **64.3** |
| *mni_Beng* | 38.2 | 39.0 | **38.2** | 50.9 | 50.0 | **52.9** |
| *npi_Deva* | 52.8 | 50.7 | **57.2** | 66.3 | 65.5 | **68.1** |
| *ory_Orya* | **49.9** | 46.0 | 49.2 | 63.2 | 62.7 | **64.9** |
| *pan_Guru* | 52.6 | 50.6 | **53.5** | 65.4 | 64.2 | **66.4** |
| *snd_Arab* | 51.8 | 49.8 | **44.9** | 64.3 | 61.0 | **65.1** |
| *tam_Taml* | 54.8 | 52.6 | **57.2** | 59.4 | 57.7 | **61.3** |
| *tel_Telu* | 56.7 | 54.6 | **59.4** | 65.1 | 62.8 | **66.1** |
| *urd_Arab* | 50.1 | 49.4 | **52.2** | 62.0 | 59.9 | **62.0** |
| *Avg.* | 50.9 | 49.0 | 52.5 | 62.5 | 61.0 | 63.8 |

Table 26: chrF++ scores of SM4T (SeamlessM4T-Large), SM4Tv2 (SeamlessM4T-Large v2) and IT2 on the IN22-Gen (left) and IN22-Conv (right) Evaluation sets in the En-Indic and Indic-En directions. *Avg.* means the average score of all the supported languages.

| | IN22-Gen | | | | | | IN22-Conv | | | | | |
|---|---|---|---|---|---|---|---|---|---|---|---|---|
| | En-Indic | | | Indic-En | | | En-Indic | | | Indic-En | | |
| *language* | SM4T | SM4Tv2 | IT2 | SM4T | SM4Tv2 | IT2 | SM4T | SM4Tv2 | IT2 | SM4T | SM4Tv2 | IT2 |
| *asm_Beng* | 43.8 | 40.6 | **47.1** | 63.8 | 62.4 | **65.8** | 45.5 | 43.9 | **46.8** | 60.5 | 60.6 | **62.9** |
| *ben_Beng* | 47.9 | 46.2 | **51.8** | 61.7 | 58.7 | **63.2** | 47.7 | 46.8 | **49.7** | 57.8 | 57.2 | **58.4** |
| *guj_Gujr* | 49.1 | 47.5 | **53.5** | 64.9 | 62.5 | **66.5** | 49.7 | 48.9 | **53.1** | 61.7 | 60.8 | **62.0** |
| *hin_Deva* | 53.5 | 52.8 | **56.7** | 62.6 | 59.8 | **65.4** | 47.1 | 47.1 | **49.6** | 59.5 | 58.3 | **60.1** |
| *kan_Knda* | 47.5 | 46.4 | **51.0** | 62.7 | 59.9 | **64.2** | 32.3 | 32.0 | **33.8** | 46.3 | 45.2 | **47.5** |
| *mai_Deva* | 45.4 | 41.9 | **48.7** | 63.2 | 61.4 | **64.8** | 42.8 | 41.6 | **44.3** | 56.5 | 55.6 | **57.8** |
| *mal_Mlym* | 46.9 | 45.1 | **50.9** | 61.6 | 57.9 | **64.5** | 42.0 | 41.3 | **45.7** | 53.4 | 51.4 | **54.3** |
| *mar_Deva* | 45.3 | 43.3 | **51.0** | 61.6 | 60.1 | **63.7** | 46.0 | 44.5 | **48.6** | 57.6 | 57.1 | **58.5** |
| *npi_Deva* | 46.8 | 44.2 | **49.0** | 66.0 | 64.8 | **67.7** | 47.7 | 46.5 | **51.5** | 61.1 | 61.2 | **63.0** |
| *ory_Orya* | **45.2** | 40.9 | 43.9 | 64.2 | 62.6 | **66.2** | **42.7** | 41.0 | 40.2 | 60.2 | 60.0 | **60.3** |
| *pan_Guru* | 49.7 | 48.0 | **50.6** | 61.1 | 59.4 | **63.4** | 56.5 | 55.2 | **57.8** | 61.6 | 61.2 | **62.7** |
| *tam_Taml* | 47.5 | 45.9 | **49.5** | **57.9** | 54.8 | **59.8** | 37.4 | 37.0 | **39.1** | **46.2** | 45.4 | 45.8 |
| *tel_Telu* | 49.1 | 47.2 | **52.4** | 62.6 | 58.9 | **64.8** | 39.8 | 39.3 | **45.5** | 52.8 | 51.6 | **52.9** |
| *urd_Arab* | 62.7 | 60.5 | **68.2** | 69.4 | 66.2 | **73.0** | 55.0 | 54.2 | **61.6** | 62.8 | 62.7 | **65.5** |
| *Avg.* | 48.6 | 46.5 | 51.7 | 63.1 | 60.7 | 65.2 | 45.2 | 44.2 | 47.7 | 57.0 | 56.3 | 58.0 |

## B.6 Results on WMT & UFAL

WMT has created benchmarks for selected Indic languages as part of shared tasks in 2014 (Hindi) (Bojar et al., 2014), 2019 (Gujarati) (Barrault et al., 2019) and 2020 (Tamil) (Barrault et al., 2020).

Table 27: chrF++ scores of all the systems on the NTREX (Federmann et al., 2022) Evaluation set in the En-Indic and Indic-En direction. The best-performing system is bolded, while underlined results indicate significant performance difference where IT2 outperforms the system. *Avg* means the average score of all the languages that system X supports. Δ represents the difference between the average scores of IT2 and the average scores of system X for the subset of languages that both X and IT2 support. A positive value for Δ indicates IT2 is better than X and vice-versa.

| | En-Indic | | | | | | Indic-En | | | | | |
|---|---|---|---|---|---|---|---|---|---|---|---|---|
| *Language* | IT1 | M100 | N1.2 | IT2 | Goog | Az | IT1 | M100 | N1.2 | IT2 | Goog | Az |
| *ben_Beng* | 48.4 | 45.8 | 50.8 | **54.0** | 53.5 | 52.0 | 55.9 | 53.8 | 60.4 | 62.9 | **63.3** | 59.9 |
| *guj_Gujr* | 44.4 | 19.5 | 47.8 | **49.6** | 49.3 | 49.7 | 57.5 | 10.9 | 63.7 | 66.8 | 66.7 | 61.9 |
| *hin_Deva* | 50.0 | 48.0 | 51.6 | 53.3 | **53.7** | 52.1 | 57.4 | 55.9 | 61.5 | 63.7 | 63.6 | 59.7 |
| *kan_Knda* | 49.2 | 14.3 | 51.2 | **54.1** | 54.0 | 54.1 | 52.6 | 12.0 | 57.9 | 61.2 | 61.3 | 57.3 |
| *mal_Mlym* | 43.4 | 32.6 | 41.7 | **48.6** | 48.0 | 47.0 | 51.9 | 47.3 | 56.7 | 59.6 | 60.0 | 56.5 |
| *mar_Deva* | 40.6 | 36.5 | 43.5 | **47.0** | 46.4 | 44.5 | 54.0 | 48.3 | 59.7 | 62.7 | 63.0 | 57.5 |
| *npi_Deva* | - | 14.2 | 41.7 | **45.0** | 44.7 | 41.5 | - | 37.4 | 62.2 | 64.4 | **65.5** | 59.8 |
| *pan_Guru* | 47.5 | 27.7 | 49.1 | 50.3 | **51.6** | 50.3 | 56.7 | 43.0 | 61.8 | 64.9 | 65.0 | 60.4 |
| *snd_Arab* | - | 25.1 | 39.7 | **43.3** | 42.1 | 41.1 | - | 17.8 | 55.8 | 58.2 | 58.5 | 52.1 |
| *tam_Taml* | 41.8 | 14.8 | 43.7 | **45.9** | 45.4 | 45.4 | 49.4 | 29.5 | 54.5 | 57.0 | 57.2 | 53.4 |
| *tel_Telu* | 42.0 | - | 43.9 | **46.7** | 46.8 | 43.8 | 48.7 | - | 53.1 | 55.6 | 55.8 | 52.2 |
| *urd_Arab* | - | 41.7 | 51.4 | **53.7** | 53.1 | 52.9 | - | 48.2 | 60.6 | 62.5 | **63.0** | 59.6 |
| *Avg.* | 45.3 | 29.1 | 46.3 | 49.3 | 49.1 | 47.9 | 53.8 | 36.7 | 59.0 | 61.6 | 61.9 | 57.6 |
| Δ | 4.6 | 20.4 | 3.0 | - | 0.2 | 1.4 | 7.7 | 25.5 | 2.6 | - | -0.3 | 4.0 |

Table 28: COMET scores of all the systems on the NTREX (Federmann et al., 2022) Evaluation set in the En-Indic and Indic-En direction. The best performing system is bolded, while underlined results indicate significant performance difference where IT2 outperforms the system.

| | En-Indic | | | | | | Indic-En | | | | | |
|---|---|---|---|---|---|---|---|---|---|---|---|---|
| *Language* | IT1 | M100 | N1.2 | IT2 | Goog | Az | IT1 | M100 | N1.2 | IT2 | Goog | Az |
| *ben_Beng* | 85.3 | 82.6 | 86.1 | 86.4 | 85.2 | **86.7** | 86.5 | 85.8 | 88.4 | 88.9 | **89.3** | 88.0 |
| *guj_Gujr* | 86.8 | 61.6 | 86.7 | **87.9** | 87.1 | 87.5 | 86.3 | 36.4 | 88.8 | **89.6** | 89.7 | 87.6 |
| *hin_Deva* | 77.6 | 74.3 | 77.9 | **78.7** | 77.9 | 78.7 | 86.2 | 85.6 | 88.0 | **88.4** | 88.5 | 86.9 |
| *kan_Knda* | 84.1 | 51.6 | 84.5 | **85.6** | 84.2 | 85.8 | 84.5 | 35.9 | 86.7 | 87.7 | 87.7 | 86.2 |
| *mal_Mlym* | 85.7 | 75.4 | 86.3 | **87.5** | 86.5 | 87.4 | 85.5 | 81.5 | 87.6 | 88.3 | **88.7** | 87.3 |
| *mar_Deva* | 71.8 | 65.8 | 73.9 | **74.6** | 73.1 | 74.5 | 85.4 | 80.4 | 87.6 | **88.4** | 88.5 | 86.5 |
| *npi_Deva* | - | 51.8 | 79.1 | **80.6** | 79.8 | 79.9 | - | 68.4 | 89.1 | 89.4 | **90.1** | 87.8 |
| *pan_Guru* | 82.4 | 60.5 | **83.1** | 83.0 | 82.9 | 83.2 | 84.3 | 73.6 | 86.9 | 87.6 | **87.8** | 85.4 |
| *tam_Taml* | 85.5 | 53.4 | 86.0 | 86.4 | 86.2 | **87.3** | 82.9 | 62.6 | 85.2 | 85.9 | **86.2** | 84.2 |
| *tel_Telu* | 83.5 | - | 83.4 | **85.1** | 84.4 | 85.3 | 83.6 | - | 86.2 | 87.0 | **87.1** | 85.3 |
| *urd_Arab* | - | 72.7 | 81.0 | 82.2 | 82.3 | **83.7** | - | 79.9 | 87.3 | 87.8 | **88.0** | 86.9 |

UFAL (Ramasamy et al., 2012) is an English-Tamil bilingual benchmark created from publicly available websites. The benchmark consists of English sentences from domains such as cinema, news, and some biblical sources.

Detailed results are reported in Tables 36 to 38.

## B.7 COMET Scores for IN22 & FLORES

We report COMET (Rei et al., 2022) scores for IN22 and FLORES (Costa-jussà et al., 2022) in Tables 39 to 41

## B.8 BLEU Scores for IN22 & FLORES

We report BLEU (Papineni et al., 2002) scores for IN22 and FLORES (Costa-jussà et al., 2022) in Tables 42 to 44

Table 29: BLEU scores of all the systems on the NTREX (Federmann et al., 2022). Evaluation set in the En-Indic and Indic-En direction. The best performing system is bolded, while underlined results indicate significant performance difference where IT2 outperforms the system.

| | En-Indic | | | | | | Indic-En | | | | | |
|---|---|---|---|---|---|---|---|---|---|---|---|---|
| Language | IT1 | M100 | N1.2 | IT2 | Goog | Az | IT1 | M100 | N1.2 | IT2 | Goog | Az |
| ben_Beng | 17.7 | 15.3 | 19.8 | **22.9** | **23** | 20.8 | 30.7 | 27.7 | 36.6 | **40.2** | 40.5 | 35.4 |
| guj_Gujr | 15.4 | 3.1 | 18.7 | **20.4** | **20.7** | 20.5 | 32.1 | 0.5 | 40.5 | **45.2** | 44.6 | 37.4 |
| hin_Deva | 26.4 | 24.3 | 28.2 | 30.5 | **31.2** | 29 | 31.1 | 28.9 | 37.7 | **41.3** | 40.7 | 34.6 |
| kan_Knda | 16.4 | 1.1 | 18.5 | 22 | **22.7** | 22.7 | 26.9 | 0.5 | 33.8 | **38.6** | 38.6 | 31.8 |
| mal_Mlym | 11 | 5.4 | 8.1 | **14.5** | **14.6** | 14 | 25.6 | 19.5 | 31.4 | 34.8 | **35.9** | 28.9 |
| mar_Deva | 10.5 | 8.2 | 12.2 | 14.6 | **15.1** | 12.7 | 28 | 21.8 | 35.6 | **40.1** | 40.1 | 31.1 |
| npi_Deva | - | 0.8 | 11.5 | **13.7** | **13.7** | 10.8 | - | 10.3 | 38.9 | 42.4 | **43.4** | 35.1 |
| pan_Guru | 22.6 | 8.5 | 24.5 | 25.5 | **26.8** | 24.6 | 31.5 | 14 | 38.6 | **43.3** | 43.2 | 35.9 |
| snd_Arab | - | 6.4 | 13.7 | **18.7** | 16 | 15 | - | 2 | 33.3 | **37.1** | 37 | 28.5 |
| tam_Taml | 9 | 0.8 | 9.9 | 11.8 | **11.9** | 11.4 | 23.5 | 6.1 | 30.2 | 33.4 | **33.6** | 26.9 |
| tel_Telu | 11 | - | 12.1 | 15.4 | **15.6** | 12 | 22.8 | - | 28.7 | 32.6 | 32.6 | 27.1 |
| urd_Arab | - | 18.3 | 27.7 | **30.5** | 30.1 | 29.5 | - | 22 | 36.5 | 39.4 | **39.6** | 35.1 |
| Avg. | 15.6 | 8.4 | 17.1 | 20 | 20.1 | 18.6 | 28 | 13.9 | 35.2 | 39 | 39.2 | 32.3 |
| Δ | 4.1 | 12.1 | 2.9 | - | -0.1 | 1.4 | 10.8 | 25.7 | 3.8 | - | -0.2 | 6.7 |

Table 30: chrF++ scores of all the systems on the WAT-2020 (Nakazawa et al., 2020). Evaluation set in the En-Indic and Indic-En direction. The best performing system is bolded, while underlined results indicate significant performance difference where IT2 outperforms the system. *Avg* means the average score of all the languages that system X supports. Δ represents the difference between the average scores of IT2 and the average scores of system X for the subset of languages that both X and IT2 support. A positive value for Δ indicates IT2 is better than X and vice-versa.

| | En-Indic | | | | | | Indic-En | | | | | |
|---|---|---|---|---|---|---|---|---|---|---|---|---|
| Language | IT1 | M100 | N1.2 | IT2 | Goog | Az | IT1 | M100 | N1.2 | IT2 | Goog | Az |
| ben_Beng | **38.9** | 31.7 | 36.6 | 37.9 | 36.4 | 37.5 | **44.0** | 36.1 | 42.5 | **44.2** | 42.9 | 43.2 |
| guj_Gujr | 42.7 | 18.2 | 41.2 | 41.9 | 41.2 | **45.4** | 48.6 | 9.2 | 48.1 | 49.3 | 48.0 | **49.6** |
| hin_Deva | **43.3** | 35.6 | 41.3 | 41.8 | 42.7 | 41.7 | 48.1 | 40.5 | 47.0 | 48.9 | **49.5** | 49.5 |
| mal_Mlym | 38.4 | 29.8 | 36.2 | **38.8** | 38.2 | 38.6 | 44.5 | 31.9 | 43.1 | 45.0 | 43.5 | **45.5** |
| mar_Deva | **41.6** | 31.9 | 39.8 | 41.0 | 40.5 | 40.7 | 44.9 | 34.4 | 44.2 | **45.8** | 45.0 | 44.9 |
| tam_Taml | 37.5 | 15.1 | 36.4 | **37.9** | 36.8 | 37.9 | **42.9** | 19.8 | 41.7 | **42.9** | 41.3 | 43.0 |
| tel_Telu | 37.2 | - | 36.7 | 37.7 | 36.8 | **38.0** | 43.0 | - | 42.2 | **43.7** | 42.5 | 43.8 |
| Avg. | 39.9 | 27.1 | 38.3 | 39.6 | 38.9 | 40.0 | 45.1 | 28.7 | 44.1 | 45.7 | 44.7 | 45.6 |
| Δ | -0.3 | 12.8 | 1.4 | - | 0.7 | -0.4 | 0.6 | 17.3 | 1.6 | - | 1.0 | 0.1 |

Table 31: chrF++ scores of all the systems on the WAT-2021 (Nakazawa et al., 2021a). Evaluation set in the En-Indic and Indic-En direction. The best performing system is bolded, while underlined results indicate significant performance difference where IT2 outperforms the system. *Avg* means the average score of all the languages that system X supports. Δ represents the difference between the average scores of IT2 and the average scores of system X for the subset of languages that both X and IT2 support. A positive value for Δ indicates IT2 is better than X and vice-versa.

| | En-Indic | | | | | | Indic-En | | | | | |
|---|---|---|---|---|---|---|---|---|---|---|---|---|
| *Language* | IT1 | M100 | N1.2 | IT2 | Goog | Az | IT1 | M100 | N1.2 | IT2 | Goog | Az |
| *ben_Beng* | **45.4** | 34.7 | 41.4 | 42.4 | 39.1 | 41.6 | **53.7** | 42.5 | 51.6 | 52.5 | 49.9 | 50.0 |
| *guj_Gujr* | 53.9 | 21.4 | 51.8 | 52.1 | 48.9 | 58.2 | 62.8 | 8.0 | 61.2 | **62.9** | 59.9 | 62.1 |
| *hin_Deva* | **60.8** | 51.0 | 59.2 | 59.7 | 59.3 | 59.6 | **66.1** | 54.2 | 63.4 | 65.1 | 64.9 | 65.9 |
| *kan_Knda* | **52.5** | 17.6 | 50.2 | 50.9 | 49.0 | 51.8 | **60.0** | 8.9 | 58.3 | 60.3 | 57.0 | 55.0 |
| *mal_Mlym* | **49.5** | 32.7 | 44.9 | 49.2 | 47.5 | 46.4 | **58.4** | 35.0 | 56.2 | 58.3 | 55.2 | 57.7 |
| *mar_Deva* | **50.4** | 36.2 | 47.8 | 49.0 | 47.5 | 48.0 | **57.1** | 40.2 | 55.2 | **57.1** | 54.3 | 55.1 |
| *ory_Orya* | **48.5** | 7.4 | 47.5 | 44.2 | 40.2 | 45.4 | **57.2** | 13.2 | 55.7 | 56.8 | 52.9 | 56.0 |
| *pan_Guru* | 56.1 | 25.6 | 53.0 | 54.2 | 52.6 | **58.7** | 65.2 | 31.9 | 62.9 | 64.8 | 62.2 | 63.5 |
| *tam_Taml* | **48.8** | 14.3 | 46.0 | 47.5 | 45.7 | 47.2 | **56.6** | 18.6 | 54.0 | 55.6 | 51.6 | 54.0 |
| *tel_Telu* | **46.7** | - | 44.9 | 45.3 | 43.0 | 43.0 | **59.7** | - | 56.5 | 59.6 | 56.0 | 58.3 |
| *Avg.* | 51.3 | 26.8 | 48.7 | 49.4 | 47.3 | 50.0 | 59.7 | 28.1 | 57.5 | 59.3 | 56.4 | 57.8 |
| Δ | -1.9 | 23.1 | 0.7 | - | 2.1 | -0.6 | 31.2 | 1.8 | - | 2.9 | 1.5 | |

Table 32: COMET scores of all the systems on the WAT 2020 (Nakazawa et al., 2020). Evaluation set in the En-Indic and Indic-En direction. The best performing system is bolded, while underlined results indicate significant performance difference where IT2 outperforms the system.

| | En-Indic | | | | | | Indic-En | | | | | |
|---|---|---|---|---|---|---|---|---|---|---|---|---|
| *Language* | IT1 | M100 | N1.2 | IT2 | Goog | Az | IT1 | M100 | N1.2 | IT2 | Goog | Az |
| *ben_Beng* | 86.4 | 82.7 | 86.1 | **86.6** | 85.6 | 86.5 | 83.6 | 78.2 | 83.6 | **83.9** | 83.8 | 83.5 |
| *guj_Gujr* | 90.2 | 66.6 | 89.9 | **90.4** | 90.1 | 90.5 | 86.4 | 35.8 | 86.6 | **86.8** | 86.4 | 86.5 |
| *hin_Deva* | 81.5 | 77.0 | 81.3 | 81.5 | **81.7** | 81.3 | **84.2** | 76.8 | 83.8 | 84.1 | 84.0 | 84.0 |
| *mal_Mlym* | 87.9 | 80.5 | 88.2 | 88.1 | 87.6 | **88.5** | 83.9 | 73.0 | 84.0 | **84.6** | 84.3 | **84.5** |
| *mar_Deva* | 77.6 | 69.5 | 77.1 | **77.8** | 77.2 | 77.7 | 83.7 | 72.9 | 83.9 | **84.2** | 84.0 | 83.9 |
| *tam_Taml* | 89.0 | 57.1 | 88.7 | **89.4** | 88.8 | 89.3 | 82.4 | 55.7 | 82.7 | **82.8** | 82.5 | 82.5 |
| *tel_Telu* | 86.3 | - | 86.1 | **86.9** | 86.3 | **86.9** | 83.1 | - | 83.2 | **83.7** | 83.4 | 83.4 |

Table 33: COMET scores of all the systems on the WAT 2021 (Nakazawa et al., 2021a). Evaluation set in the En-Indic and Indic-En direction. The best performing system is bolded, while underlined results indicate significant performance difference where IT2 outperforms the system.

| | En-Indic | | | | | | Indic-En | | | | | |
|---|---|---|---|---|---|---|---|---|---|---|---|---|
| *Language* | IT1 | M100 | N1.2 | IT2 | Goog | Az | IT1 | M100 | N1.2 | IT2 | Goog | Az |
| *ben_Beng* | **88.2** | 84.4 | 87.5 | 87.9 | 86.6 | 87.6 | **86.9** | 82.6 | 87.0 | 87.1 | 86.7 | 86.5 |
| *guj_Gujr* | 92.2 | 70.5 | 92.0 | 92.1 | 91.5 | **92.6** | 90.4 | 35.0 | 90.6 | **90.8** | 90.3 | 90.5 |
| *hin_Deva* | **86.4** | 82.3 | 86.1 | 86.1 | 86.1 | 86.2 | 90.5 | 86.2 | 90.3 | **90.7** | 90.5 | **90.7** |
| *kan_Knda* | **90.2** | 60.7 | 89.9 | 90.1 | 89.3 | **90.2** | 88.2 | 34.7 | 88.5 | **88.7** | 88.1 | 87.1 |
| *mal_Mlym* | 90.9 | 82.8 | 91.1 | 90.9 | 90.3 | **91.5** | 88.7 | 74.6 | 88.7 | **89.3** | 88.5 | 89.0 |
| *mar_Deva* | **81.2** | 72.9 | 80.7 | 80.9 | 80.1 | 80.7 | 87.8 | 77.3 | 87.8 | **88.1** | 87.5 | 87.6 |
| *ory_Orya* | 88.0 | 41.6 | **88.1** | 83.5 | 83.0 | 87.7 | 88.0 | 39.8 | 88.3 | **88.4** | 87.4 | 88.0 |
| *pan_Guru* | 89.3 | 70.3 | 89.0 | 88.9 | 88.9 | **89.6** | 90.0 | 69.7 | 90.0 | **90.2** | 89.7 | 89.7 |
| *tam_Taml* | **92.1** | 53.6 | 91.7 | 91.9 | 91.3 | 91.8 | 87.1 | 53.5 | 87.1 | **87.4** | 86.4 | 86.6 |
| *tel_Telu* | **86.6** | - | 86.3 | 86.4 | 85.8 | 86.3 | 88.3 | - | 87.9 | **88.7** | 87.9 | 88.1 |

Table 34: BLEU scores of all the systems on the WAT-2020 (Nakazawa et al., 2020). Evaluation set in the En-Indic and Indic-En direction. The best performing system is bolded, while underlined results indicate significant performance difference where IT2 outperforms the system. *Avg* means the average score of all the languages that system X supports. Δ represents the difference between the average scores of IT2 and the average scores of system X for the subset of languages that both X and IT2 support. A positive value for Δ indicates IT2 is better than X and vice-versa.

| | En-Indic | | | | | | Indic-En | | | | | |
|---|---|---|---|---|---|---|---|---|---|---|---|---|
| *Language* | IT1 | M100 | N1.2 | IT2 | Goog | Az | IT1 | M100 | N1.2 | IT2 | Goog | Az |
| *ben_Beng* | **12** | 6.1 | 9.7 | 9.8 | 8.5 | 9.7 | **19.9** | 12.7 | 18.1 | 19.5 | 17.5 | 18.1 |
| *guj_Gujr* | 15.5 | 2.4 | 14 | 14.2 | 13.5 | **18.6** | 24.1 | 0.3 | 23 | **24.2** | 22.1 | 24.3 |
| *hin_Deva* | **20.1** | 12.3 | 18 | 18 | 19.5 | 17.9 | 23.6 | 15.7 | 22.2 | **24.2** | 24.3 | 24.6 |
| *mal_Mlym* | **7.3** | 2.9 | 5.1 | 6.9 | 6.5 | 6.7 | **20.4** | 9 | 18.8 | 20.5 | 18.5 | 20.7 |
| *mar_Deva* | **13.2** | 6.4 | 11.5 | 11.7 | 11.4 | 11.6 | **20.4** | 11.2 | 19.3 | 20.6 | 19.2 | 19.5 |
| *tam_Taml* | **6.2** | 0.7 | 5.4 | 5.9 | 5.5 | **6.1** | 18.2 | 2 | 16.8 | 17.9 | 16 | 17.1 |
| *tel_Telu* | 8 | - | 7.4 | 7.5 | 7 | **8.4** | 18.5 | - | 17.5 | **18.8** | 17.4 | 18.5 |
| *Avg.* | 11.8 | 5.1 | 10.2 | 10.6 | 10.3 | 11.3 | 20.7 | 8.5 | 19.4 | 20.8 | 19.3 | 20.4 |
| Delta | -1.2 | 6 | 0.4 | - | 0.3 | -0.7 | 0.1 | 12.7 | 1.4 | - | 1.5 | 0.4 |

Table 35: BLEU scores of all the systems on the WAT-2021 (Nakazawa et al., 2021a). Evaluation set in the En-Indic and Indic-En direction. The best performing system is bolded, while underlined results indicate significant performance difference where IT2 outperforms the system. *Avg* means the average score of all the languages that system X supports. Δ represents the difference between the average scores of IT2 and the average scores of system X for the subset of languages that both X and IT2 support. A positive value for Δ indicates IT2 is better than X and vice-versa.

| | En-Indic | | | | | | Indic-En | | | | | |
|---|---|---|---|---|---|---|---|---|---|---|---|---|
| *Language* | IT1 | M100 | N1.2 | IT2 | Goog | Az | IT1 | M100 | N1.2 | IT2 | Goog | Az |
| *ben_Beng* | **15.8** | 7.4 | 12.1 | 12.6 | 9.5 | 12.1 | **29.5** | 15.4 | 25.9 | 25.7 | 22 | 22.5 |
| *guj_Gujr* | 25.8 | 3.5 | 23.8 | 23.9 | 20.4 | **32.7** | **40.2** | 0.1 | 37.2 | 38.8 | 34.7 | 36.9 |
| *hin_Deva* | **38.8** | 27.3 | 36.7 | 37.6 | 37.2 | 36.4 | **43.9** | 28.8 | 39.7 | 41.6 | 40.6 | 43.1 |
| *kan_Knda* | **19.2** | 1.1 | 16.6 | 16.7 | 14.9 | 18.3 | **36.5** | 0 | 33.8 | 36.3 | 31.3 | 29.5 |
| *mal_Mlym* | **15.1** | 3.9 | 9.2 | 13.7 | 12.4 | 9.5 | **34.6** | 9.7 | 31.2 | 33.6 | 29.2 | 32.4 |
| *mar_Deva* | **20.3** | 8.6 | 17.5 | 18.1 | 16.9 | 17.3 | **33.5** | 14.4 | 30.2 | 32.2 | 28 | 29.7 |
| *ory_Orya* | **19.1** | 0.1 | 17.9 | 13.6 | 10.6 | 15.1 | **34.4** | 0.2 | 31.6 | 32.7 | 27.7 | 30.6 |
| *pan_Guru* | 33.9 | 6.7 | 30 | 31.1 | 29.7 | **37.7** | **43.2** | 6.2 | 39.3 | 41.5 | 37.6 | 38.9 |
| *tam_Taml* | **13.6** | 0.8 | 11.4 | 12.3 | 11.1 | 12.6 | **33.1** | 1.8 | 29.1 | 31.1 | 25.6 | 27 |
| *tel_Telu* | **14.5** | - | 12.9 | 12 | 10.2 | 9.6 | **36.1** | - | 31.6 | 34.4 | 29 | 31.1 |
| *Avg.* | 21.6 | 6.6 | 18.8 | 19.2 | 17.3 | 20.1 | 36.5 | 8.5 | 33 | 34.8 | 30.6 | 32.2 |
| Δ | -2.4 | 13.4 | 0.4 | - | 1.9 | -0.9 | -1.7 | 26.3 | 1.8 | - | 4.2 | 2.6 |

Table 36: chrF++ scores of all the systems on the WMT (Bojar et al., 2014; Barrault et al., 2019; 2020) shared tasks and UFAL (Ramasamy et al., 2012) in the En-Indic and Indic-En direction. The best performing system is bolded, while underlined results indicate significant performance difference where IT2 outperforms the system.

| | | En-Indic | | | | | | Indic-En | | | | | |
|---|---|---|---|---|---|---|---|---|---|---|---|---|---|
| Benchmark | *Language* | IT1 | M100 | N1.2 | IT2 | Goog | Az | IT1 | M100 | N1.2 | IT2 | Goog | Az |
| UFAL | *tam_Taml* | 45.5 | 15.4 | 44.9 | 43.9 | 43.9 | **45.7** | 53.3 | 25.3 | 52.4 | 53.2 | 51.2 | **53.8** |
| WMT14 | *hin_Deva* | 50.5 | 45.9 | 50.7 | 52.1 | **52.7** | 51.9 | 56.6 | 53.6 | 60.4 | 62.1 | **62.7** | 60.4 |
| WMT19 | *guj_Gujr* | 48.8 | 20.7 | 55.4 | 56.3 | 56.8 | **62.2** | 50.5 | 7.9 | 56.4 | 57 | **58.4** | 58.3 |
| WMT20 | *tam_Taml* | 45.7 | 14.4 | 47.5 | **49.2** | 48.2 | **49.2** | 45.8 | 17.5 | 48.1 | 51.3 | **53.5** | 52.3 |

Table 37: COMET scores of all the systems on the WMT (Bojar et al., 2014; Barrault et al., 2019; 2020) shared tasks and UFAL (Ramasamy et al., 2012) in the En-Indic and Indic-En direction. The best performing system is bolded, while underlined results indicate significant performance difference where IT2 outperforms the system.

| Benchmark | Language | En-Indic | | | | | | Indic-En | | | | | |
|---|---|---|---|---|---|---|---|---|---|---|---|---|---|
| | | IT1 | M100 | N1.2 | IT2 | Goog | Az | IT1 | M100 | N1.2 | IT2 | Goog | Az |
| UFAL | *tam_Taml* | 85.8 | 54.3 | 86.3 | 85.8 | 85.4 | **86.8** | 82.0 | 55.9 | 82.7 | **83.0** | 82.5 | 82.7 |
| WMT14 | *hin_Deva* | 81.2 | 77.5 | 81.3 | **81.7** | 81.6 | 81.7 | 84.1 | 80.2 | 86.2 | **86.8** | 86.4 | 85.4 |
| WMT19 | *guj_Gujr* | 86.4 | 61.0 | 86.7 | 87.8 | 87.3 | **88.3** | 82.6 | 30.6 | 84.9 | **85.9** | 85.7 | 85.1 |
| WMT20 | *tam_Taml* | 87.7 | 53.4 | 87.9 | 88.4 | 87.8 | **89.1** | 81.2 | 46.0 | 83.1 | **84.4** | 84.0 | 83.5 |

Table 38: BLEU scores of all the systems on the WMT (Bojar et al., 2014; Barrault et al., 2019; 2020) shared tasks and UFAL (Ramasamy et al., 2012) in the En-Indic and Indic-En direction. The best performing system is bolded, while underlined results indicate significant performance difference where IT2 outperforms the system.

| Benchmark | Language | En-Indic | | | | | | Indic-En | | | | | |
|---|---|---|---|---|---|---|---|---|---|---|---|---|---|
| | | IT1 | M100 | N1.2 | IT2 | Goog | Az | IT1 | M100 | N1.2 | IT2 | Goog | Az |
| UFAL | *tam_Taml* | **10.9** | 0.9 | **10.6** | 8.9 | 9.6 | **10.8** | **30.2** | 4.3 | 28.5 | 28.8 | 25.7 | 28.3 |
| WMT14 | *hin_Deva* | 25.6 | 21.0 | 25.8 | **27.8** | **28.1** | 27.0 | 29.7 | 26.5 | 35.1 | **37.5** | **37.2** | 34.1 |
| WMT19 | *guj_Gujr* | 19.5 | 4.2 | 26.0 | 26.6 | 27.9 | **33.8** | 25.1 | 0.5 | 31.1 | 31.6 | **33.2** | **33.2** |
| WMT20 | *tam_Taml* | 10.3 | 0.7 | 10.9 | **12.6** | **12.0** | **12.1** | 18.5 | 1.7 | 20.6 | 23.2 | **25.5** | 22.4 |

Table 39: COMET scores of all the systems on the IN22-Gen Evaluation set in the En-Indic and Indic-En direction. The best performing system is bolded, while underlined results indicate significant performance difference where IT2 outperforms the system.

| Language | En-Indic | | | | | | | Indic-En | | | | | | |
|---|---|---|---|---|---|---|---|---|---|---|---|---|---|---|
| | IT1 | M100 | N1.2 | N54 | IT2 | Goog | Az | IT1 | M100 | N1.2 | N54 | IT2 | Goog | Az |
| *asm_Beng* | 81.1 | - | 83.4 | 83.2 | **84.7** | 84.0 | 83.5 | 84.1 | - | 87.3 | **88.3** | 87.5 | 87.7 | 85.7 |
| *ben_Beng* | 85.4 | 80.6 | 85.6 | 85.7 | **86.8** | 85.2 | 86.2 | 86.8 | 83.8 | 87.8 | **88.6** | 88.1 | **88.7** | 87.5 |
| *guj_Gujr* | 87.5 | 62.0 | 87.6 | 87.6 | **88.6** | 87.7 | 88.0 | 88.0 | 34.7 | 89.3 | **89.9** | 89.7 | 89.5 | 88.1 |
| *hin_Deva* | 79.5 | 75.2 | 79.6 | 80.0 | **80.5** | 79.2 | 79.3 | 87.8 | 84.7 | 88.4 | **89.1** | 89.2 | 88.7 | 87.9 |
| *kan_Knda* | 84.0 | 52.7 | 84.5 | 84.9 | **85.7** | 83.6 | 85.3 | 86.5 | 34.1 | 87.9 | **88.5** | 87.9 | 88.0 | 86.8 |
| *mal_Mlym* | 86.1 | 73.7 | 86.4 | 87.1 | **87.7** | 86.7 | 87.3 | 86.5 | 77.2 | 87.6 | 88.5 | **88.9** | 87.9 | 86.9 |
| *mar_Deva* | 73.4 | 65.0 | 73.7 | 74.7 | **76.1** | 73.7 | 75.3 | 85.7 | 77.2 | 87.1 | **87.9** | 87.5 | **87.6** | 86.3 |
| *npi_Deva* | - | 54.2 | 80.3 | 78.6 | **82.7** | 80.7 | 81.6 | - | 69.5 | 89.6 | 90.4 | 89.8 | **90.6** | 88.9 |
| *ory_Orya* | 82.2 | 39.1 | 82.9 | 82.8 | 79.5 | 77.4 | **83.6** | 87.4 | 38.2 | 88.5 | **89.4** | 89.0 | 88.4 | 86.7 |
| *pan_Guru* | 82.5 | 60.8 | 82.6 | **82.8** | **83.0** | 82.8 | 82.8 | 84.6 | 67.5 | 86.2 | **87.0** | **87.0** | 86.3 | 84.5 |
| *tam_Taml* | 87.1 | 45.0 | 87.3 | 87.5 | 88.2 | 87.5 | **88.5** | 84.9 | 56.4 | 86.2 | **87.0** | **87.0** | 87.2 | 86.0 |
| *tel_Telu* | 85.1 | - | 85.9 | 86.2 | **87.1** | 86.0 | 86.9 | 86.4 | - | 87.8 | **88.6** | 88.7 | 88.6 | 87.1 |
| *urd_Arab* | - | 73.8 | 84.2 | 84.6 | 85.3 | 85.0 | **86.5** | - | 79.0 | 88.2 | 89.0 | **89.2** | 88.9 | 87.9 |

Table 40: COMET scores of all the systems on the IN22-Conv Evaluation set in the En-Indic and Indic-En direction. The best performing system is bolded, while underlined results indicate significant performance difference where IT2 outperforms the system.

| | En-Indic | | | | | | | Indic-En | | | | | | |
|---|---|---|---|---|---|---|---|---|---|---|---|---|---|---|
| Language | IT1 | M100 | N1.2 | N54 | IT2 | Goog | Az | IT1 | M100 | N1.2 | N54 | IT2 | Goog | Az |
| asm_Beng | 83.2 | - | 85.7 | 85.6 | **86.5** | 84.7 | 85.8 | 84.7 | - | 87.9 | 88.1 | 89.3 | **90.4** | 88.7 |
| ben_Beng | 89.5 | 85.7 | 89.4 | 89.7 | **90.1** | 88.3 | 89.8 | 88.3 | 84.6 | 89.0 | 89.5 | **89.7** | **89.9** | 89.7 |
| guj_Gujr | 91.4 | 70.2 | 90.7 | 91.2 | **92.1** | 91.6 | 91.5 | 90.1 | 38.3 | 91.3 | **91.7** | **91.7** | **91.9** | 91.1 |
| hin_Deva | **85.0** | 81.3 | 83.9 | 83.3 | 85.2 | 85.1 | 84.7 | 89.9 | 85.5 | 90.5 | **90.8** | **90.8** | **90.8** | 90.5 |
| kan_Knda | 84.2 | 58.9 | 83.7 | 84.7 | **85.1** | 84.3 | 84.9 | 81.6 | 36.7 | 81.7 | 82.0 | **84.0** | 83.2 | 83.4 |
| mal_Mlym | 89.4 | 82.4 | 89.7 | **90.2** | 90.1 | 89.4 | 90.0 | 87.2 | 79.4 | 87.9 | **88.3** | **88.5** | **88.8** | 88.5 |
| mar_Deva | 80.2 | 72.5 | 81.0 | **82.1** | 81.9 | 80.9 | 81.3 | 87.8 | 77.3 | 88.8 | 89.1 | 89.4 | **90.0** | 89.3 |
| npi_Deva | - | 57.0 | 84.6 | 83.5 | **86.8** | 85.1 | 85.0 | - | 57.6 | 91.0 | 90.9 | 91.4 | **92.2** | 91.4 |
| ory_Orya | 86.2 | 46.4 | **87.1** | **87.3** | 82.9 | 82.3 | 86.8 | 88.9 | 41.6 | 90.3 | **90.6** | 90.4 | 89.5 | 89.3 |
| pan_Guru | 88.2 | 67.4 | 88.3 | 88.8 | 88.8 | **89.1** | 88.6 | 88.6 | 72.7 | 89.6 | **90.1** | **90.2** | **90.2** | 89.2 |
| tam_Taml | 87.6 | 67.2 | 85.9 | 84.5 | **88.0** | 87.6 | 88.3 | 83.6 | 63.9 | 84.8 | **85.5** | 85.0 | **85.6** | 84.9 |
| tel_Telu | 88.1 | - | 84.2 | 83.0 | **89.6** | 89.0 | **89.6** | 85.8 | - | 87.3 | 88.0 | 87.7 | **88.5** | 87.8 |
| urd_Arab | - | 79.6 | 85.9 | 85.1 | 88.7 | **89.4** | 89.0 | - | 80.9 | 90.0 | 90.3 | 90.9 | **91.2** | 90.8 |

Table 41: COMET scores of all the systems on the FLORES-200 (Costa-jussà et al., 2022) Evaluation set in the En-Indic and Indic-En direction. The best performing system is bolded, while underlined results indicate significant performance difference where IT2 outperforms the system.

| | En-Indic | | | | | | | Indic-En | | | | | | |
|---|---|---|---|---|---|---|---|---|---|---|---|---|---|---|
| Language | IT1 | M100 | N1.2 | N54 | IT2 | Goog | Az | IT1 | M100 | N1.2 | N54 | IT2 | Goog | Az |
| asm_Beng | 79.4 | - | 82.2 | 81.5 | **83.7** | 82.6 | 82.6 | 81.0 | - | 85.4 | **86.6** | 85.8 | **86.4** | 83.5 |
| ben_Beng | 86.1 | 82.6 | 86.3 | 87.1 | **87.5** | 86.6 | 87.2 | 87.3 | 85.8 | 88.7 | 89.3 | 89.1 | **89.6** | 88.3 |
| guj_Gujr | 87.9 | 62.0 | 87.6 | 87.7 | **89.1** | 88.7 | 88.9 | 88.3 | 36.1 | 90.2 | 90.8 | 90.7 | **91.1** | 89.4 |
| hin_Deva | 80.6 | 77.6 | 81.1 | 81.2 | **81.5** | 81.3 | 80.9 | 88.4 | 87.6 | 89.8 | **90.3** | **90.3** | **90.4** | 89.5 |
| kan_Knda | 85.5 | 52.7 | 86.3 | 86.4 | **87.4** | 86.5 | **87.4** | 85.8 | 34.4 | 88.0 | **88.6** | 88.5 | **88.7** | 87.2 |
| mal_Mlym | 87.1 | 77.4 | 87.7 | 88.3 | **89.5** | 89.0 | 89.3 | 87.2 | 83.1 | 89.0 | 89.5 | 89.6 | **89.9** | 88.4 |
| mar_Deva | 73.7 | 67.7 | 74.7 | 75.6 | **76.4** | 75.9 | 76.2 | 86.4 | 81.4 | 88.4 | 89.0 | 88.8 | **89.3** | 87.9 |
| npi_Deva | - | 51.2 | 80.0 | 74.9 | **84.5** | 83.5 | 83.0 | - | 72.8 | 90.7 | 91.1 | 91.2 | **91.5** | 89.8 |
| ory_Orya | 83.6 | 38.8 | 84.4 | 84.2 | 79.9 | 80.9 | **84.8** | 86.8 | 38.5 | 89.1 | **89.9** | 89.8 | 89.6 | 87.9 |
| pan_Guru | 83.8 | 61.2 | 84.1 | 83.6 | 84.4 | **84.5** | 84.6 | 87.6 | 76.2 | 89.3 | **89.9** | 89.7 | **89.9** | 88.2 |
| tam_Taml | 88.0 | 44.6 | 89.1 | 88.6 | **89.9** | 89.5 | **89.9** | 85.1 | 64.3 | 87.4 | **88.0** | 87.7 | **88.2** | 86.2 |
| tel_Telu | 85.9 | - | 86.5 | 86.8 | **87.8** | 87.5 | **87.8** | 86.6 | - | 88.6 | **89.4** | 89.3 | **89.5** | 88.0 |
| urd_Arab | - | 73.4 | 82.2 | 81.8 | 82.6 | 83.0 | **83.6** | - | 79.0 | 87.5 | **88.3** | 87.7 | **88.4** | 86.4 |

Table 42: BLEU scores of all the systems on the IN22-Gen Evaluation set in the En-Indic and Indic-En direction. The best performing system is bolded, while underlined results indicate significant performance difference where IT2 outperforms the system. *Avg* means the average score of all the languages that system X supports. Δ represents the difference between the average scores of IT2 and the average scores of system X for the subset of languages that both X and IT2 support. A positive value for Δ indicates IT2 is better than X and vice-versa. † indicates completely off-target translations.

| | En-Indic | | | | | | | Indic-En | | | | | | |
|---|---|---|---|---|---|---|---|---|---|---|---|---|---|---|
| *Language* | IT1 | M100 | N1.2 | N54 | IT2 | Goog | Az | IT1 | M100 | N1.2 | N54 | IT2 | Goog | Az |
| *asm_Beng* | 9.9 | - | 13.9 | 15.4 | 19.4 | 16.9 | 16.2 | 32.5 | - | 40.4 | **44.6** | 43.1 | 42 | 35.4 |
| *ben_Beng* | 18.1 | 11.3 | 16.6 | 18.3 | 20.8 | 18.3 | 18 | 33.4 | 26.3 | 36.1 | 39.3 | 39 | **39.8** | 34.9 |
| *brx_Deva* | - | - | - | - | **16.9** | - | - | - | - | - | - | **40.2** | - | - |
| *doi_Deva* | - | - | - | - | **33.5** | 22.2 | - | - | - | - | - | **53.5** | 45.1 | - |
| *gom_Deva* | - | - | - | - | **18.8** | 11.6 | 11.5 | - | - | - | - | **35.3** | 33 | 25.8 |
| *guj_Gujr* | 17.9 | 3.9 | 18.7 | 20.3 | **25.7** | 23.3 | 21.2 | 36.3 | 0.4 | 40.2 | 43.4 | **43.7** | 43 | 37.1 |
| *hin_Deva* | 28.3 | 22.1 | 27.6 | 28.9 | **33.5** | 30.2 | 29.2 | 36.1 | 27.1 | 37.4 | 41 | **42.5** | 39.8 | 36.1 |
| *kan_Knda* | 13.4 | 1 | 13.4 | 14.9 | **18** | 14.2 | 15.2 | 34.8 | 0.1 | 39 | 42.7 | 40.8 | 41 | 35 |
| *kas_Arab* | - | - | 9.9 | 10.5 | **14.4** | - | - | - | - | 31.5 | 35 | **38.3** | - | - |
| *mai_Deva* | - | - | 15.5 | 15.1 | **19.3** | 9.3 | 14.5 | - | - | 37.9 | **41.8** | 40.8 | 39.8 | 36.2 |
| *mal_Mlym* | 13.9 | 4.4 | 11.9 | 13.1 | **16.4** | 13.7 | 13.6 | 31.4 | 17.5 | 34.8 | 38.6 | **41.4** | 37.9 | 33.3 |
| *mar_Deva* | 13.9 | 7 | 14.5 | 15.6 | **21.7** | 16.2 | 17.5 | 33.5 | 20 | 37.2 | 40.8 | 40.2 | **40.6** | 35.1 |
| *mni_Mtei* | - | - | - | - | **17.5** | 10.8 | - | - | - | - | - | **35.1** | 27.5 | - |
| *npi_Deva* | - | 2.6 | 14.4 | 14.8 | 16.8 | 13.8 | 14.4 | - | 12.8 | 42.2 | 46 | 45.1 | **46.8** | 39.9 |
| *ory_Orya* | 10.2 | 0.1 | 12.2 | 11.8 | **14.5** | 10.7 | **14.1** | 36.7 | 0 | 40.7 | **44.7** | 43.8 | 40.4 | 34.7 |
| *pan_Guru* | 23.5 | 7.2 | 23.9 | 25.3 | 25.5 | **29.9** | 25.2 | 33.5 | 10.4 | 37.4 | 40.6 | **41.4** | 39.6 | 34.7 |
| *san_Deva* | - | - | 3.7 | 4.3 | **11.1** | 5.5 | - | - | - | 24.2 | 27.2 | **29.8** | 28.6 | - |
| *sat_Olck* | - | - | 0.0 † | 3.8 | **5.5** | - | - | - | - | 12.3 | 18.7 | **21.8** | - | - |
| *snd_Deva* | - | - | - | - | **14** | - | - | - | - | - | - | **35** | - | - |
| *tam_Taml* | 11.9 | 1.4 | 12.6 | 13 | **14.4** | 14 | 14.5 | 28.9 | 4.9 | 32.5 | 35 | **35.9** | 34.9 | 29.4 |
| *tel_Telu* | 15.5 | - | 15.1 | 17.1 | **19.4** | 17.7 | 17.7 | 33.5 | - | 37.6 | 41.5 | **42.3** | 41.3 | 35.7 |
| *urd_Arab* | - | 23.1 | 42 | 43.8 | 49.7 | 44.1 | **51.4** | - | 26.5 | 46.5 | 50.5 | **53.7** | 50.9 | 46.3 |
| *Avg.* | 16 | 7.6 | 15.6 | 16.8 | 20.3 | 17.9 | 19.6 | 33.7 | 13.3 | 35.8 | 39.5 | 40.1 | 39.6 | 35.3 |
| Δ | 7.3 | 15.8 | 4.8 | 1.7 | - | 4.4 | 2.7 | 8.6 | 29.2 | 4.4 | 0.9 | - | 2.3 | 6.6 |

Table 43: BLEU scores of all the systems on the IN22-Conv Evaluation set in the En-Indic and Indic-En direction. The best performing system is bolded, while underlined results indicate significant performance difference where IT2 outperforms the system. *Avg* means the average score of all the languages that system X supports. $\Delta$ represents the difference between the average scores of IT2 and the average scores of system X for the subset of languages that both X and IT2 support. A positive value for $\Delta$ indicates IT2 is better than X and vice-versa. † indicates completely off-target translations.

| | En-Indic | | | | | | | Indic-En | | | | | | |
|---|---|---|---|---|---|---|---|---|---|---|---|---|---|---|
| *Language* | IT1 | M100 | N1.2 | N54 | IT2 | Goog | Az | IT1 | M100 | N1.2 | N54 | IT2 | Goog | Az |
| *asm_Beng* | 11.6 | - | 16.7 | 17.8 | **19.7** | 17.6 | **19.5** | 31.3 | - | 38.6 | 40.4 | **43.8** | **44.6** | 41.7 |
| *ben_Beng* | 20.1 | 13 | 19.3 | 20.7 | 21.3 | **21.5** | 20.3 | 32.9 | 25.8 | 33.3 | 35 | 36.4 | **37.6** | 36 |
| *brx_Deva* | - | - | - | - | **15.4** | - | - | - | - | - | - | **35.5** | - | - |
| *doi_Deva* | - | - | - | - | **32.4** | 17.6 | - | - | - | - | - | **45.6** | 42.6 | - |
| *gom_Deva* | - | - | - | - | **14.2** | 12 | 10.4 | - | - | - | - | **29.9** | 29.5 | 23.7 |
| *guj_Gujr* | 23.2 | 4 | 22.8 | 24.1 | **27.2** | 26.7 | 25.7 | 34.7 | 0.3 | 39.7 | 39.9 | **41.1** | **41** | 39.1 |
| *hin_Deva* | 28.4 | 22.3 | 27.1 | 28.4 | 30.1 | **30.7** | 28 | 35.5 | 28.3 | 37.3 | 38.4 | **39.3** | 38.3 | 37.7 |
| *kan_Knda* | 6.1 | 0.5 | 5.8 | **6.5** | **6.7** | 6.3 | 6.2 | 21.1 | 0.2 | 22.5 | 22.9 | **24.9** | 24.4 | 23.6 |
| *kas_Arab* | - | - | 4.5 | 4.6 | **11.3** | - | - | - | - | 23.3 | 24.1 | **31.8** | - | - |
| *mai_Deva* | - | - | 15.4 | 15.7 | **18.9** | 10.6 | 11.9 | - | - | 32.6 | 33.8 | 35.3 | **36.6** | 32.1 |
| *mal_Mlym* | **11.1** | 3.9 | 8.3 | 7.6 | **11.3** | 11.1 | 10.8 | 27.6 | 16.2 | 28 | 29.5 | **31.6** | 31.1 | 30.8 |
| *mar_Deva* | 15.5 | 8.9 | 16.9 | 18.6 | **19.4** | 17.7 | 17.6 | 32.2 | 18.9 | 34.1 | 35.7 | 36.7 | **37.7** | 35.9 |
| *mni_Mtei* | - | - | - | - | **14.2** | 6.9 | - | - | - | - | - | **31.9** | 25.7 | - |
| *npi_Deva* | - | 1.6 | 15.7 | 16.4 | **21.2** | 16.4 | 15.8 | - | 3.5 | 38.9 | 39.6 | **42.4** | **43.1** | 40.8 |
| *ory_Orya* | 11.3 | 0.3 | **13.8** | **13.8** | 12.3 | 10.3 | **14.1** | 33.6 | 0.2 | 38.4 | **38.9** | 38.8 | 37.4 | 35.3 |
| *pan_Guru* | 32 | 7 | 32.1 | 33.8 | 35.7 | **41.3** | 33.2 | 36.8 | 7.3 | 39.6 | 41.1 | **43** | 39.5 | 40.7 |
| *san_Deva* | - | - | 2.8 | 4.7 | **6.3** | 5.2 | - | - | - | 17.8 | 17.2 | **26.1** | 26.7 | - |
| *sat_Olck* | - | - | 0.0 † | 3 | **6.6** | - | - | - | - | 113 | 17.8 | **23.1** | - | - |
| *snd_Deva* | - | - | - | - | **7.4** | - | - | - | - | - | - | **27.5** | - | - |
| *tam_Taml* | 7.7 | 1.5 | 7.1 | 7.4 | 7.6 | **8** | 8.4 | 20.8 | 4 | 23 | 24.1 | 22.7 | 23.3 | 22.8 |
| *tel_Telu* | 12 | - | 9.8 | 10.5 | **14.1** | 13.4 | 13.8 | 26.3 | - | 29.5 | 31.6 | 31 | 31.5 | 31.1 |
| *urd_Arab* | - | 19.4 | 35.6 | 35.3 | **43.7** | 42.2 | 40.1 | - | 26.5 | 40.3 | 41.7 | **45.9** | 45.6 | 44.9 |
| *Avg.* | 16.3 | 7.5 | 14.9 | 15.8 | 18 | 17.5 | 18.4 | 30.3 | 11.9 | 31.1 | 32.5 | 34.7 | 35.3 | 34.4 |
| $\Delta$ | 4.5 | 14 | 3.5 | 2.8 | - | 2.7 | 1.8 | 6 | 24.7 | 3.8 | 3.4 | - | 0.9 | 1.8 |

Table 44: BLEU scores of all the systems on the FLORES-200 (Costa-jussà et al., 2022) devtest set in the En-Indic and Indic-En direction. The best performing system is bolded, while underlined results indicate significant performance difference where IT2 outperforms the system. *Avg* means the average score of all the languages that system X supports. Δ represents the difference between the average scores of IT2 and the average scores of system X for the subset of languages that both X and IT2 support. A positive value for Δ indicates IT2 is better than X and vice-versa. † indicates completely off-target translations.

| | En-Indic | | | | | | | Indic-En | | | | | | |
|---|---|---|---|---|---|---|---|---|---|---|---|---|---|---|
| *Language* | IT1 | M100 | N1.2 | N54 | IT2 | Goog | Az | IT1 | M100 | N1.2 | N54 | IT2 | Goog | Az |
| *asm_Beng* | 7.6 | - | 11.4 | 11.7 | **14** | 12.2 | **13.8** | 23.4 | - | 31.3 | **33.9** | 32.5 | 32.9 | 27.1 |
| *ben_Beng* | 19.7 | 15.3 | 20.2 | 22.1 | **24.7** | 24.3 | 23.4 | 31.8 | 29.4 | 36.6 | 38.7 | 38.6 | **39.7** | 35.3 |
| *guj_Gujr* | 22.1 | 4.8 | 23.9 | 25.2 | **27.8** | 27.1 | 26.6 | 34.1 | 1.2 | 42.5 | 44.6 | 45.3 | **46.2** | 38.6 |
| *hin_Deva* | 34.5 | 30.9 | 34.3 | 36.7 | 38.6 | **39** | 38.4 | 37.5 | 35.6 | 42.1 | 44.4 | **46.1** | 46.4 | 43.1 |
| *kan_Knda* | 18.3 | 1.6 | 20.6 | 22.1 | 24.1 | **24.6** | 24.2 | 28.7 | 0.7 | 34.8 | 36.9 | **37.8** | 38.4 | 32.5 |
| *kas_Arab* | - | - | 10 | 10.5 | **11.9** | - | - | - | - | 33.7 | **36.7** | 36.1 | - | - |
| *kas_Deva* | - | - | 1.9 | **2** | **2.2** | - | - | - | - | 23.9 | **27** | 25.1 | - | - |
| *mai_Deva* | - | - | 16.5 | 18.2 | 19 | 11.8 | **20.8** | - | - | 44.1 | 46.7 | **48.2** | 46.6 | 41.8 |
| *mal_Mlym* | 15.9 | 7.9 | 14.1 | 18.3 | **22** | 22.4 | 22 | 31.4 | 25.3 | 37.6 | 39.1 | **41** | 41 | 35.8 |
| *mar_Deva* | 15.8 | 10.1 | 16.2 | 17.9 | 19.9 | 20.7 | 18.3 | 31 | 24.6 | 37.1 | 40.3 | 41.1 | **42.1** | 37.3 |
| *mni_Beng* | - | - | 7.7 | **10.4** | 8.6 | - | - | - | - | 27 | 27.5 | **28.5** | - | - |
| *npi_Deva* | - | 1.7 | 18.7 | 18.5 | 25.5 | 23.9 | 20.9 | - | 14 | 42.3 | 44.5 | 46.3 | **46.5** | 39.8 |
| *ory_Orya* | 13.6 | 0.3 | 17.1 | 16.9 | 17.3 | **24.4** | 18.6 | 29.8 | 0.5 | 38.2 | 41.6 | 42.4 | 41.6 | 35.1 |
| *pan_Guru* | 26.7 | 8.6 | 27.1 | 27.7 | 29.6 | **31.1** | 30.1 | 35.8 | 15.2 | 42.2 | 44.8 | 44.9 | **45.8** | 38.2 |
| *san_Deva* | - | - | 2.2 | 2.3 | **3.2** | 3.4 | - | - | - | 23.3 | 26.1 | **26.6** | 25 | - |
| *sat_Olck* | - | - | 0.1 † | **4.9** | 4.1 | - | - | - | - | 14.5 | **21.7** | 16.7 | - | - |
| *snd_Arab* | - | 10.8 | 25.3 | 26.4 | 20.2 | 27.3 | 27.7 | - | 2.7 | 42 | **45** | 43.6 | **45.5** | 36.3 |
| *tam_Taml* | 15.6 | 0.9 | 18.6 | 19.8 | **22.6** | 21.1 | 21.3 | 28.4 | 8.3 | 34.4 | 36.8 | **37.8** | 37.7 | 31.1 |
| *tel_Telu* | 21.3 | - | 23.1 | 25.3 | **27.8** | 27.2 | 25.3 | 33.4 | - | 40.9 | 43.6 | **44.7** | 45.1 | 39.6 |
| *urd_Arab* | - | 16.9 | 25.8 | 27.2 | **29.1** | 28.2 | 28.2 | - | 22.2 | 36.8 | **39.6** | 38.1 | **40** | 34.4 |
| *Avg.* | 19.2 | 9.2 | 16.7 | 18.2 | 19.6 | 23.0 | 24 | 31.4 | 15 | 35.3 | 38 | 38.1 | 41.3 | 36.4 |
| Δ | 5.2 | 15.9 | 2.9 | 1.4 | - | -0.2 | 0.1 | 9.7 | 26.9 | 2.8 | 0.1 | - | -0.3 | 5.5 |

## C    Human Evaluation

Automated evaluation metrics provide a convenient and quick way to evaluate MT systems. However, as reported by previous works (Kocmi et al., 2021; Moghe et al., 2022), the degree of correlation between automatic evaluation metrics and human ratings is not particularly strong. To obtain a more comprehensive understanding of the model's performance, it is imperative to conduct human evaluations (Kocmi et al., 2021).

We conduct a small-scale human evaluation exercise to verify if the quality of our model outputs correlates with the improvements observed using automatic metrics. This exercise focused on the En-Indic direction and included 50 examples each from the Wikipedia and Web sources subset to yield a total of 100 sentence pairs from IN22-Gen. We seek to study human evaluation of sentences of diverse lengths (refer Figure 10) and uniformly sample sentences from each bucket. Our human evaluators belong to the same pool of translators who created the IN22 benchmark. They are fluent speakers of English and the respective native language under study. Based on the availability of annotators, we conduct human evaluation studies for the following languages: Assamese, Bengali, Bodo, Dogri, Konkani, Gujarati, Hindi, Kannada, Malayalam, Marathi, Nepali, Punjabi, Santali, Tamil, Telugu and Urdu. We compare IndicTrans2 model outputs along with those of NLLB (Costa-jussà et al., 2022), Google Translate, and Azure Translate. The annotators were not specifically aware of which output was generated by which system.

We use the XSTS methodology proposed by Licht et al. (2022) and adopted by Costa-jussà et al. (2022) for comparing different multilingual machine translation (MT) systems. XSTS relies on human raters to assess translations without using reference translations, focusing more on adequacy (meaning preservation) than fluency. This approach is particularly suitable for low-resource languages with relatively lower translation quality. XSTS also exhibits better inter-annotator agreement than Direct Assessment (Graham et al., 2013) as demonstrated by prior research Licht et al. (2022).

Brief instructions for human annotations are provided below. Raters choose scores between 1 to 5. We refer the readers to Figure 1 in Licht et al. (2022) for the detailed definition of the scores.

- Score of 1 indicates the sentences are unrelated to each other or maybe in similar topics but differ in more than half of their core concepts.

- Score of 2 indicates that the sentences are about similar topics but some key details about the main subject, verb, or object are either different or absent.

- Score of 3 indicates that the sentences are equivalent to each other but with unimportant differences.

- Score of 4 indicates that the sentences are paraphrases of each other but have minor differences in emphasis, formality, idioms, etc.

- Score of 5 indicates the sentences mean the same with no difference in emphasis, formality, idioms, etc.

It is known that there is some variance in human evaluators, with some being overly critical while others being excessively generous when assessing MT outputs. Recent studies by Licht et al. (2022) and Costa-jussà et al. (2022) emphasize the importance of having a calibration set to ensure that XSTS scores are comparable across languages. To address this concern, our evaluation methodology employs a sample of the calibration set, comprising pairs of English sentences released by NLLB Team (Costa-jussà et al., 2022). From each of the 5 scoring classes described in Licht et al. (2022), we uniformly sample 10 sentences, forming a calibration set with 50 sentence pairs. The task framework employed for this purpose closely aligns with the approach suggested in Costa-jussà et al. (2022). To account for extreme calibration shifts, we use the *moderated calibration* adjustment as proposed in Costa-jussà et al. (2022).

**Overall results.** Our findings indicate that IndicTrans2 outperforms Google and NLLB 54B significantly, and performs comparably with Azure. Statistical significance is computed using ANOVA with posthoc Tukey HSD test ($p \leq 0.05$) following similar human evaluation in data-to-text generation (Puduppully & Lapata, 2021; Puduppully et al., 2022). However, it should be acknowledged that the sample size of sentences used for human evaluation is limited, and therefore, these results must be interpreted with caution. Future work should expand the human evaluation to cover all 22 Indic languages and also include IN22-Conv set to gain more fine-grained insights.

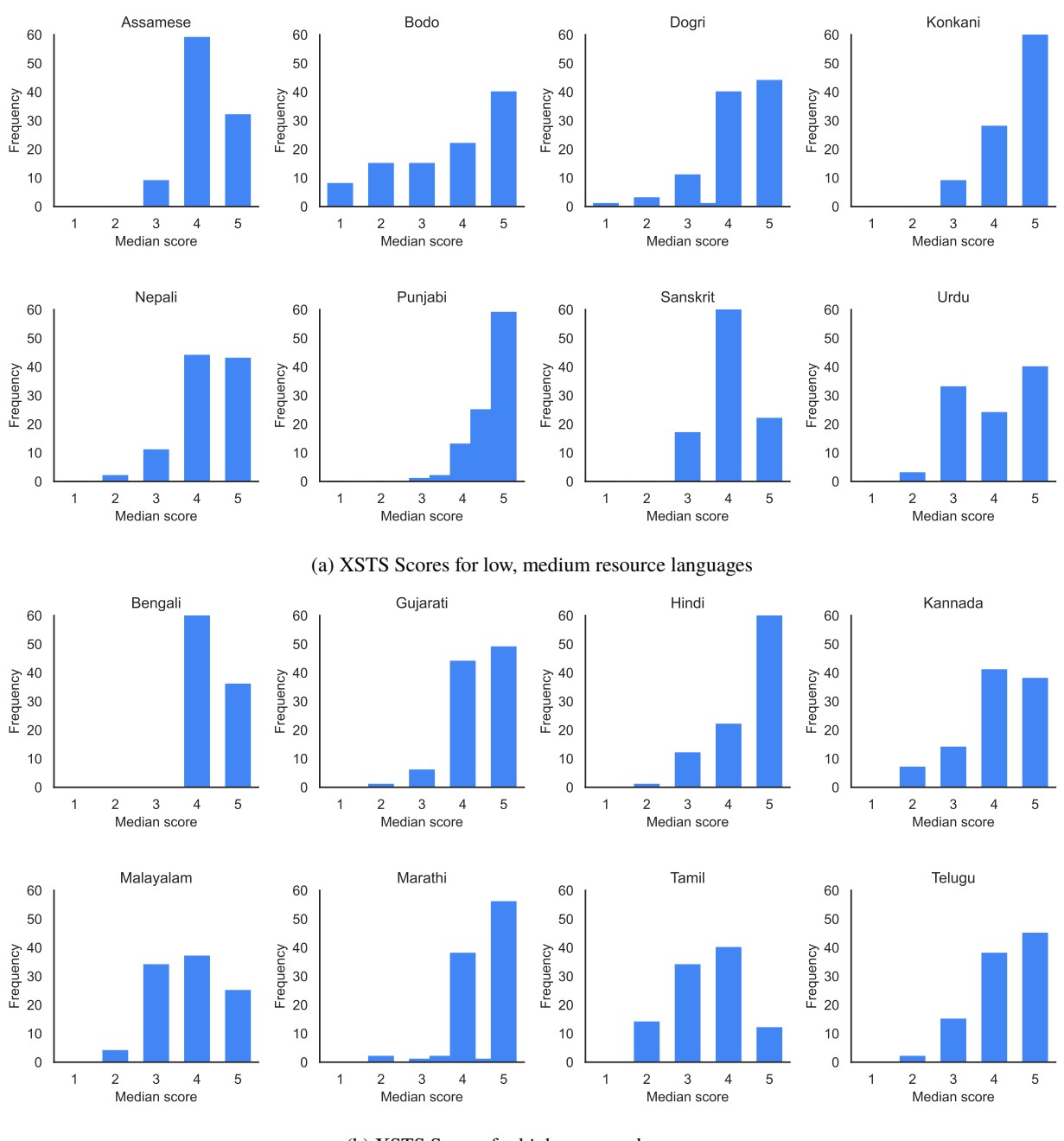

(a) XSTS Scores for low, medium resource languages

(b) XSTS Scores for high resource languages

Figure 9: Distribution of XSTS scores for low, medium and high resource languages in IN22

**High vs. Low Resource Languages.** Figure 9 in Appendix C depicts the trends in the distribution of ratings for a selected set of languages, with low and medium resource languages in the upper half, and high resource languages in the lower half. IndicTrans2 outperforms other models significantly in low-resource languages like Konkani, Sanskrit, and Nepali. Most languages supported by IndicTrans2 achieve close to a 4 XSTS rating. High-resource languages, such as Hindi, Bengali, and Telugu, show a right-skewed distribution with many sentence pairs receiving higher ratings. On the other hand, medium-performance languages like Bodo exhibit a more symmetrical distribution around the rating.

Table 45: Post calibration results for human evaluation for En-XX language pairs using XSTS methodology. We compare between four model outputs: Azure, Google, NLLB (N54B) (Costa-jussà et al., 2022), and IndicTrans2 (IT2). – indicates languages not supported by a model. The $^\dagger$ after a value indicates statistically significant difference from IndicTrans2 using ANOVA with post-hoc Tukey HSD test ($p \leq 0.05$). $\Delta$ represents the difference of pre-calibration and post-calibration XSTS score for IndicTrans2, with a positive value indicating improvement in scores post-calibration and vice-versa. Overall, IndicTrans2 is the top-ranked system comparable with Azure, and significantly better than Google and NLLB.

| language | Azure | Google | N54B | IT2 | $\Delta$ |
|---|---|---|---|---|---|
| asm_Beng | 3.44 | 3.63 | **3.83** | 3.62 | -0.61 |
| ben_Beng | 3.94$^\dagger$ | 3.92$^\dagger$ | 4.05 | **4.18** | -0.18 |
| brx_Deva | – | – | – | **3.75** | 0.04 |
| doi_Deva | – | 4.13 | – | **4.32** | 0.08 |
| gom_Deva | – | 3.84$^\dagger$ | – | **4.45** | -0.09 |
| guj_Gujr | 4.50 | 4.26$^\dagger$ | 4.28$^\dagger$ | **4.53** | 0.12 |
| hin_Deva | 4.27$^\dagger$ | 4.23$^\dagger$ | 4.40 | **4.56** | 0.05 |
| kan_Knda | 4.12 | 3.86 | 4.01 | **4.18** | 0.08 |
| mal_Mlym | 3.96 | 3.73$^\dagger$ | 3.84 | **4.06** | 0.23 |
| mar_Deva | 4.27 | 3.89$^\dagger$ | 4.12 | **4.41** | -0.11 |
| npi_Deva | 3.89$^\dagger$ | 3.87$^\dagger$ | 3.81$^\dagger$ | **4.41** | 0.13 |
| pan_Guru | 4.09 | 3.94 | 4.10 | **4.25** | -0.45 |
| san_Deva | – | 2.87$^\dagger$ | 2.83$^\dagger$ | **3.68** | -0.37 |
| tam_Taml | **4.00** | 3.79 | 3.79 | 3.90 | 0.40 |
| tel_Telu | **4.29** | 3.94$^\dagger$ | 4.24 | **4.29** | 0.03 |
| urd_Arab | 4.06 | 3.68$^\dagger$ | 3.76$^\dagger$ | **4.25** | 0.24 |
| Average | 4.07 | 3.84$^\dagger$ | 3.93$^\dagger$ | **4.18** | -0.02 |

**Calibration.** The column $\Delta$ in Table 45 indicates the revision in scores post-calibration for IndicTrans2, with a positive value indicating improvement in scores and vice-versa. We present the results comparing pre and post-calibration procedures for all the models in Table 46 in Appendix C. We see that scores of languages get adjusted. Assamese and Bengali are two related languages written using the same script and sharing substantial vocabulary. At the same time, Bengali is high-resource in comparison to Assamese; Bengali belongs to class 5 whereas Assamese belongs to class 2 in terms of the language resourcefulness classification (Joshi et al., 2020). From Table 46, we see that the scores for Assamese and Bengali are comparable pre-calibration; however, after calibration, the scores for Assamese drop compared to that of Bengali. Among languages for which the scores change by more than 0.2 points, Punjabi and Sanskrit scores drop post calibration whereas Malayalam, Tamil, and Urdu scores improve. These findings underscore the significance of calibration in ensuring the reliability and comparability of XSTS scores across different languages and models. Overall, we see an average change of 0.23 in XSTS scores for IndicTrans2. Importantly, the relative ranking of the machine translation models based on XSTS scores remains unchanged, with IndicTrans2 outperforming NLLB, and Google, and comparable with Azure.

**Correlation with Automatic Metrics.** The correlation between XSTS scores and automatic metrics is an important aspect of evaluating machine translation performance. Our analysis reveals that XSTS scores for IndicTrans2 exhibit moderate correlation with two widely used automatic metrics, namely BLEU, and chrF++. Specifically, we observe Spearman rank correlations of 0.49 and 0.12, respectively, with BLEU and chrF++ across all languages, but the correlations increase to 0.67 and 0.25, respectively, when Urdu is excluded from the analysis. This observation can be partly attributed to the influence of Urdu tokenization, which had a greater impact on the BLEU and chrF++ scores when compared to other languages. This can be due to the higher fertility of Urdu when using the UrduHack tokenizer, which led to inflated scores for both metrics. As a result, the correlation was reduced between these metrics and the actual quality of translations, deviating from the trend observed in other languages. In contrast, we find no correlation between XSTS scores and the COMET metric, which is designed to assess the fluency and adequacy of machine translations. Additionally, we observe no correlation between BLEU/chrF++ and COMET scores, indicating that these

Table 46: Comparison of XSTS score before and after applying calibration

| | Pre-Calibration | | | | Post-Calibration | | | |
|---|---|---|---|---|---|---|---|---|
| *language* | Azure | Google | N54B | IT2 | Azure | Google | N54B | IT2 |
| *asm_Beng* | 4.05 | 4.24 | **4.44** | 4.23 | 3.44 | 3.63 | **3.83** | 3.62 |
| *ben_Beng* | 4.13 | 4.11 | 4.24 | **4.36** | 3.94 | 3.92 | 4.05 | **4.18** |
| *brx_Deva* | - | - | - | **3.71** | - | - | - | **3.75** |
| *doi_Deva* | - | 4.03 | - | **4.24** | - | 4.13 | - | **4.32** |
| *gom_Deva* | - | 3.93 | - | **4.54** | - | 3.84 | - | **4.45** |
| *guj_Gujr* | 4.37 | 4.10 | 4.12 | **4.41** | 4.50 | 4.26 | 4.28 | **4.53** |
| *hin_Deva* | 4.20 | 4.15 | 4.34 | **4.51** | 4.27 | 4.23 | 4.40 | **4.56** |
| *kan_Knda* | 4.04 | 3.77 | 3.92 | **4.10** | 4.12 | 3.86 | 4.01 | **4.18** |
| *mal_Mlym* | 3.72 | 3.47 | 3.59 | **3.83** | 3.96 | 3.73 | 3.84 | **4.06** |
| *mar_Deva* | 4.38 | 4.00 | 4.23 | **4.52** | 4.27 | 3.89 | 4.12 | **4.41** |
| *npi_Deva* | 3.71 | 3.69 | 3.63 | **4.28** | 3.89 | 3.87 | 3.81 | **4.41** |
| *pan_Guru* | 4.54 | 4.39 | 4.54 | **4.70** | 4.09 | 3.94 | 4.10 | **4.25** |
| *san_Deva* | - | 3.23 | 3.19 | **4.05** | - | 2.87 | 2.83 | **3.68** |
| *tam_Taml* | **3.61** | 3.39 | 3.39 | 3.50 | **4.00** | 3.79 | 3.79 | 3.90 |
| *tel_Telu* | **4.26** | 3.90 | 4.21 | **4.26** | **4.29** | 3.94 | 4.24 | **4.29** |
| *urd_Arab* | 3.80 | 3.39 | 3.47 | **4.01** | 4.06 | 3.68 | 3.76 | **4.25** |
| *Average* | 4.07 | 3.85 | 3.95 | **4.20** | 4.07 | 3.84 | 3.93 | **4.18** |

metrics capture different aspects of machine translation quality. Nonetheless, further investigation is necessary to gain a deeper understanding of the relationship between metrics.

## D   Distilled Models

This section presents a detailed description of our student model architecture and distillation training hyperparameters. We share the weight of the decoder embedding and output projection to compress the student models as much as possible. This also allows us to have equal-sized student models for both directions. This is particularly useful for the En-Indic model, as the output projection is a significant fraction of the model parameters ($\approx$ 60M). Tables 50 and 51 present the comparison between the student and teacher models on FLORES200 and IN22-Conv respectively.

Table 47: Student architecture description. Specifically, our student models use the *base18L* architecture, following Gumma et al. (2023).

| Hyperaparameter | Value |
|---|---|
| Model dim | 512 |
| FFN dim | 2048 |
| Encoder Layers | 18 |
| Decoder Layers | 18 |
| Activation | GELU (Hendrycks & Gimpel, 2016) |
| Pre-Normalization | True (Xiong et al., 2020) |
| Embedding LayerNorm | True |
| Share decoder input output embed | True |

Table 48: Number of parameters in teacher and distilled student models.

| #Params | Indic-En | En-Indic |
|---|---|---|
| Teacher | 1.02B | 1.11B |
| Student | 211.77M | 211.77M |

Table 49: Hyperparameter set for Knowledge Distillation. The rest of the parameters not mentioned in the table are the same as the ones used for training IT2 (see Table 10).

| Hyperparameters | Stage 1 Distillation | Stage 2 fine-tuning |
|---|---|---|
| Learning rate | $7e$-4 (`en-xx`), $1e$-3 (`xx-en`) | $3e$-5 |
| Criterion | KL-Divergence | Cross-entropy |
| Label smoothing (Szegedy et al., 2016) | — | 0.1 |
| Effective batch size | 262K | 8K |
| Checkpoint metric | BLEU @ beam = 5 | BLEU @ beam = 5 |

Table 50: chrF++ scores of Indic-En and En-Indic distilled models on FLORES-200. Distilled (*Dist*) is the model trained with Word-level KD. Δ is the difference between the distilled Model fine-tuned on seed data (*Dist-Seed*) & IT2. Higher values of Δ are preferable.

| | Indic-En | | | | En-Indic | | | |
|---|---|---|---|---|---|---|---|---|
| *language* | IT2 | Dist | Dist-Seed | Δ | IT2 | Dist | Dist-Seed | Δ |
| *asm_Beng* | 56.9 | 56.9 | 56.1 | -0.8 | 43.3 | 42.7 | 43.0 | -0.3 |
| *ben_Beng* | 62.4 | 61.4 | 61.4 | -1.0 | 54.3 | 54.0 | 54.0 | -0.3 |
| *guj_Gujr* | 67.0 | 65.5 | 65.6 | -1.4 | 56.0 | 55.9 | 55.8 | -0.2 |
| *hin_Deva* | 67.5 | 66.0 | 66.0 | -1.5 | 59.6 | 59.3 | 59.3 | -0.3 |
| *kan_Knda* | 61.5 | 60.0 | 60.2 | -1.3 | 56.1 | 56.0 | 55.9 | -0.2 |
| *kas_Arab* | 59.7 | 57.6 | 57.6 | -2.1 | 39.7 | 40.0 | 40.1 | 0.4 |
| *kas_Deva* | 48.3 | 45.3 | 45.6 | -2.7 | 19.2 | 19.3 | 19.8 | 0.6 |
| *mai_Deva* | 69.5 | 67.3 | 67.3 | -2.2 | 50.5 | 50.8 | 51.0 | 0.5 |
| *mal_Mlym* | 64.3 | 62.7 | 62.8 | -1.5 | 57.3 | 57.0 | 57.1 | -0.2 |
| *mar_Deva* | 64.3 | 63.1 | 63.1 | -1.2 | 51.3 | 51.3 | 51.1 | -0.2 |
| *mni_Beng* | 52.9 | 51.0 | 50.9 | -2.0 | 38.2 | 37.5 | 37.2 | -1.0 |
| *npi_Deva* | 68.1 | 66.2 | 66.2 | -1.9 | 57.2 | 57.1 | 57.2 | 0.0 |
| *ory_Orya* | 64.9 | 63.2 | 63.2 | -1.7 | 49.2 | 48.6 | 48.7 | -0.5 |
| *pan_Guru* | 66.4 | 64.8 | 65.0 | -1.4 | 53.5 | 53.5 | 53.5 | 0.0 |
| *san_Deva* | 51.6 | 49.9 | 49.9 | -1.7 | 31.6 | 31.5 | 31.3 | -0.3 |
| *sat_Olck* | 39.3 | 40.5 | 40.9 | 1.6 | 28.4 | 28.2 | 28.6 | 0.2 |
| *snd_Arab* | 65.1 | 63.4 | 63.5 | -1.6 | 44.9 | 45.1 | 45.0 | 0.1 |
| *tam_Taml* | 61.3 | 59.4 | 59.3 | -2.0 | 57.2 | 57.0 | 57.0 | -0.2 |
| *tel_Telu* | 66.1 | 64.9 | 64.8 | -1.3 | 59.4 | 59.4 | 59.5 | 0.1 |
| *urd_Arab* | 62.0 | 60.6 | 60.7 | -1.3 | 52.2 | 52.0 | 52.2 | 0.0 |
| *Average* | 61.0 | 59.4 | 59.5 | -1.5 | 48.0 | 47.8 | 47.9 | -0.1 |

Table 51: chrF++ scores of Indic-En and En-Indic distilled models on IN22-Conv. Distilled (*Dist*) is the model trained with Word-level KD. Δ is the difference between the distilled Model fine-tuned on seed data (*Dist-Seed*) & IT2. Higher values of Δ are preferable.

| language | Indic-En | | | | En-Indic | | | |
|---|---|---|---|---|---|---|---|---|
| | IT2 | Dist | Dist-Seed | Δ | IT2 | Dist | Dist-Seed | Δ |
| *asm_Beng* | 62.9 | 62.3 | 63.0 | 0.1 | 46.8 | 46.1 | 46.6 | -0.2 |
| *ben_Beng* | 58.4 | 58.3 | 58.7 | 0.3 | 49.7 | 49.6 | 49.8 | 0.1 |
| *brx_Deva* | 56.3 | 55.2 | 55.2 | -1.1 | 45.3 | 45.4 | 45.4 | 0.1 |
| *doi_Deva* | 65.0 | 63.8 | 63.5 | -1.5 | 53.9 | 52.3 | 53.0 | -0.9 |
| *gom_Deva* | 51.7 | 50.8 | 50.8 | -0.9 | 42.5 | 41.8 | 41.7 | -0.8 |
| *guj_Gujr* | 62.0 | 61.5 | 62.0 | 0.0 | 53.1 | 53.1 | 53.1 | 0.0 |
| *hin_Deva* | 60.1 | 59.9 | 60.3 | 0.2 | 49.6 | 49.3 | 49.4 | -0.2 |
| *kan_Knda* | 47.5 | 48.3 | 48.6 | 1.1 | 33.8 | 33.6 | 33.8 | 0.0 |
| *kas_Arab* | 52.6 | 50.2 | 50.5 | -2.1 | 35.6 | 33.6 | 34.9 | -0.7 |
| *mai_Deva* | 57.8 | 56.9 | 57.2 | -0.6 | 44.3 | 43.8 | 44.3 | 0.0 |
| *mal_Mlym* | 54.3 | 53.8 | 53.9 | -0.4 | 45.7 | 45.5 | 45.6 | -0.1 |
| *mar_Deva* | 58.5 | 58.4 | 58.8 | 0.3 | 48.6 | 48.4 | 48.7 | 0.1 |
| *mni_Mtei* | 52.5 | 51.4 | 51.6 | -0.9 | 40.2 | 39.5 | 40.0 | -0.2 |
| *npi_Deva* | 63.0 | 63.1 | 63.3 | 0.3 | 51.5 | 51.1 | 51.3 | -0.2 |
| *ory_Orya* | 60.3 | 60.3 | 60.7 | 0.4 | 40.2 | 39.8 | 40.0 | -0.2 |
| *pan_Guru* | 62.7 | 61.7 | 62.1 | -0.6 | 57.8 | 57.6 | 57.5 | -0.3 |
| *san_Deva* | 48.3 | 46.9 | 46.9 | -1.4 | 35.5 | 34.6 | 34.8 | -0.7 |
| *sat_Olck* | 43.5 | 45.9 | 46.3 | 2.8 | 34.6 | 34.2 | 34.8 | 0.2 |
| *snd_Deva* | 49.6 | 50.1 | 50.5 | 0.9 | 30.3 | 30.1 | 30.0 | -0.3 |
| *tam_Taml* | 45.8 | 45.7 | 45.7 | -0.1 | 39.1 | 38.6 | 38.7 | -0.4 |
| *tel_Telu* | 52.9 | 52.3 | 53.0 | 0.1 | 45.5 | 44.7 | 45.1 | -0.4 |
| *urd_Arab* | 65.5 | 64.4 | 64.5 | -1.0 | 61.6 | 61.5 | 61.4 | -0.2 |
| *Average* | 56.0 | 55.5 | 55.8 | -0.2 | 44.8 | 44.3 | 44.5 | -0.3 |

# E  Additional details about IN22 Benchmark

This section provides a detailed overview of the source and domain diversity of the different subsets of the IN22 benchmark.

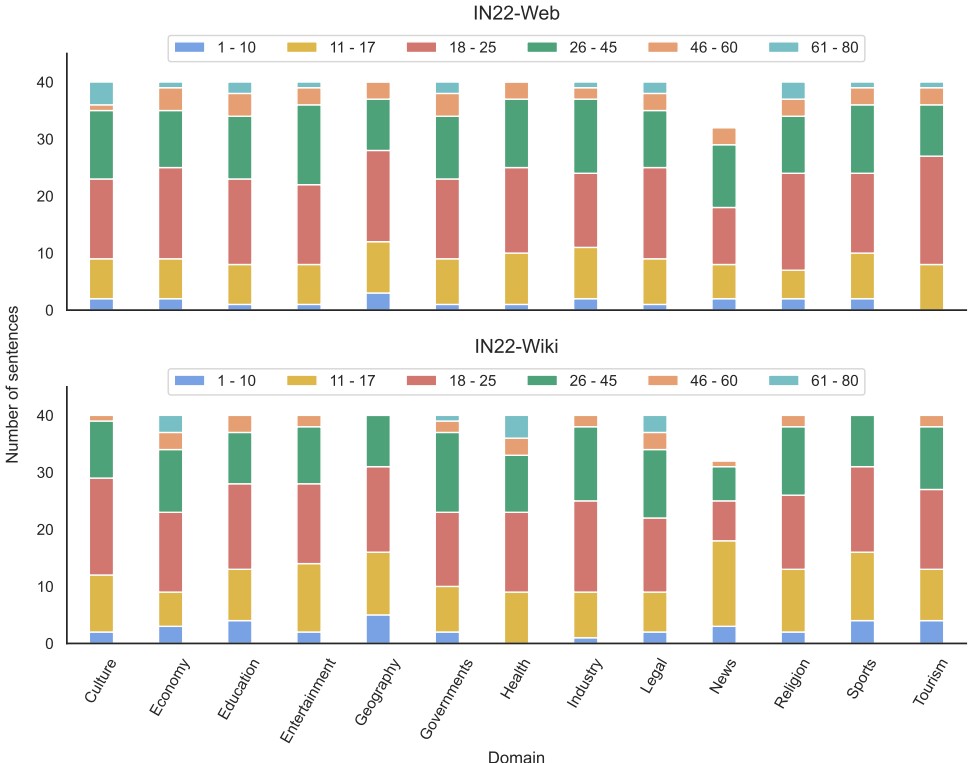

Figure 10: Domain vs. length distribution for the sentences from Web Sources (top) and Wikipedia (bottom) subsets of IN22

Table 52: Comparison of diversity of domains in FLORES-200 and IN22

| FLORES domain | IN22 domain |
|---|---|
| crime, disasters, politics | news |
| entertainment | entertainment |
| geography | geography |
| health | health |
| nature, science | education |
| sports | sports |
| travel | tourism |
| - | culture |
| politics | government |
| - | industry |
| - | economy |
| - | legal |
| - | religion |

Table 53: Statistics of the Conversational Subset of IN22

| Statistic | Value |
|---|---|
| Number of unique conversations | 44 |
| Average turns per conversation $\pm$ std dev | $34.2 \pm 4.9$ |
| Number of unique topics | 23 |
| Randomly selected 5 topics | 'Government schemes', 'Movies', 'Historical Architectures', 'Geography of India', 'Legal Affidavit/documents' |
| Number of unique domains | 16 |
| Randomly selected 5 domains | 'arts', 'history', 'school life', 'healthcare', 'legal' |
| Number of unique prompts | 44 |
| Randomly selected 5 prompts | 'Joint Affidavit for Registration of Marriage', 'Diploma in web designing', 'Qutub Minar- visiting time and student discounts', 'How do you take out time for your hobbies ?', 'Social and Economic inequalities' |
| Number of unique scenarios | 37 |
| Randomly selected 5 scenarios | 'How to apply for a loan', 'Asking for the date/timing of the voting date', 'Housing/Colony', 'Learning Music', 'Challenges/Issues in Sports sector' |
| Avg number of speakers per conversation $\pm$ std dev | $2.0 \pm 0.0$ |

## E.1 Source Selection

For the Wikipedia subset, we carefully chose English source sentences from various Wikipedia categories to ensure broad coverage across different domains. Initially, we selected article pages within those categories and identify all the sentences as potential candidates. For each of these sentences, we construct a context window with a block size of 3, which typically includes one sentence before and after the candidate sentence. To satisfy the length criteria, we filter out sentences that are less than 6 or more than 80 words. To minimize overlaps with the FLORES-200 test set (Costa-jussà et al., 2022), we discard the sentences that share 4-gram or higher overlaps with any sentence in the FLORES-200 dev and devtest sets. The candidate sentence domains are manually annotated as described above. Following this, we randomly select the final set of sentences based on domain and length constraints. The detailed buckets are presented in Figure 10. It is important to note that we did not translate all the sentences within the context block, deviating from the approach followed in FLORES. This deviation was necessary to ensure the optimal length and domain diversity constraints were met.

For the Web Sources, we identified various Govt. of India websites and digital libraries that could be sources of multi-domain content with a focus on Indian topics. Many benchmarks like FLORES (Costa-jussà et al., 2022), NTREX (Federmann et al., 2022) do not have a fair representation of India-centric content, and we try to address this in the creation of this subset. We relied on PDF format documents to discover sentences that are hopefully not part of publicly available crawls like CommonCrawl (Xue et al., 2021; Conneau et al., 2020) or IndicCorp (Kakwani et al., 2020; Doddapaneni et al., 2023). The selection of sentences for translation follows a similar procedure to the Wikipedia subset. Figure 10 provides the bucket-wise and domain-wise distribution.

For the Conversation subset, we first create English conversations with a set of prompts and scenarios. The prompts are predefined topics or themes that are used to initiate a conversation. A prompt can be thought of as the starting point of a conversation, which sets the tone and direction for the interaction between the two speakers. For example, a prompt could be "Travel plans for the summer" or "Discussing a new project at work". The prompt is designed to encourage the speakers to discuss a particular topic or theme, and it serves as the foundation for the conversation. On the other hand, a scenario is a specific situation or context in which the conversation takes place. It provides additional context for the speakers and helps to shape the conversation. For example, a scenario could be "Planning a family vacation to Europe" or "Brainstorming ideas for a marketing campaign". The scenario provides a specific context for the prompt, which guides the speakers in their conversation. To create a conversation, two annotators from our annotator team played out

Table 54: An example from the Conversation subset of IN22 featuring a conversation between two speakers: a kid and his mother. The example belongs to the cultural domain, with festivities as a topic, the prompt of 14th April being a holiday, and the scenario being 'Historical importance'. Note that the speaker information is part of metadata and is not part of the text to be translated. Each turn in the conversation is a distinct instance from the benchmark. It is possible to reconstruct a conversation using the metadata released along with the translations.

| Speaker | Turn |
| --- | --- |
| Speaker 1 | Mom, let's go for a movie tomorrow. |
| Speaker 1 | I don't have to go to school. |
| Speaker 1 | It is a holiday. |
| Speaker 2 | Oh, tomorrow is the 14th of April right? |
| Speaker 2 | Your dad will also have the day off from work. |
| Speaker 2 | We can make a movie plan! |
| Speaker 1 | That's a good news! |
| Speaker 1 | Why is it a holiday though? |
| Speaker 1 | Are all schools, colleges and offices closed tomorrow? |
| Speaker 2 | It is Ambedkar Jayanti tomorrow! |
| Speaker 2 | This day is celebrated annually to mark the birth of Dr. B. R Ambedkar. |
| Speaker 2 | Have you heard of him? |
| Speaker 1 | I think I have seen him in my History and Civics book. |
| Speaker 1 | Is he related to our Constitution? |
| Speaker 2 | Absolutely! He is known as the father of the Indian Constitution. |
| Speaker 2 | He was a civil rights activist who played a major role in formulating the Constitution. |
| Speaker 2 | He played a crucial part in shaping the vibrant democratic structure that India prides itself upon. |
| Speaker 1 | I remember now! |
| | . . . |

the two speaker roles. Once a conversation is ready, it is then translated into 22 Indic languages. During translation, the translators have the entire conversation context available to them.

# F  Translation Workflow

## F.1  Translation Stages

**Source Sentence Selection Stage.**  The workflow begins with the selection of sentences to be translated based on various criteria to be met like domain coverage, length distribution, licensing constraints, *etc.* This helps in ensuring the right set of sentences as required for the project are shortlisted for translation. To ensure a broader vocabulary coverage, the sentences are taken from multiple domains such as News, Tourism, Business, Entertainment, History, Geography, Culture, Sports, and Health.

**Source Verification Stage.**  Once the candidate pool of source sentences is created, it is verified by annotators to ensure the correctness of the source sentences and metadata. This ensures that the sentences selected are valid, of good quality, and translatable. Shoonya efficiently supports a verification workflow where the annotator reads a sentence (with context) and selects any one of the given tags: *1. Clean*, *2. Difficult vocabulary*, *3. Context Incomplete*, *4. Ambiguous sentence*, and *5. Profane*. Sentences with minor errors such as spelling mistakes, and punctuation errors are corrected manually. If any sentence in a paragraph is discarded, the whole paragraph gets rejected, as context-agnostic translations might turn out ambiguous. In addition, the annotators might also add metadata like the domain and the topic to the source sentences.

**Translation Stage.**  The selected source sentences are translated by translators across all 22 Indic languages. To ensure quality, standard translation guidelines have been developed and iterated before starting the translation task. There is an active discussion amongst translators to ensure consistency. Translators of one language team help translators of another language team who are from the same language family or share geographical boundaries. This ensures the authenticity of transliterated words and cross-cultural nuances and gives a human touch to the output.

The translator is provided with:

- Source sentence and three context sentences around the source sentence to help resolve translation ambiguities.

- Translation outputs from one of the following engines (IndicTrans1 with fallback to Google Translate for unsupported language), which can be post-edited. Translators could post-edit, translate from scratch, or use any alternative MT system as a starting point. Note that post-editing support is provided only for the creation of training data. Providing MT as a reference helps translators speed up and overcome the existing mistakes in current translation models. A few low-resource languages like Kashmiri, Konkani, and Santali, where MT systems are not available, are supported by the output of other related languages such as Urdu, Marathi, and Bengali. This helps translators of low-resource languages to reuse syntactic structures and vocabulary from related languages (as long as such vocabulary is acceptable in the target language). To create test sets, the translators are expected to translate the sentences from scratch and are not shown any outputs from an MT system.

- To help translate technical vocabulary, the translators can consult dictionaries and glossaries using IndicGlossary[35]. IndicGlossary contains approximately 2 million glossary items across 13 different Indic language pairs and about 20 domains aggregated from various sources. These glossaries are sourced from the Commission for Scientific and Technical Terminology (CSTT) and Technology Development for Indian Languages (TDIL) which are the recommended sources for translation terminologies for different domains (Science, Engineering/Technology, Medical Science, Humanities, Social Sciences, Agricultural Science, and Veterinary Science).

For some low-resource languages, some translators were not proficient in English but had proficiency in another Indic language (called the pivot language). For these languages, the translators are provided with the pivot language translation, which they use to translate into their native language. We used this method for the following languages: Dogri (pivot Hindi), Konkani (pivot Marathi), Maithili (pivot Hindi), and Santali (pivot Bengali).

---

[35]https://github.com/AI4Bharat/Indic-Glossaries

**Quality Check Stage.** Simultaneous review of the translated sentences is required, as it helps provide feedback to the translators and improves the overall quality. For this, we have dedicated reviewers in each language who are translators with 5+ years of experience. The job of the reviewer is to improve the overall quality of the translation by correcting grammatical errors (if any), choosing better syntactic structures (if required), and rectifying inappropriate dialectical features to make the translations more standard. The reviewer manually verifies and corrects each translated sentence (if needed) to ensure adherence to the guidelines by selecting any one of the options on Shoonya, *1. Accepted*, *2. Accepted with Changes*, and *3. Rejected*. Rejected sentences go back to the translator with a note from the reviewer. The reviewer then corrects the translation based on the inputs provided in the note from the translator. The corrected sentences then go back to the translator for a second round of review.

## F.2 Translation Guidelines.

We developed an extensive set of translation guidelines to help the translators and ensure translation consistency and quality across annotators and languages. These have been developed starting with the guidelines prepared by LDC[36] for the BOLT Chinese-English translation task and further adapted for our scenarios and tasks. It is challenging to translate in 22 languages from 4 different language families, following the same set of rules, as syntax and availability of resources vary drastically across them. However, the guidelines were created considering the main goal of "getting as natural translations as possible". In the guidelines, we ensured the inclusivity of all unique linguistic features such as distinct word orders (SVO in Kashmiri), PNG agreement, tense/aspect differentiation in Manipuri, sociocultural nuances, extreme dialectic variations and challenges like right-to-left writing (Urdu), scripts like Meitei Mayek and Ol Chiki, languages which don't support longer syntactic structures like English, sentences with many subordinate clauses, languages spoken in multiple regions such as Sindhi, unavailability of modern vocabulary in languages like Sanskrit, inaccessibility of domain-specific dictionaries and glossaries in languages like Bodo, Santali and reviving the original form of languages like Assamese, Odia which are highly influenced by high resource languages in the same area (e.g., Bengali). The detailed guidelines are published as a standalone document here[37]. Some key highlights from these guidelines.

- The general principle is that the translation should maintain the meaning, style, tone, and register of the source. No information should be added or deleted.

- Official native scripts of the languages should be used.

- Named entities and borrowed words can either be translated or transliterated. The exact choice depends on the accepted convention in the language, if both choices exist. We avoid coining new translations if none exist, and the words are transliterated instead.

- Numbers, dates, and units are to be handled as per natural conventions in the target language.

- In the context of historical events/people, translators can use more formal/older conventions or terms. For more recent events/people, using more casual/colloquial conventions or terms is preferred.

- For test sets, sentences would be translated from scratch without aid from any MT output to avoid bias towards outputs of any MT system.

## F.3 Shoonya Translation Interface

Translations are performed using the translation task supported in Shoonya[8]. Shoonya has helped improve translator productivity and project management by providing features like transliteration support, context view, post-editing, quality control, and cross-lingual support. Performing reviews in real-time has helped the team improve the quality of translations whilst rectifying their mistakes. Shoonya supports right-to-left writing, which helps Urdu and Kashmiri

---

[36]https://catalog.ldc.upenn.edu/docs/LDC2008T18/
[37]https://github.com/AI4Bharat/IndicTrans2/blob/main/translation_guidelines.pdf

translators to speed up their typing. Simple features like 'Find and Replace', marking sentences as drafts, getting randomized sentences across domains, daily progress tab, etc. helped translators improve their productivity and collaborate closely with their peers. Below is a summary of Shoonya's features that have benefitted the translation task.

- *Transliteration Support*: Romanized input with automatic transliteration to native scripts to help translators not proficient in native script keyboards. The transliteration is powered by the open-source IndicXlit models (Madhani et al., 2023), which provide transliteration support for 20+ Indian languages.

- *Context View*: When translating a sentence, it helps to have the context in which the sentence is being translated to resolve any ambiguities. Shoonya allows translators to see paragraph-level context when translating an individual sentence.

- *Post-Editing*: Shoonya enables populating automatic translations from IndicTrans1 models, currently supporting 11 Indic languages. The translators can post-edit these initial translations.

- *Quality Control*: Shoonya offers various automated maker-checker flows to evaluate the quality of translated data. To further ensure quality, we implement a two-level maker-checker paradigm, in which an experienced reviewer verifies each translation for conformance to the translation guidelines. This approach involves two levels of processing for each sentence, providing a robust mechanism for ensuring high translation quality.

- *Cross-lingual Support*: For low-resource languages, Shoonya supports showing annotators translations in other related languages. For instance, given the task of translating English to Santali, the translators may have difficulty fully understanding the English sentence. In such cases, we also show the translators a Bengali translation (a language they are proficient in) of the same sentence to aid them with the task. This is a common scenario for many low-resource languages (Costa-jussà et al., 2022; Ebrahimi et al., 2022; Marivate et al., 2020).

# G   Language of India

This section provides an overview of Indian languages based on the 2011 census data, employing language classification by Joshi et al. (2020).

Table 55: Overview of Indian languages. Number of Native Speakers as per 2011 census. Language classification done according to the taxonomy introduced by Joshi et al. (2020), which classifies languages into 6 classes from 0 to 5 with 0 indicating extremely low resource and 5 indicating high resource language. Many of these languages are spoken across multiple states in the country. Sample column indicates the word "Bharat" written in different scripts.

| Language Code | Name | Family | Sub-family | Script | Sample | Class | #Native Speakers |
|---|---|---|---|---|---|---|---|
| asm_Beng | Assamese | Indo-Aryan | Eastern Indo-Aryan | Bengali | ভাৰত | 2 | 15.3M |
| ben_Beng | Bengali | Indo-Aryan | Eastern Indo-Aryan | Bengali | ভারত | 5 | 97.2M |
| brx_Deva | Bodo | Sino-Tibetan | Boroic | Devanagari | भारत | 1 | 1.4M |
| doi_Deva | Dogri | Indo-Aryan | Northern Indo-Aryan | Devanagari | भारत | 1 | 2.5M |
| gom_Deva | Konkani | Indo-Aryan | Southern Indo-Aryan | Devanagari | भारत | 1 | 2.2M |
| guj_Gujr | Gujarati | Indo-Aryan | Western Indo-Aryan | Gujarati | ભારત | 4 | 55.4M |
| hin_Deva | Hindi | Indo-Aryan | Central Indo-Aryan | Devanagari | भारत | 5 | 528.3M |
| kan_Knda | Kannada | Dravidian | South Dravidian | Kannada | ಭಾರತ್ | 5 | 43.7M |
| kas_Arab kas_Deva | Kashmiri | Indo-Aryan | Northern Indo-Aryan | Perso-Arabic Devanagari | بھارت भारत | 1 | 6.7M |
| mai_Deva | Maithili | Indo-Aryan | Eastern Indo-Aryan | Devanagari | भारत | 1 | 13.5M |
| mal_Mlym | Malayalam | Dravidian | Southern Dravidian | Malayalam | ഭാരത് | 4 | 34.8M |
| mar_Deva | Marathi | Indo-Aryan | Southern Indo-Aryan | Devanagari | भारत | 4 | 83.0M |
| mni_Beng mni_Mtei | Manipuri | Sino-Tibetan | Central Tibeto-Burman | Bengali Meitei | ভারত ꯑꯣꯏꯅꯥꯛ | 1 | 1.7M |
| npi_Deva | Nepali | Indo-Aryan | Northern Indo-Aryan | Devanagari | भारत | 2 | 2.9M |
| ory_Orya | Odia | Indo-Aryan | Eastern Indo-Aryan | Odia | ଭାରତ | 3 | 37.5M |
| pan_Guru | Punjabi | Indo-Aryan | North Western Indo-Aryan | Gurmukhi | ਭਾਰਤ | 3 | 33.1M |
| san_Deva | Sanskrit | Indo-Aryan | Indo-Aryan | Devanagari | भारत | 2 | 0.02M |
| sat_Olck | Santali | Austroasiatic | Munda | Ol Chiki | ᱵᱷᱟᱨᱚᱛ | 1 | 7.3M |
| snd_Arab snd_Deva | Sindhi | Indo-Aryan | North Western Indo-Aryan | Arabic Devanagari | پارت भारत | 1 | 2.7M |

| | | | | | | | |
|---|---|---|---|---|---|---|---|
| *tam_Taml* | *Tamil* | *Dravidian* | *South Dravidian* | *Tamil* | பாரத் | 4 | 69.0M |
| *tel_Telu* | *Telugu* | *Dravidian* | *South Central Dravidian* | *Telugu* | భారత్ | 4 | 81.1M |
| *urd_Arab* | *Urdu* | *Indo-Aryan* | *Central Indo-Aryan* | *Urdu* | بھارت | 5 | 50.7M |

# H Examples of language translation

This section shows a sentence translated into all Indian languages as an illustrative example.

Table 57: The table shows an example of the same sentence translated into all 22 Indian languages. The sentence translated is "All human beings are born free and equal in dignity and rights." from the UN Declaration on Human Rights.

| Language | Translation | Romanized Sentence |
|---|---|---|
| asm_Beng | সকলো মানুহ স্বাধীন হৈ জন্ম গ্ৰহণ কৰে আৰু মৰ্যাদা আৰু অধিকাৰ সকলোৱে সমান। | Xokolu manuh swadhin hoi jonmogrohon kore aru marjyada aru odhikaar xokolure xoman. |
| ben_Beng | মানুষ জন্মগতভাবে স্বাধীন এবং সম্মান ও অধিকার সবারই সমান। | Manush jonmogotobhabe swadhin ebong somman o odhikar sobari soman. |
| brx_Deva | गासिबो सुबुंआनो उदांयै जोनोम जादों आरो गासिबो सुबुंनि मान आरो मोनथायफोरा समान। | gasibw subungyanw udangywi jwnwm jadwng arw gasibw subungni man arw mwnthaiphwra soman. |
| doi_Deva | सबभै मनुक्ख मान– प्रतिश्ठा ते अधिकारें दे संदर्भ च सुतैंतर ते इक बरोबर पैदा होंदे न। | Sabhe manukh maan-pratishtha teh adhikaarein de sandarbh ch sutaintar teh ek barobar paida haundey n. |
| gom_Deva | सगळे मनीस स्वतंत्र म्हूण जल्माक येतात आनी प्रति–ष्ठा आनी हक्कांचे नदरेन समान आसतात. | sagle manis swatantra mhun jalmak yetat and pratishtha and hakkache nadren samaan astat. |
| guj_Gujr | બધાં મનુષ્યો સ્વતંત્ર જન્મે છે અને ગરિમા અને અધિકારોમાં સમાન હોય છે. | badha manushyo swatantra janme chhe ane garima ane adhikaro ma saman hoy chhe. |
| hin_Deva | सभी मनुष्य स्वतंत्र पैदा होते हैं और गरिमा और अधि–कारों में समान होते हैं। | sabhi manushya swatantra paida hote hain aur garima aur adhikaron mein samaan hote hain. |
| kan_Knda | ಎಲ್ಲಾ ಮನುಷ್ಯರು ಹುಟ್ಟಿನಿಂದಲೇ ಸ್ವತಂತ್ರರೂ ಘನತೆ ಹಾಗೂ ಹಕ್ಕುಗಳ ದೃಷ್ಟಿಯಿಂದ ಸಮಾನರೂ ಆಗಿರುತ್ತಾರೆ. | ellā manushyaru huttinindalē svatantrarū ghanate hāgū hakkugala drishtiyinda samānarū āgiruttāre. |
| kas_Deva | सलीम इंसान छी जनमी आज़ाद बे बराबर डिग्निटी बे हक़ | salim insaan chi janmi Azad be braaber dignity be haq |
| kas_Arab | برابر مٔنز حقوقس تٕ عزت تٕہ آزاد چھ انسان تمام - | Tamaam insaan chi Azad ta yezath ta haqooqs Manz braaber. |
| mai_Deva | सभ मनुष स्वतंत्र पैदा होयत अछि आओर अधिकार आ प्रतिष्ठा मे बराबर होयत अछि। | Sabh manukh swatantra paida hoyat achhi aaor adhikaar aa prtishtha me barabar hoyat achhi . |
| mal_Mlym | എല്ലാ മനുഷ്യരും സ്വാതന്ത്രരായി ജനിച്ച-വരും ഒപ്പം അന്തസിലും അവകാശങ്ങ-ളിലും തുല്യരുമാണ്. | ellaa manushyarum swathanthrarayi janichavarum oppam anthassilum avakaashangalilum thulyarumaanu. |
| mar_Deva | सर्व मनुष्य स्वतंत्र व्यक्ती म्हणून जन्माला येतात आणि प्रतिष्ठा आणि हक्कांच्या दुष्टीकोनातून समान असतात. | sarv manushya swatantrya vyakti mhanun janmala yetat aani pratishtha ani hakkanchya dushtikonatun samaan astat. |
| mni_Beng | মীওইবা খুদিংমক নীংতম্বা অমসূং ইকায়খুম্নবা অম-সূং হক্শেলগী লম্দা চপ মান্ননা লৈমিন্নরি। | Mioiba khudingmak ningtamba amasung ikaikhumnaba amasung hakselgi lamda chap mannana leiminnari. |
| mni_Mtei | ꯃꯤꯑꯣꯢꯕ ꯄꯨꯝꯅꯃꯛ ꯅꯤꯡꯇꯝꯕ ꯑꯃꯁꯨꯡ ꯏꯀꯥꯏꯈꯨꯝꯅꯕ ꯑꯃꯁꯨꯡ ꯍꯥꯛꯁꯤꯡ ꯃꯥꯟꯅꯅ ꯄꯣꯛ ꯏ ॥ | Mioiba pumnamak ningtamba amasung ikaikhumnaba amasung haksing mannana pok I. |
| npi_Deva | सबै मानिस स्वतन्त्र जन्मन्छन् र सम्मान तथा अधि–कारमा समान हुन्छन् छन्। | sabai maanis swatantra janmanchan ra sammaan tathaa adhikaarmaa samaan chan. |
| ory_Orya | ସମସ୍ତ ମନୁଷ୍ୟ ଜନ୍ମଗତ ଭାବେ ସ୍ୱାଧୀନ ଓ ସମ୍ମାନ ତଥା ଅଧିକାର ଦୃଷ୍ଟିରୁ ସମାନ | samasta manushya janmagata bhaabe swadhina o sammaana tathaa adhikaara drushtiru samaan. |
| pan_Guru | ਸੱਭ ਮਨੁੱਖ ਅਜ਼ਾਦ ਪੈਦਾ ਹੁੰਦੇ ਹਨ ਅਤੇ ਮਾਣ-ਸਨਮਾਨ ਤੇ ਅਧਿਕਾਰਾ ਵਿੱਚ ਬਰਾਬਰ ਹਨ। | Sabh manukh azaad paida hunde han ate mann saman te adhekara vich barabar han. |

| | | |
|---|---|---|
| *san_Deva* | सर्वे मानवजीविनः जन्मनः एव स्वतन्त्राः, मानदृष्ट्या अधिकारदृष्ट्या समानाश्च। | sarve maanavajivinah janmanah eva svatantraah, maanadrishtyaa adhikaaradrishtyaa samaanaashca. |
| *sat_Olck* | ᱥᱟᱱᱟᱢ ᱢᱟᱱᱢᱤ.ᱜᱮᱹ ᱯᱷᱩᱨᱜᱟᱞ.ᱯ ᱚᱛᱮᱫ ᱠᱩ ᱡᱟᱱᱟᱢᱚᱜ-ᱟ ᱟᱨ ᱢᱟᱱ ᱟᱨ ᱟ.ᱭᱫᱟ.ᱨ ᱨᱮ ᱠᱩ ᱥᱚᱢᱟᱱ ᱜᱤᱭᱟ | Sanam manmi ge phurgal ated ku janamog-a ar man ar aydar re ku soman giya. |
| *snd_Arab* | آهن ٿيا ۽ پيدا برابر ۾ حقن ۽ عزت ۽ آزاد انسان سڀ ۔ | Sabh insaan aazad paida thiya aahin, ain izzat ain hakkan mein barabar aahin. |
| *snd_Deva* | सभई इंसान आज़ाद, ऐं मान ऐं हकनि में हिक जहिड़ा ज़ावल आहिन। | Sabhai Insan aazad, ain maan ain hakan mein hik jahida javal aahin. |
| *tam_Taml* | மனிதர்கள் பிறப்பால் சுதந்திரமானவர்கள், மற்றும் சம உரிமைகளும் கண்ணியமும் கொண்டவர்கள். | manitharkal pirappaal suthanthiramaanavarkal, matrum sama urimaykalum kanniyamum kondavarkal. |
| *tel_Telu* | మనుషులంతా స్వేచ్ఛగా గౌరవమర్యాదలు, హక్కులలో సమానత్వంతో పుడతారు. | manushulantaa svecchagaa gouravamaryadalu, hakkulalo samantvamto pudataru. |
| *urd_Arab* | حقوق اور عزت اور ہیں ہوئے پیدا آزاد انسان تمام ہیں۔ برابر سے لحاظ کے | tamam insan azad paida hue hain aur izzat aur huqooq ke lihaz se barabar hain. |

# I   Language Coverage of various MT models

This section provides an overview of the Indian languages supported by different open-source and commercial NMT systems.

Table 59: Coverage of the 22 languages listed in the $8^{th}$ Schedule of the Constitution of India by various NMT systems

| language | IndicTrans1 | IndicTrans2 | Azure | NLLB-200 | Google Translate |
|---|---|---|---|---|---|
| asm_Beng | ✓ | ✓ | ✓ | ✓ | ✓ |
| ben_Beng | ✓ | ✓ | ✓ | ✓ | ✓ |
| brx_Deva | ✗ | ✓ | ✗ | ✗ | ✗ |
| doi_Deva | ✗ | ✓ | ✗ | ✗ | ✓ |
| gom_Deva | ✗ | ✓ | ✓ | ✗ | ✓ |
| guj_Gujr | ✓ | ✓ | ✓ | ✓ | ✓ |
| hin_Deva | ✓ | ✓ | ✓ | ✓ | ✓ |
| kan_Knda | ✓ | ✓ | ✓ | ✓ | ✓ |
| kas_Arab | ✗ | ✓ | ✗ | ✓ | ✗ |
| kas_Deva | ✗ | ✓ | ✗ | ✓ | ✗ |
| mai_Deva | ✗ | ✓ | ✓ | ✓ | ✓ |
| mal_Mlym | ✓ | ✓ | ✓ | ✓ | ✓ |
| mar_Deva | ✓ | ✓ | ✓ | ✓ | ✓ |
| mni_Beng | ✗ | ✓ | ✗ | ✓ | ✗ |
| mni_Mtei | ✗ | ✓ | ✗ | ✗ | ✓ |
| npi_Deva | ✗ | ✓ | ✓ | ✓ | ✓ |
| ory_Orya | ✓ | ✓ | ✓ | ✓ | ✓ |
| pan_Guru | ✓ | ✓ | ✓ | ✓ | ✓ |
| san_Deva | ✗ | ✓ | ✗ | ✓ | ✓ |
| sat_Olck | ✗ | ✓ | ✗ | ✓ | ✗ |
| snd_Arab | ✗ | ✓ | ✓ | ✓ | ✓ |
| snd_Deva | ✗ | ✓ | ✗ | ✗ | ✗ |
| tam_Taml | ✓ | ✓ | ✓ | ✓ | ✓ |
| tel_Telu | ✓ | ✓ | ✓ | ✓ | ✓ |
| urd_Arab | ✗ | ✓ | ✓ | ✓ | ✓ |
| **# languages** | 11 | 22 | 16 | 19 | 19 |
| **# language-script combinations** | 11 | 25 | 16 | 20 | 19 |

## J  Model Card - IndicTrans2

Following Mitchell et al. (2019), we provide a model card for our IndicTrans2 models.

### J.1  Model Details

- **Person or organization developing model:** IndicTrans2 models are multilingual translation models developed by AI4Bharat.[38]

- **Model data:** IndicTrans2 models were released on May 26, 2023.

- **Model version:** IndicTrans2 models described in this paper are version 1.0.0.

- **Model type:** IndicTrans2 models are 18-layer encoder-decoder transformer models with 1.1B parameters, one for English to Indic translation direction while the other for Indic to English translation.

- **Information about training algorithms, parameters, fairness constraints or other applied approaches, and features:** IndicTrans2 models were trained with the exact training configuration described in Section 5.5 and the training data described in Section 5.1. Sections 5.2, 5.3 and 5.7 describes the data preprocessing, tokenization, and postprocessing steps, correspondingly, that have been followed during the training.

- **Paper or other resources for more information:** See the rest of this paper for more details on IndicTrans2 models. More details are also available in `IndicTrans2`,[39] our open-source GitHub repository.

- **License:** IndicTrans2 models are made available through a permissive MIT license.[40]

- **Where to send questions or comments about the model:** Please open an issue[41] on our open-source GitHub repository.

### J.2  Intended Use

- **Primary intended uses:** IndicTrans2 models are machine translation models designed for research and commercial purposes, especially for Indic languages. These models enable the translation of the text, either in single or batch format, across 22 different Indic languages, including 25 language-script combinations to and from English. In addition, these models offer support for translation between Indic languages using a pivot-based approach. Further information on how to use the models can be found at `IndicTrans2`, our open-source GitHub repository.

- **Primary intended users:** The primary intended users of the IndicTrans2 models are:

  - Researchers aiming to explore and advance language technologies for Indic languages.
  - Individuals seeking to translate content to and from the supported Indic languages for various day-to-day use cases.
  - Organizations interested in translating their proprietary or internal content into the supported Indic languages.

- **Out-of-scope use cases:** IndicTrans2 models are released under MIT license without any usage limitations.

---

[38] https://ai4bharat.iitm.ac.in/
[39] https://github.com/ai4Bharat/IndicTrans2
[40] https://opensource.org/license/mit/
[41] https://github.com/AI4Bharat/IndicTrans2/issues

### J.3 Data, Metrics, Limitations, and Recommendations

- **Training dataset:** Section 5.1 described the parallel corpora used for training our models. Table 1 provides the statistics of the bitext pairs from different sources. In addition, we augment the training data with synthetic data as described in Section 5.6 generated from the intermediate IndicTrans2 models for training the final versions of the IndicTrans2 models.

- **Fine-tuning dataset:** BPCC-H-Wiki (see Section 3.3) and NLLB-Seed (Costa-jussà et al., 2022) were used for the fine-tuning of the IndicTrans2 models as described in Section 5.5. Table 1 provides the statistics of the gold-standard bitext pairs from different sources.

- **Evaluation dataset:** Section 6.2 describes all the benchmarks including FLORES-200 and our IN22 considered for evaluation of our IndicTrans2 models. The generation and evaluation procedure for the IndicTrans2 models are outlined in Section 6.4 and Section 6.5. Additionally, the baselines compared in this paper also follow the same evaluation procedure as the IndicTrans2 models.

- **Metrics:** IndicTrans2 models were evaluated using several metrics such as chrF++, BLEU and COMET as described in Section 6.3. We use chrF++ as our primary metric. In addition, we also conduct the human evaluation with the XSTS protocol on a small portion of IN22 Combined evaluation set (see Appendix C).

- **Limitations:** Section 9 describes the known caveats of the IndicTrans2 models.

- **Recommendations for future work:** IndicTrans2 serves as a strong foundational translation model with extensive support for all 22 scheduled Indic languages. However, it is important to acknowledge that there are additional languages that are currently not supported (see Section 2). There is a potential to expand IndicTrans2 models to support more languages or improve the performance of the existing supported languages through minimal fine-tuning.

## K   Model Card - IndicTrans2-M2M

Following Mitchell et al. (2019), we provide a model card for IndicTrans2-M2M models.

### K.1   Model Details

- **Person or organization developing model:** IndicTrans2-M2M models are multilingual translation models developed by AI4Bharat.[42]

- **Model data:** IndicTrans2-M2M models were released on Dec 01, 2023.

- **Model version:** IndicTrans2-M2M models described in this paper are version 1.0.0.

- **Model type:** IndicTrans2-M2M and IndicTrans2-Dist-M2M models are 18-layer encoder-decoder transformer models with 1.2BM parameters and the compact variant with 350M parameters, respectively, supporting Indic to Indic translation.

- **Information about training algorithms, parameters, fairness constraints or other applied approaches, and features:** IndicTrans2-M2M and IndicTrans2-Dist-M2M models were trained with the exact training configuration and the training data described in Section 7.5.2. Sections 5.2, 5.3 and 5.7 describes the data preprocessing, tokenization, and postprocessing steps, correspondingly, that have been followed during the training.

- **Paper or other resources for more information:** See the rest of this paper for more details on IndicTrans2-M2M and IndicTrans2-Dist-M2M models. More details are also available in `IndicTrans2`,[43] our open-source GitHub repository.

- **License:** IndicTrans2-M2M and IndicTrans2-Dist-M2M models are made available through a permissive MIT license.[44]

- **Where to send questions or comments about the model:** Please open an issue[45] on our open-source GitHub repository.

### K.2   Intended Use

- **Primary intended uses:** IndicTrans2-M2M and IndicTrans2-Dist-M2M models are machine translation models designed for research and commercial purposes, especially for Indic languages. These models enable the translation of the text, either in single or batch format, between 22 different Indic languages. Further information on how to use the models can be found at `IndicTrans2`, our open-source GitHub repository.

- **Primary intended users:** The primary intended users of the IndicTrans2-M2M and IndicTrans2-Dist-M2M models are:

    - Researchers aiming to explore and advance language technologies for Indic languages.
    - Individuals seeking to translate content to and from the supported Indic languages for various day-to-day use cases.
    - Organizations interested in translating their proprietary or internal content into the supported Indic languages.

- **Out-of-scope use cases:** IndicTrans2-M2M and IndicTrans2-Dist-M2M are released under MIT license without any usage limitations.

---

[42] https://ai4bharat.iitm.ac.in/
[43] https://github.com/ai4Bharat/IndicTrans2
[44] https://opensource.org/license/mit/
[45] https://github.com/AI4Bharat/IndicTrans2/issues

### K.3 Data, Metrics, Limitations, and Recommendations

- **Training dataset:** Section 7.5.2 described the parallel corpora used for training our models.

- **Evaluation dataset:** Section 6.2 describes all the benchmarks including FLORES-200 and our IN22 considered for evaluation of our IndicTrans2-M2M and IndicTrans2-Dist-M2M models. The generation and evaluation procedure for the IndicTrans2-Dist models are outlined in Section 6.4 and Section 6.5. Additionally, the baselines compared in this paper also follow the same evaluation procedure as the IndicTrans2-M2M and IndicTrans2-Dist-M2M models.

- **Metrics:** IndicTrans2-M2M and IndicTrans2-Dist-M2M models were evaluated using several metrics such as chrF++, BLEU and COMET as described in Section 6.3. We use chrF++ as our primary metric.

- **Limitations:** Section 9 describes the known caveats of the IndicTrans2-M2M and IndicTrans2-Dist-M2M models models. We also do not conduct an XSTS human evaluation for the IndicTrans2-Dist models.

- **Recommendations for future work:** IndicTrans2-M2M and IndicTrans2-Dist-M2M models serves as a strong foundational translation model with extensive support for all 22 scheduled Indic languages. However, it is important to acknowledge that there are additional languages that are currently not supported (see Section 2). There is a potential to expand IndicTrans2-M2M and IndicTrans2-Dist-M2M models to support more languages or improve the performance of the existing supported languages through minimal fine-tuning.

# L   Model Card - IndicTrans2-Dist

Following Mitchell et al. (2019), we provide a model card for IndicTrans2-Dist models.

## L.1   Model Details

- **Person or organization developing model:** IndicTrans2-Dist models are multilingual translation models developed by AI4Bharat.[46]

- **Model data:** IndicTrans2-Dist models were released on Dec 01, 2023.

- **Model version:** IndicTrans2-Dist models described in this paper are version 1.0.0.

- **Model type:** IndicTrans2-Dist models are 18-layer encoder-decoder transformer models with 211M parameters, one for English to Indic translation direction while the other for Indic to English translation.

- **Information about training algorithms, parameters, fairness constraints or other applied approaches, and features:** IndicTrans2-Dist models were trained with the exact training configuration described in Appendix D and the training data described in Section 7.6. Sections 5.2, 5.3 and 5.7 describes the data preprocessing, tokenization, and postprocessing steps, correspondingly, that have been followed during the training.

- **Paper or other resources for more information:** See the rest of this paper for more details on IndicTrans2-Dist models. More details are also available in `IndicTrans2`,[47] our open-source GitHub repository.

- **License:** IndicTrans2-Dist models are made available through a permissive MIT license.[48]

- **Where to send questions or comments about the model:** Please open an issue[49] on our open-source GitHub repository.

## L.2   Intended Use

- **Primary intended uses:** IndicTrans2-Dist models are machine translation models designed for research and commercial purposes, especially for Indic languages. These models enable the translation of the text, either in single or batch format, across 22 different Indic languages, including 25 language-script combinations to and from English. In addition, these models offer support for translation between Indic languages using a pivot-based approach. Further information on how to use the models can be found at `IndicTrans2`, our open-source GitHub repository.

- **Primary intended users:** The primary intended users of the IndicTrans2-Dist models are:

  - Researchers aiming to explore and advance language technologies for Indic languages.
  - Individuals seeking to translate content to and from the supported Indic languages for various day-to-day use cases.
  - Organizations interested in translating their proprietary or internal content into the supported Indic languages.

- **Out-of-scope use cases:** IndicTrans2 models are released under MIT license without any usage limitations.

---

[46] https://ai4bharat.iitm.ac.in/
[47] https://github.com/ai4Bharat/IndicTrans2
[48] https://opensource.org/license/mit/
[49] https://github.com/AI4Bharat/IndicTrans2/issues

### L.3   Data, Metrics, Limitations, and Recommendations

- **Training dataset:** Section 7.6 described the parallel corpora used for training our models.

- **Fine-tuning dataset:** BPCC-H-Wiki (see Section 3.3) and NLLB-Seed (Costa-jussà et al., 2022) were used for the fine-tuning of the IndicTrans2-Dist models as described in Section 5.5. Table 1 provides the statistics of the gold-standard bitext pairs from different sources.

- **Evaluation dataset:** Section 6.2 describes all the benchmarks including FLORES-200 and our IN22 considered for evaluation of our IndicTrans2-Dist models. The generation and evaluation procedure for the IndicTrans2-Dist models are outlined in Section 6.4 and Section 6.5. Additionally, the baselines compared in this paper also follow the same evaluation procedure as the IndicTrans2-Dist models.

- **Metrics:** IndicTrans2-Dist models were evaluated using several metrics such as chrF++, BLEU and COMET as described in Section 6.3. We use chrF++ as our primary metric.

- **Limitations:** Section 9 describes the known caveats of the IndicTrans2-Dist models. We also do not conduct an XSTS human evaluation for the IndicTrans2-Dist models.

- **Recommendations for future work:** IndicTrans2-Dist serves as a compact yet strong foundational translation model with extensive support for all 22 scheduled Indic languages. However, it is important to acknowledge that there are additional languages that are currently not supported (see Section 2). There is a potential to expand IndicTrans2-Dist models to support more languages or improve the performance of the existing supported languages through minimal fine-tuning.

# M    Dataset Card

Following Gebru et al. (2021); Pushkarna et al. (2022), we provide a dataset card for our Bharat Parallel Corpus Collection, the dataset used to train IndicTrans2 as well as IN22, our benchmark testsets for Indic languages.

## M.1    Dataset Description

- **Dataset summary:**

  - Bharat Parallel Corpus Collection (BPCC) is a comprehensive and publicly accessible parallel corpus comprising existing and newly added data for all 22 scheduled Indic languages. It consists of two components: BPCC-Mined and BPCC-Human. BPCC contains a total of ~230M bitext pairs (see Table 1). BPCC-Mined comprises ~228 million pairs, with around ~126 million pairs newly mined as part of this work. On the other hand, BPCC-Human consists of 2.2 million gold standard En-X pairs, with additional contributions of 644K bitext pairs from English sentences sourced from Wikipedia (forming the Bharat Parallel Corpus Collection-H-Wiki subset) and 139K sentences covering content from day-to-day use cases (forming the Bharat Parallel Corpus Collection-H-Daily subset). It is worth highlighting that BPCC provides the first available datasets for many languages and significantly increases the available data for all languages covered.

  - IN22 is a newly created comprehensive benchmark for evaluating machine translation performance in multi-domain, n-way parallel contexts across 22 Indic languages. It has been created from three distinct subsets, namely IN22-Wiki, IN22-Web, and IN22-Conv. The Wikipedia and Web sources subsets offer diverse content spanning news, entertainment, culture, legal, and India-centric topics. IN22-Wiki and IN22-Web have been combined and considered for evaluation purposes and released as IN22-Gen. Meanwhile, the conversation domain subset IN22-Conv is designed to assess translation quality in typical day-to-day conversational-style applications.

- **How to use the data?** We provide the links to access the data and directions for usage in the README of `IndicTrans2`,[50] our open-sourced GitHub repository.

- **Supported tasks and leaderboards:** The provided data is primarily intended for training machine translation models. It serves as a valuable training corpus for developing and improving such models. Furthermore, the IN22 benchmark dataset is included, which serves as a robust evaluation set for assessing the performance of machine translation models. Initial results of the IndicTrans2 models and the existing baselines are available in our open-source GitHub repository, providing insights and comparisons as of the release date.

- **Languages:** The dataset covers a total of 22 scheduled Indic languages with multiple scripts, amounting to a total of 25 language-script combinations.

## M.2    Dataset Creation

- **Curation rationale:**

  - BPCC is created to train and improve machine translation models. It is a valuable resource for developing and improving such models. Section 4 provides a detailed description of the procedure followed for the mined data collection, referred to as BPCC-Mined. Similarly, Section 3.3 outlines the motivation and annotation procedure for creating high-quality seed data, referred to as BPCC-Human. Additionally, section 4.3 provides insights into the curation and filtration process of the existing mined parallel corpora.

  - IN22 benchmark dataset is created to serve as a reliable evaluation set for assessing the performance of machine translation models. Section 3.2 provides a comprehensive discussion about the subsets of the evaluation benchmark and the creation procedure for each subset.

- **Source data:**

---

[50]https://github.com/AI4Bharat/IndicTrans2

- Table 1 summarizes the parallel corpora from different sources. BPCC-Mined (see Section 4) component is typically mined from IndicCorp v2 (Doddapaneni et al., 2023) and the Internet Archive.[51] BPCC-Human-Wiki is a multi-domain seed data collection that is sourced from Wikipedia articles whereas BPCC-Human-Daily is an in-house created dataset covering content from day-to-day conversations and use cases (see Section 3.3).
- IN22 benchmark is a comprehensive multi-domain benchmark that consists of three subsets: IN22-Wiki, sourced from Wikipedia articles; IN22-Web, sourced from PDFs available on various government websites and open-source books; and IN22-Conv, an in-house benchmark created through the interplay between annotators (see Section 3.2).

- **Annotations:** The annotations were performed on the Shoonya platform.[52] Section 3 provides further details about the procedure and the guidelines followed for annotations.

- **Personal and sensitive information:** Given that a substantial portion of the dataset is mined from publicly available websites at a large scale, we acknowledge the possibility of unintentional inclusion of personal and sensitive information. If there are any concerns regarding potential leakages of such information, please reach out to `miteshk@cse.iitm.ac.in` for further assistance and resolution.

## M.3 Considerations for Using the Data

- **Social impact of the dataset:** The dataset has a notable social impact, as it is specifically constructed and curated to improve the translation quality of all 22 scheduled Indic languages. It also provides support for low-resourced languages that utilize multiple scripts. This dataset contributes to the overall improvement of translation models, benefiting a wide range of users, helping bridge language barriers, and facilitating better communication and understanding across diverse linguistic communities.

- **Discussion of biases:** The current work does not explicitly examine biases in the data. However, we acknowledge the importance of studying biases and hope to conduct further investigations in this area in the future.

## M.4 Additional Information

- **Dataset Curators:** The dataset was curated by AI4Bharat, who collected data from various existing sources, including contributions from the existing dataset contributors. AI4Bharat also releases mined data, in-house created seed data, and benchmarks.

- **Licensing Information:** Bharat Parallel Corpus Collection (BPCC) consists of the largest publicly available parallel corpora for Indic languages. It includes various types of corpora obtained from different sources. The licensing information for each category is as follows:

  - Existing Mined Corpora (NLLB & Samanantar): These corpora are released under the CC0 license.[53]
  - Existing Seed Corpora (NLLB, ILCI, MASSIVE): The seed corpora are also released under the CC0 license.[53]
  - Newly added mined corpora (Samanantar++ & Comparable): The newly added mined corpora are also released under the CC0 license.[53]
  - Newly added seed corpora (Wiki & Daily): The newly added seed corpora are released under the CC BY 4.0 license.[54]
  - Newly created IN-22 test set: The IN22 test set is released under the CC BY 4.0 license.[54]

---

[51] https://archive.org
[52] https://ai4bharat.iitm.ac.in/shoonya
[53] https://creativecommons.org/share-your-work/public-domain/cc0/
[54] http://creativecommons.org/licenses/by/4.0

