# OpenReview forum: "IndicTrans2: Towards High-Quality and Accessible Machine Translation Models for all 22 Scheduled Indian Languages"
_TMLR — Accepted by TMLR_

### Review · Reviewer_zBPR · 2023-10-05

**Summary Of Contributions:**

This work presents a machine translation model for 22 Indic languages including high resource and low resource languages. This work involves creating a benchmark data to assess the quality of 22 Indic languages from/to English, collecting large parallel resources to construct the translation model and comparing the quality of the model agains other models trained on different data.

**Audience:**

No

**Broader Impact Concerns:**

No particular concerns besides those listed limitations in Section 9 of the manuscript.

**Claims And Evidence:**

No

**Requested Changes:**

* This work needs to investigate any insights into the model by differentiating the data sources and language variations. Currently there are multiple sources, e.g., monolingual and comparable corpora, and it is very hard to isolate the impact of training data.

* This work has limited work on zero-shot translation directions by presenting only empirical results in Appendix, although it should be a way to understand the behavior of a multilingual model, e.g., how representations might be affected by inclusion of particular languages. Note that the collected corpora sounds a collection of multiple sources without any systematic comparisons of language directions, e.g., English centric/non-centric directions.

**Strengths And Weaknesses:**

Strengths

* Large scale data collection for Indic language that will be of interest to those who are working for those languages. The quality controlled benchmark data will be of use for future investigations.

* Performance gains when compared with other open/closed models trained on separate training data.

Weaknesses

* The major contribution of this work is the data collection for the benchmark dataset and training data. However, the contribution does not fall into any major research scopes of TMLR listed in https://jmlr.org/tmlr/editorial-policies.html. Note that TMLR is expecting the contribution to "the understanding of the computational and mathematical principles that enable intelligence through learning, be it in brains or in machines."

* Data mining from monolingual sources and comparable translation sources are rather standard method employed in numbers of prior work as noted in the cited papers, e.g., LaBSE for sentence pair scoring and margin-based scoring. All the preprocessing and tokenization methods are also kind of standard ways, e.g., SentencePiece. Translation model is also coming from a rather standard Transformer with tweaks in prior work. Thus, it is hard to find new contributions.

* Clarity is an issue. It is not clear whether the collected data for training is English-to-X or X-to-Y where X and Y come from Indic languages. There exists no description on whether the trained model is a multilingual model, i.e., a single model that is capable to translate to/from any Indic languages in addition to English, or a collection of bilingual model, i.e., a model capable to translate only a single direction.

* Analysis is based only on objective metrics, e.g., Comet, English ChF++, only. Human evaluation is presented in Appendix, but it would be better to assess the quality in a qualitative manner so that it is easy to find the strengths and weakness of the proposed approach.

---

> ### Author Response · Authors · 2023-10-21
> **Rebuttal by Authors**
>
> We thank the reviewer for constructive and valuable comments. We are encouraged by positive comments, including the benefits of our benchmarks and data contributions to the low-resource MT community and the competitiveness of our model with SOTA commercial systems. We provide comments below to the concerns raised by reviewer.
>
> ### 1. Novelty and compatibility of our work with the scope of TMLR
>
> **While our work might have limited theoretical aspects, we believe our research presents interesting empirical findings and practical insights that can be a valuable resource in advancing the field and enabling real-world impact to the field of low-resource machine translation (MT).** Firstly, we conduct a comprehensive analysis of efficient and cost-effective training strategies, including optimal sentence embedding selection, filtering thresholds for mining data, and the efficacy of a multi-stage training and fine-tuning curriculum for training MT models with limited computational resources.
>
> Previous studies [A, B] show that MT performance is influenced by data scale but minimally affected by noisy translation pairs. Recent works, such as NLLB [C], apply a universal filtering threshold across different language groups and consume all mined data for large-scale model training. However, our experiments (see Table 15 in the manuscript) reveal that strict filtering yields superior models with just a fraction of the mined data (20%). Manual verification confirms the presence of misaligned pairs in the NLLB data released by AI2. Our findings show that conducting such language-group-specific studies for different design choices can effectively improve the training efficiency. We believe that our work can serve as a blueprint for conducting similar studies across different language families and consequently, these language-family-specific insights would be extremely useful for improving training efficiency and performance of massively multilingual models such as NLLB in the future.
>
> Furthermore, our work also conducts a robust evaluation of various MT systems across different content types, encompassing general and conversational settings. There are limited benchmarks for evaluating MT systems, and most rely on general-purpose benchmarks like FLORES-200 for evaluation. However, such benchmarks may not be very representative of the demographics of the region where these models would be deployed for real-word usage. Therefore, as a part of our work we also introduce a novel India-centric benchmark specifically designed for this purpose. **We believe that our work would also motivate other groups to develop region / demography-centric benchmarks to gain more fine-grained evaluations of the practicality and deployment suitability of  these models in real-life scenarios.**
>
> Our work stands out as the first to develop an MT model supporting all 22 scheduled Indian languages. Additionally, we also distill the pre-trained models of 1.2B parameters into 200M parameters without compromising the performance, making them accessible for resource-constrained settings. **We believe the research community, especially those exploring low-resource settings, and leveraging linguistic relatedness, would find value in our work.** Our models, despite their compactness, have proved they can compete with larger multilingual models. **We anticipate our work will stimulate further research interest into such low-resource settings and urge small academic/research groups to strive for developing compact, open-source SOTA models that can compete with massively multilingual models thereby transforming the landscape of multilingual MT models.**
>
> [A] Gordon et al. Data and Parameter Scaling Laws for Neural Machine Translation. EMNLP, 2021.
>
> [B] Bansal et al. Data Scaling Laws in NMT: The Effect of Noise and Architecture. ICML, 2022
>
> [C] Costa-jussà et al. No Language Left Behind: Scaling Human-Centered Machine Translation, ArXiv, 2022.

---

> > ### Author Response · Authors · 2023-10-21
> > **Continuation of Above Rebuttal Response**
> >
> > ### 2. Subset-wise analysis of mined bitext data
> >
> > We would like to highlight that our collected data corpora is “systematically” and thoughtfully created to ensure “multi-domain” coverage and with a view to ensure that the trained model generalize well to different text styles and domains.
> >
> > The component of the data : BPCC-Mined consists of 2 subsets monolingual and comparable. Monolingual subset is created by large-scale mining where the focus is on “scale” and we prioritize recall over precision. The corpora thus created are opportunistic in nature and may be highly skewed in terms of domain distribution as predominantly data available on the web is from news sources. Therefore, to complement this we also do a mining from comparable corpora. Comparable corpora involves a human-in-the-loop to identify diverse sources spanning different domains. Post which the document-level alignment is leveraged to have document as a search-space to obtain precise matches and very low false positive rate. Thus comparable mining is not only high-quality but also has coverage of domains which might lack or not have any representation in monolingual subset. Table 1 shows that bitext pairs from the comparable subset has higher average cosine similarity than the monolingual mined subset.
> >
> > | **language** | **monolingual** | **comparable** |
> > |:------------:|:---------------:|:--------------:|
> > |   asm_Beng   |      0.8361     |     0.8565     |
> > |   ben_Beng   |      0.8510     |     0.8649     |
> > |   guj_Gujr   |      0.8478     |     0.8685     |
> > |   hin_Deva   |      0.8505     |     0.8674     |
> > |   kan_Knda   |      0.8483     |     0.8677     |
> > |   mal_Mlym   |      0.8470     |     0.8674     |
> > |   mar_Deva   |      0.8462     |     0.8655     |
> > |   ory_Orya   |      0.8419     |     0.8617     |
> > |   pan_Guru   |      0.8482     |     0.8799     |
> > |   tam_Taml   |      0.8466     |     0.8652     |
> > |   tel_Telu   |      0.8475     |     0.8670     |
> > |   urd_Arab   |      0.8413     |     0.8702     |
> >
> > ### 3. Clarity about the trained models
> >
> > Our IndicTrans2 models are English-centric one-to-many and many-to-one multilingual models trained on English-centric data. In the abstract as well as multiple sections like 5.4 (Architecture), 7.5 (Indic-Indic evaluation) and 9 (Limitations and Future Work), we do highlight that the models we train are "English-centric." Additionally, we mention “transfer learning” in different sections of the draft, indicating that we train multilingual models to enable cross-lingual transfers from high-resource to low-resource. We will revise the writing in the final draft to make these details more clear.

---

> > > ### Author Response · Authors · 2023-10-21
> > > **Continuation of Above Rebuttal Response**
> > >
> > > ### 4. Qualitative fine-grained evaluation
> > >
> > > We acknowledge that objective metrics alone may not fully be indicative of a model's real-world effectiveness. However, we strive to conduct a multi-metric objective evaluation adhering to the best evaluation practices by reporting all the widely used metrics and further reinforcing our findings with statistical significance tests. Additionally, we report domain-specific ChrF++ scores on our benchmark - IN22-Gen for various systems averaged across language in Table 2. Our model consistently outperforms baselines in the En-Indic direction and performs comparably in the Indic-En direction across all domains. This demonstrates the importance of including diverse subsets of data to achieve multi-domain coverage.
> > >
> > > Considering the resource and time constraints, it was not feasible to conduct a full-scale human evaluation, therefore we conduct and report the findings of a preliminary human evaluation. However, our models have been deployed internally and externally across various domains and would serve as good human feedback data for qualitative analysis. Therefore, we leave this as future work extending our current submission.
> > >
> > > Table 2: Domain-wise evaluation on IN22-Gen averaged across languages.
> > >
> > > |               |   En-Indic   |             |        |       |   Indic-En   |             |        |       |
> > > |---------------|:------------:|:-----------:|:------:|:-----:|:------------:|:-----------:|:------:|:-----:|
> > > |     Domain    | NLLB 54B MoE | IndicTrans2 | Google | Azure | NLLB 54B MoE | IndicTrans2 | Google | Azure |
> > > |    Culture    |     47.6     |     51.0    |  49.0  |  49.6 |     59.8     |     61.5    |  60.9  |  56.3 |
> > > |    Economy    |     46.2     |     49.8    |  45.6  |  47.5 |     67.1     |     66.5    |  66.0  |  62.4 |
> > > |   Education   |     45.8     |     49.0    |  46.2  |  47.6 |     62.9     |     62.6    |  63.4  |  58.8 |
> > > | Entertainment |     48.5     |     51.5    |  49.2  |  50.5 |     64.2     |     64.2    |  63.6  |  61.1 |
> > > |   Geography   |     49.2     |     52.0    |  50.3  |  51.1 |     65.7     |     65.3    |  66.2  |  62.1 |
> > > |  Governments  |     50.9     |     54.5    |  50.9  |  52.3 |     68.3     |     68.0    |  66.4  |  63.1 |
> > > |     Health    |     49.0     |     52.3    |  49.6  |  50.2 |     67.1     |     67.0    |  66.6  |  62.5 |
> > > |    Industry   |     48.4     |     52.7    |  48.9  |  50.4 |     66.1     |     66.3    |  66.1  |  62.7 |
> > > |     Legal     |     49.0     |     52.4    |  49.1  |  50.6 |     65.3     |     65.4    |  65.5  |  61.3 |
> > > |      Nws      |     52.1     |     54.7    |  52.5  |  53.8 |     68.7     |     69.6    |  68.7  |  66.1 |
> > > |    Religion   |     45.1     |     47.9    |  46.7  |  47.6 |     58.0     |     59.3    |  59.2  |  56.0 |
> > > |     Sports    |     49.5     |     52.2    |  50.0  |  50.4 |     67.0     |     67.0    |  67.0  |  63.4 |
> > > |    Tourism    |     49.8     |     53.5    |  51.6  |  51.8 |     65.3     |     66.0    |  65.1  |  62.6 |
> > >
> > > ### 5. Zero-shot experimentation
> > >
> > > We appreciate the reviewer's suggestion to investigate zero-shot translation directions and analyze the behavior of our multilingual MT model in more depth. However, we would like to emphasize that our main focus in this work is to identify the optimal training recipe for developing SOTA models that can compete with massively multilingual MT models and commercial systems. While analyzing zero-shot translation would indeed be an interesting avenue to explore, it would require significant computational resources to train different language combinations and assess the impact of including specific languages at the representation level. At this point, we are currently studying zero-shot Indic-Indic translation, which we believe would be better suited for a separate publication that extends our current submission.

---

### Review · Reviewer_SHNd · 2023-10-05

**Summary Of Contributions:**

This work aims to address three major challenges that hinder the development of Indic machine translation: (i) no parallel training data spanning all 22 languages, (ii) no robust benchmarks covering all these languages and containing content relevant to India, and (iii) no existing translation models which support all 22 scheduled languages of India.

To address those challenges, this work releases the largest publicly available parallel corpora for Indic languages, releases the first $n$-way parallel benchmark covering all 22 Indian languages, and presents the first translation model to support all 22 languages. Overall, this work contributes to the community through data, benchmarks, and models. Those contributions may facilitate the development of Indic machine translation.

**Audience:**

Yes

**Broader Impact Concerns:**

This work will open-source data, benchmarks, and models. Those contributions may facilitate the development of Indic machine translation.

**Claims And Evidence:**

Yes

**Requested Changes:**

Please refer to the weaknesses above.

**Strengths And Weaknesses:**

Strengths:

1. This work makes solid contributions regarding data, benchmarks, and models for Indic machine translation. It can serve as a foundation for developing translation systems for Indic as well as low-resource languages.

2. The paper provides exhaustive details for constructing the data, benchmarks, and models. Those details are beneficial for reproduction. In addition, the paper presents the background of the Indic languages. This work is well-motivated.

3. The proposed techniques achieve impressive performance compared with existing large language models. Those proposed models would be beneficial to the community.

Weaknesses:

1. Although the contributions of this work are clear and sound, there are too many details in the main body of the paper.  Many contents can be moved to the Appendix for interested readers. There is no need to describe the widely used techniques in detail. For example, the pre-processing steps in Section 5.2, the concept and steps for data augmentation in Section 5.6, and the exhaustive details of metrics in Section 6.3. Those details can be significantly reduced or moved to the Appendix.

2. The paper is very long. There are inconsistent descriptions and writing mistakes. A more thorough proofreading is required. Examples are as follows.

- The abstract mentions four contributions. However, the conclusion only summarizes them as three axes.

- An extra stop mark in the title of Section 3.2.3.

---

> ### Author Response · Authors · 2023-10-21
> **Rebuttal by Authors**
>
> We thank the reviewer for constructive and valuable comments about the work serving as a foundation for developing similar SOTA MT systems across different language families and low-resource languages. Further, we also appreciate that the reviewer finds our contributions across different fronts valuable and also reproducible. We provide our responses to the reviewers’ concerns below.
>
> ### 1. Reorganization of content in main paper and appendix.
>
> We appreciate the reviewer's feedback regarding the excessive level of detail in our paper. We included these details in order to provide a comprehensive understanding of the widely used techniques and metrics employed in our work, particularly for readers who may be unfamiliar with these concepts. We will revise the final camera-ready version to restructure the supplementary details to the Appendix.
>
> ### 2. Regarding Proof-reading.
>
> Thank you for bringing these concerns to our attention. We will thoroughly review and revise our final camera version to ensure that these inconsistencies are eliminated.

---

### Review · Reviewer_2oVS · 2023-10-08

**Summary Of Contributions:**

The paper discusses what are the needs to achieve multilingual machine translation system in India for the 22 "scheduled" languages.

**Audience:**

Yes

**Broader Impact Concerns:**

No concerns, I only imagine the paper to bring the country together by building a system that works for these 22 languages.

**Claims And Evidence:**

Yes

**Requested Changes:**

No requested changes

**Strengths And Weaknesses:**

Strength
- The word "scheduled" when preceded by "tribe" or "caste" is a discriminative word that has often created societal divide in the country
- This paper addresses the 22 scheduled languages as listen in the Constitution of India
- The paper is very well motivated and has covered societal impacts such as in education and law. It also discusses how the educational text, transcribed speeches and legal documents can be used to create data
- The paper has covered aspects of user-anonymization (privacy) and open access (accessibility)

There is a data mining workflow described which covers
- workflow scalability
- addresses mining from monolingual corpora for building the multi-lingual system

Under the modeling section, the paper discusses the steps such as training data and its preparation
- Preprocessing
- Data deduplication has been discussed to improve efficiency
- Postprocessing
- Script unification

While achieving the bigger goal, the paper notes a thorough evaluation
Quantitative
- BLEU, ChrF++, COMET
Qualitative
- human evaluations

---

> ### Author Response · Authors · 2023-10-21
> **Rebuttal by Authors**
>
> We thank the reviewer for constructive and valuable comments. We are encouraged by the positive comments, including scalability of our mining pipeline, domain diversity of data we mined (comparable subset), robust evaluation practices, and also the societal impact of our work.

---

### Review · Reviewer_GByW · 2023-10-17

**Summary Of Contributions:**

This paper describes a family of contributions that enable the creation of IndicTrans2, a multilingual model that allows XX>En and En>XX translation for all 22 of India's scheduled languages.  One major resource contribution is new training corpora including new manually-sourced seed data, newly crawled data (using both sentence-mining and site-specific document-mining strategies) and newly-filtered versions of existing corpora. Another major resource contribution is a new evaluation benchmark for all 22 languages of interest drawn from either India-specific articles or conversational data in an India context. They also add 3 missing languages to FLORES-200 in order to have it also cover all 22 languages of interest. As a final resource contribution, they describe the creation of an MT system based on these resources, and show how it improves performance for many languages when compared to other open-source multilingual systems, and covers some that had never been covered before. Experimental contributions include studies on the value of data filtering, the value of fine-tuning with trusted seed data, and the value of data augmentation.

**Audience:**

Yes

**Broader Impact Concerns:**

None.

**Claims And Evidence:**

Yes

**Requested Changes:**

None.

**Strengths And Weaknesses:**

**Strengths:**
- This paper reflects a monumental amount of work, and the training and evaluation corpora are likely to be very useful.
- The paper is very replicable, almost all aspects of data and model construction are described in great detail.
- The authors embrace an India-first perspective which I think is valuable, especially in benchmark creation.

**Weaknesses:**
- The paper is too long. It felt like content was frequently repeated, and unnecessary details were often provided. Some experimental details (e.g. GPT3.5 translation experiments) are described in the main text, while the experiments themselves appear only in the appendix. Page limits are healthy, and this paper could have benefitted from some forced cuts.
- The paper asks very few research questions -- I try to outline the key questions in my contributions summary above. It is very much a resource paper, and as such, I'm not sure how good a fit it is for TMLR.

**Points for improvement:**
- I found the discussion of data augmentation 7.4 confusing. It was not easy to see the differences between En-Indic and Indic-En that are discussed at length in the text when I examined Figure 6 and Table 17. It sounds like the authors consider data augmentation to be a failure for En-Indic, but I couldn't see why. (Quote: "Although this model is not particularly better than the one obtained using back-translation, it does exhibit better performance...") Maybe a discussion of statistical significance would be helpful?

---

> ### Author Response · Authors · 2023-10-21
> **Rebuttal by Authors**
>
> We thank the reviewer for constructive and valuable comments. We appreciate that the reviewer finds our idea of region-centric benchmark to be interesting, the detailed descriptions of different components of our pipeline to be helpful for its reproducibility and benefits of different artifacts which will be released for further extensions. We provide our responses to the concerns raised by the reviewer below.
>
> ### 1. Paper Length and Reorganization
>
> We appreciate the reviewer's feedback regarding the excessive level of detail in our paper. We included these details in order to provide a comprehensive understanding of the widely used techniques and metrics employed in our work, particularly for readers who may be unfamiliar with these concepts. We will revise the final camera-ready version to restructure the supplementary details to the Appendix.
>
> ### 2. Limited Research Questions addressed
>
> **While our work might have limited theoretical aspects, we believe our research presents interesting empirical findings and practical insights that can be a valuable resource in advancing the field and enabling real-world impact to the field of low-resource machine translation (MT).** Firstly, we conduct a comprehensive analysis of efficient and cost-effective training strategies, including optimal sentence embedding selection, filtering thresholds for mining data, and the efficacy of a multi-stage training and fine-tuning curriculum for training MT models with limited computational resources.
>
> Previous studies [A, B] show that MT performance is influenced by data scale but minimally affected by noisy translation pairs. Recent works, such as NLLB [C], apply a universal filtering threshold across different language groups and consume all mined data for large-scale model training. However, our experiments (see Table 15 in the manuscript) reveal that strict filtering yields superior models with just a fraction of the mined data (20%). Manual verification confirms the presence of misaligned pairs in the NLLB data released by AI2. Our findings show that conducting such language-group-specific studies for different design choices can effectively improve the training efficiency. We believe that our work can serve as a blueprint for conducting similar studies across different language families and consequently, these language-family-specific insights would be extremely useful for improving training efficiency and performance of massively multilingual models such as NLLB in the future.
>
> Furthermore, our work also conducts a robust evaluation of various MT systems across different content types, encompassing general and conversational settings. There are limited benchmarks for evaluating MT systems, and most rely on general-purpose benchmarks like FLORES-200 for evaluation. However, such benchmarks may not be very representative of the demographics of the region where these models would be deployed for real-word usage. Therefore, as a part of our work we also introduce a novel India-centric benchmark specifically designed for this purpose. We believe that our work would also motivate other groups to develop region / demography-centric benchmarks to gain more fine-grained evaluations of the practicality and deployment suitability of  these models in real-life scenarios.
>
> Our work stands out as the first to develop an MT model supporting all 22 scheduled Indian languages. Additionally, we also distill the pre-trained models of 1.2B parameters into 200M parameters without compromising the performance, making them accessible for resource-constrained settings. **We believe the research community, especially those exploring low-resource settings, and leveraging linguistic relatedness, would find value in our work.** Our models, despite their compactness, have proved they can compete with larger multilingual models. **We anticipate our work will stimulate further research interest into such low-resource settings and urge small academic/research groups to strive for developing compact, open-source SOTA models that can compete with massively multilingual models thereby transforming the landscape of multilingual MT models.**
>
> [A] Gordon et al. Data and Parameter Scaling Laws for Neural Machine Translation. EMNLP, 2021.
>
> [B] Bansal et al. Data Scaling Laws in NMT: The Effect of Noise and Architecture. ICML, 2022
>
> [C] Costa-jussà et al. No Language Left Behind: Scaling Human-Centered Machine Translation, ArXiv, 2022.

---

> > ### Author Response · Authors · 2023-10-21
> > **Continuation of Above Rebuttal Response**
> >
> > ### 3. Clarity on data augmentation
> >
> > Thank you for pointing out the ambiguity in our manuscript. We agree that the cited excerpt could lead to misunderstandings due to the lack of elaboration on the forward-translation and back-translation results we are contrasting. To summarize, we get strong performance improvements as a result of data augmentation in the Indic-En direction compared to En-Indic direction. Below, we provide detailed explanation to clarify the misunderstandings and will revise it in the manuscript as well.
> >
> > First of all, we conducted two data augmentation experiments for the En-Indic direction, namely back-translation and forward-translation. Further, we compared the results from these data augmented models (DA) with the best auxiliary model (OG-seed) and choose the DA (forward-translation) model as the best model for further fine-tuning.
> >
> > The cited excerpt in the review highlights that the evaluation results on DA (forward-translation) model are slightly better than the DA (back-translation) model, although both models lag best auxiliary mode (OG-seed), however we choose the DA (forward-ttranslation) model as it is slightly better. The ambiguity in the cited excerpt was resulted as we missed adding the DA (back-translation) model results to the table and apologize for the same. The results are presented below in Table 1.
> >
> > Table 1: Evaluation on IN22-Gen for En-Indic Data Augmentation experiments.
> >
> > | language | OG-seed | DA - BT | DA - FT |
> > |:--------:|:-------:|:-------:|:-------:|
> > | asm_Beng |   45.6  |   45.3  |   44.8  |
> > | ben_Beng |   50.3  |   48.6  |   48.8  |
> > | brx_Deva |   47.1  |   43.2  |   46.3  |
> > | doi_Deva |   55.7  |   52.1  |   56.2  |
> > | gom_Deva |   43.8  |   43.2  |   43.2  |
> > | guj_Gujr |   51.6  |   49.8  |   50.0  |
> > | hin_Deva |   54.6  |   53.7  |   53.6  |
> > | kan_Knda |   49.7  |   47.9  |   47.7  |
> > | kas_Arab |   38.8  |   36.6  |   38.3  |
> > | mai_Deva |   47.3  |   47.3  |   46.2  |
> > | mal_Mlym |   49.7  |   48.5  |   48.4  |
> > | mar_Deva |   48.6  |   46.6  |   46.6  |
> > | mni_Mtei |   41.3  |   38.5  |   41.8  |
> > | npi_Deva |   47.5  |   45.3  |   45.4  |
> > | ory_Orya |   41.9  |   41.0  |   41.0  |
> > | pan_Guru |   50.6  |   49.9  |   50.2  |
> > | san_Deva |   37.3  |   36.0  |   36.9  |
> > | sat_Olck |   27.3  |   12.9  |   26.5  |
> > | snd_Deva |   36.2  |   36.2  |   35.3  |
> > | tam_Taml |   48.7  |   48.2  |   47.9  |
> > | tel_Telu |   51.3  |   49.9  |   50.0  |
> > | urd_Arab |   67.1  |   64.9  |   65.4  |
> > | Avg.         |   46.9  |   44.8  |   45.9  |
> >
> > Further, post seed data fine-tuning, we observe that the DA-seed model is indeed better than the OG-seed model, so data-augmentation is helpful. However, if we compare that with the gains observed in the Indic-En direction (Table 17 or Fig 6 in the manuscript), we observe lower improvements in the En-Indic direction. This is what we wanted to highlight in the aforementioned section. We hope this clarification aids in understanding our approach and findings regarding data augmentation. We will surely rewrite this section to make these details clearer in the revised manuscript.

---

### Author Response · Authors · 2023-10-29
**Joint Reply**

Dear Reviewers,

We are truly thankful for your valuable feedback. We have tried to address all of your concerns in our responses. As the author-reviewer discussion period will end soon (November $4^{th}$), we would love to hear if you still have any concerns and we are more than happy to discuss. We are looking forward to your comments.

---

### Decision · Action_Editor_CddE · 2023-11-09

**Recommendation:** Accept as is

**Comment:**

I think all reviewers are in agreement on two things: (1) this paper answers limited research questions, and (2) this is a resource paper that provides data and benchmarks to the community for a relatively unexplored set of languages which is important. The question is mainly up to the TMLR policy on whether such a dataset contribution is valuable. After discussion with the TMLR chairs, I propose accept since https://jmlr.org/tmlr/editorial-policies.html includes "accounts of applications of existing techniques that shed light on the strengths and weaknesses of the methods" which this falls under.

**Audience:**

Reviewers GByW and zBPR both expressed doubt about the appropriateness of this paper, which is primarily a dataset contribution, for TMLR, given that dataset contributions are not explicitly listed in https://jmlr.org/tmlr/editorial-policies.html.

That said, I believe the larger community will find these data pipelining methods useful. In the era of large models, the data pipeline is ever more important. Indics in particular are a greenfield opportunity given the number of people speaking those languages and the lack of available data. As reveiwer SHNd says, "The proposed techniques achieve impressive performance compared with existing large language models. Those proposed models would be beneficial to the community."

**Claims And Evidence:**

This paper claims to contribute large training sets, diverse and high quality benchmarks, and trained models for machine translation in 22 indic languages. They release these resources through open source along with evaluations, hence backing up their claims.